# Combinatorial Pure Exploration with Bottleneck Reward Function

**Yihan Du**
IIIS, Tsinghua University
Beijing, China
duyh18@mails.tsinghua.edu.cn

**Yuko Kuroki**
The University of Tokyo / RIKEN
Tokyo, Japan
yukok@is.s.u-tokyo.ac.jp

**Wei Chen**
Microsoft Research
Beijing, China
weic@microsoft.com

## Abstract

In this paper, we study the Combinatorial Pure Exploration problem with the Bottleneck reward function (CPE-B) under the fixed-confidence (FC) and fixed-budget (FB) settings. In CPE-B, given a set of base arms and a collection of subsets of base arms (super arms) following a certain combinatorial constraint, a learner sequentially plays a base arm and observes its random reward, with the objective of finding the optimal super arm with the maximum bottleneck value, defined as the minimum expected reward of the base arms contained in the super arm. CPE-B captures a variety of practical scenarios such as network routing in communication networks, and its *unique challenges* fall on how to utilize the bottleneck property to save samples and achieve the statistical optimality. None of the existing CPE studies (most of them assume linear rewards) can be adapted to solve such challenges, and thus we develop brand-new techniques to handle them. For the FC setting, we propose novel algorithms with optimal sample complexity for a broad family of instances and establish a matching lower bound to demonstrate the optimality (within a logarithmic factor). For the FB setting, we design an algorithm which achieves the state-of-the-art error probability guarantee and is the first to run efficiently on fixed-budget path instances, compared to existing CPE algorithms. Our experimental results on the top-$k$, path and matching instances validate the empirical superiority of the proposed algorithms over their baselines.

## 1 Introduction

The Multi-Armed Bandit (MAB) problem [25, 30, 4, 2] is a classic model to solve the exploration-exploitation trade-off in online decision making. Pure exploration [3, 21, 7, 26] is an important variant of the MAB problem, which aims to identify the best arm under a given confidence or a given sample budget. There are various works studying pure exploration, such as top-$k$ arm identification [17, 21, 7, 24], top-$k$ arm under matriod constraints [9] and multi-bandit best arm identification [18, 7].

The Combinatorial Pure Exploration (CPE) framework, firstly proposed by Chen et al. [11], encompasses a rich class of pure exploration problems [3, 21, 9]. In CPE, there are a set of base arms, each associated with an unknown reward distribution. A subset of base arms is called a super arm, which follows a certain combinatorial structure. At each timestep, a learner plays a base arm and observes a random reward sampled from its distribution, with the objective to identify the optimal super arm

35th Conference on Neural Information Processing Systems (NeurIPS 2021).

with the maximum expected reward. While Chen et al. [11] provide this general CPE framework, their algorithms and analytical techniques only work under the linear reward function and cannot be applied to other nonlinear reward cases.[1]

However, in many real-world scenarios, the expected reward function is not necessarily linear. One of the common and important cases is the *bottleneck reward* function, i.e., the expected reward of a super arm is the minimum expected reward of the base arms contained in it. For example, in communication networks [5], the transmission speed of a path is usually determined by the link with the lowest rate, and a learner samples the links in order to find the optimal transmission path which maximizes its bottleneck link rate. In traffic scheduling [31], a scheduling system collects the information of road segments in order to plan an efficient route which optimizes its most congested (bottleneck) road segment. In neural architecture search [32], the overall efficiency of a network architecture is usually constrained by its worst module, and an agent samples the available modules with the objective to identify the best network architecture in combinatorial search space.

In this paper, we study the Combinatorial Pure Exploration with the Bottleneck reward function (CPE-B) which aims to identify the optimal super arm with the maximum bottleneck value by querying the base arm rewards, where the bottleneck value of a super arm is defined as the minimum expected reward of its containing base arms. We consider two popular settings in pure exploration, i.e, *fixed-confidence (FC)*, where given confidence parameter $\delta$, the learner aims to identify the optimal super arm with probability $1 - \delta$ and minimize the number of used samples (sample complexity), and *fixed-budget (FB)*, where the learner needs to use a given sample budget to find the optimal super arm and minimize the error probability.

**Challenges of CPE-B.** Compared to prior CPE works [11, 10, 20], our CPE-B aims at utilizing the bottleneck property to save samples and achieve the statistical optimality. It faces with two *unique challenges*, i.e., how to (i) achieve the tight *base-arm-gap dependent* sample complexity and (ii) avoid the dependence on *unnecessary base arms* in the results, while running in polynomial time. We use a simple example in Figure 1 to illustrate our challenges. In Figure 1, there are six edges (base arms) and three $s$-$t$ paths (super arms), and the base arm reward $w(e_i)$, base arm gap $\Delta_{e_i,e_j}$ and super arm gap $\Delta_{M_*,M_{\text{sub}}}$ are as shown in the figure. In order to identify the optimal path, all we

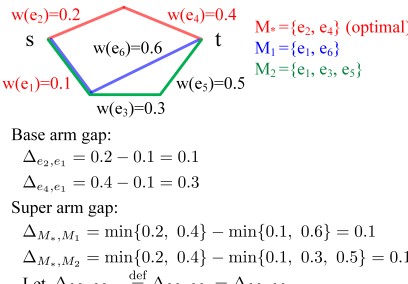

Base arm gap:
$\Delta_{e_2,e_1} = 0.2 - 0.1 = 0.1$
$\Delta_{e_4,e_1} = 0.4 - 0.1 = 0.3$
Super arm gap:
$\Delta_{M_*,M_1} = \min\{0.2,\ 0.4\} - \min\{0.1,\ 0.6\} = 0.1$
$\Delta_{M_*,M_2} = \min\{0.2,\ 0.4\} - \min\{0.1,\ 0.3,\ 0.5\} = 0.1$
Let $\Delta_{M_*,M_{\text{sub}}} \stackrel{\text{def}}{=} \Delta_{M_*,M_1} = \Delta_{M_*,M_2}$

Figure 1: Illustrating example.

need is to pull $e_1, e_2, e_4$ to determine that $e_1$ is worse than $e_2$ and $e_4$, and $e_3, e_5, e_6$ are useless for revealing the sub-optimality of $M_1$ and $M_2$. In this case, the optimal sample complexity should be $O((\frac{2}{\Delta_{e_2,e_1}^2} + \frac{1}{\Delta_{e_4,e_1}^2}) \ln \delta^{-1})$, which depends on the tight base arm gaps and only includes the critical base arms ($e_1, e_2, e_4$). However, if one naively adapts existing CPE algorithms [11, 12, 16] to work with bottleneck reward function, an inferior sample complexity of $O(\sum_{e_i,i\in[6]} \frac{1}{\Delta_{M_*,M_{\text{sub}}}^2} \ln \delta^{-1})$ is

incurred, which depends on the loose super arm gaps and contains a summation over all base arms (including the unnecessary $e_3, e_5, e_6$). Hence, our challenge falls on how to achieve such efficient sampling in an online environment, where we do not know which are critical base arms $e_1, e_2, e_4$ but want to gather just enough information to identify the optimal super arm. We remark that, none of existing CPE studies can be applied to solve the unique challenges of CPE-B, and thus we develop brand-new techniques to handle them and attain the optimal results (up to a logarithmic factor).

**Contributions.** For CPE-B in the FC setting, (i) we first develop a novel algorithm BLUCB, which employs a bottleneck-adaptive sample strategy and achieves the tight base-arm-gap dependent sample complexity. (ii) We further propose an improved algorithm BLUCB-Parallel in high confidence regime, which adopts an efficient "bottleneck-searching" offline procedure and a novel "check-near-bottleneck" stopping condition. The sample complexity of BLUCB-Parallel drops the dependence on unnecessary base arms and achieves the optimality (within a logarithmic factor) under small enough $\delta$. (iii) A matching sample complexity lower bound for the FC setting is also provided, which demonstrates the optimality of our algorithms. For the FB setting, (iv) we propose a novel algorithm BSAR with a special acceptance scheme for the bottleneck identification task. BSAR achieves the state-of-the-art error probability and is the first to run efficiently on fixed-budget path instances,

---

[1]The algorithmic designs and analytical tools (e.g., symmetric difference and exchange set) in [11] all rely on the linear property and cannot be applied to nonlinear reward cases, e.g, the bottleneck reward problem.

compared to existing CPE algorithms. All our proposed algorithms run in *polynomial time*.[2] The experimental results demonstrate that our algorithms significantly outperform the baselines. Due to space limit, we defer all the proofs to the supplementary material.

## 1.1 Related Work

In the following we briefly review the related work in the CPE literature. Chen et al. [11] firstly propose the CPE model and only consider the linear reward function (CPE-L), and their results for CPE-L are further improved by [19, 10]. Huang et al. [20] investigate the continuous and separable reward functions (CPE-CS), but their algorithm only runs efficiently on simple cardinality constraint instances. All these works consider directly sampling base arms and getting their feedback. There are also several CPE studies which consider other forms of sampling and feedback. Chen et al. [12] propose the CPE for dueling bandit setting, where at each timestep the learner pulls a duel between two base arms and observes their comparison outcome. Kuroki et al. [23] study an online densest subgraph problem, where the decision is a subgraph and the feedback is the reward sum of the edges in the chosen subgraph (i.e., full-bandit feedback). Du et al. [16] investigate CPE with the full-bandit or partial linear feedback. All of the above studies consider the pure exploration setting, while in combinatorial bandits there are other works [13, 15, 14] studying the regret minimization setting (CMAB). In CMAB, the learner plays a super arm and observes the rewards from all base arms contained in it, with goal of minimizing the regret, which is significantly different from our setting. Note that none of the above studies covers our CPE-B problem or can be adapted to solve the unique challenges of CPE-B, and thus CPE-B demands a new investigation.

## 2 Problem Formulation

In this section, we give the formal formulation of CPE-B. In this problem, a learner is given $n$ base arms numbered by $1, 2, \ldots, n$. Each base arm $e \in [n]$ is associated with an *unknown* reward distribution with the mean of $w(e)$ and an $R$-sub-Gaussian tail, which is a standard assumption in bandits [1, 11, 26, 29]. Let $\boldsymbol{w} = (w(1), \ldots, w(n))^\top$ be the expected reward vector of base arms. The learner is also given a decision class $\mathcal{M} \subseteq 2^{[n]}$, which is a collection of super arms (subsets of base arms) and generated from a certain combinatorial structure, such as $s$-$t$ paths, maximum cardinality matchings, and spanning trees. For each super arm $M \in \mathcal{M}$, we define its expected reward (also called *bottleneck value*) as $\texttt{MinW}(M, \boldsymbol{w}) = \min_{e \in M} w(e)$,[3] i.e., the minimum expected reward of its constituent base arms, which is so called *bottleneck reward function*. Let $M_* = \text{argmax}_{M \in \mathcal{M}} \texttt{MinW}(M, \boldsymbol{w})$ be the optimal super arm with the maximum bottleneck value, and $\texttt{OPT} = \texttt{MinW}(M_*, \boldsymbol{w})$ be the optimal value. Following the pure exploration literature [17, 11, 10, 12], we assume that $M_*$ is unique, and this assumption can be removed in our extension to the PAC learning setting (see the supplementary material).

At each timestep, the learner plays (or samples) a base arm $p_t \in [n]$ and observes a random reward sampled from its reward distribution, where the sample is independent among different timestep $t$. The learner's objective is to identify the optimal super arm $M_*$ from $\mathcal{M}$.

For this identification task, we study two common metrics in pure exploration [21, 7, 26, 10], i.e., fixed-confidence (FC) and fixed-budget (FB) settings. In the FC setting, given a confidence parameter $\delta \in (0, 1)$, the learner needs to identify $M_*$ with probability at least $1 - \delta$ and minimize the *sample complexity*, i.e., the number of samples used. In the FB setting, the learner is given a fixed sample budget $T$, and needs to identify $M_*$ within $T$ samples and minimize the *error probability*, i.e., the probability of returning a wrong answer.

## 3 Algorithms for the Fixed-Confidence Setting

In this section, we first propose a simple algorithm BLUCB for the FC setting, which adopts a novel bottleneck-adaptive sample strategy to obtain the tight base-arm-gap dependent sample complexity.

---

[2]Here "polynomial time" refers to polynomial time in the number of base arms $n$ (which is equal to the number of edges $E$ in our considered instances such as $s$-$t$ paths, matchings and spanning trees).

[3]In general, the second input of function MinW can be any vector: for any $M \in \mathcal{M}$ and $\boldsymbol{v} \in \mathbb{R}^n$, $\texttt{MinW}(M, \boldsymbol{v}) = \min_{e \in M} v(e)$.

---

**Algorithm 1** BLUCB, algorithm for CPE-B in the FC setting

---

1: **Input:** $\mathcal{M}$, $\delta \in (0, 1)$ and `MaxOracle`.
2: Initialize: play each $e \in [n]$ once, and update empirical means $\hat{w}_{n+1}$ and $T_{n+1}$
3: **for** $t = n + 1, n + 2, \ldots$ **do**
4:   $\mathrm{rad}_t(e) \leftarrow \sqrt{2\ln(\frac{4nt^3}{\delta})/T_t(e)}$, $\forall e \in [n]$
5:   $\underline{w}_t(e) \leftarrow \hat{w}_t(e) - \mathrm{rad}_t(e)$, $\forall e \in [n]$
6:   $\bar{w}_t(e) \leftarrow \hat{w}_t(e) + \mathrm{rad}_t(e)$, $\forall e \in [n]$
7:   $M_t \leftarrow \mathtt{MaxOracle}(\mathcal{M}, \underline{w}_t)$
8:   $\tilde{M}_t \leftarrow \mathtt{MaxOracle}(\mathcal{M} \setminus \mathcal{S}(M_t), \bar{w}_t)$
9:   **if** $\mathtt{MinW}(M_t, \underline{w}_t) \geq \mathtt{MinW}(\tilde{M}_t, \bar{w}_t)$ **then**
10:    **return** $M_t$
11:   **end if**
12:   $c_t \leftarrow \mathrm{argmin}_{e \in M_t} \underline{w}_t(e)$
13:   $d_t \leftarrow \mathrm{argmin}_{e \in \tilde{M}_t} \underline{w}_t(e)$
14:   $p_t \leftarrow \mathrm{argmax}_{e \in \{c_t, d_t\}} \mathrm{rad}_t(e)$
15:   Play $p_t$, and observe the reward
16:   Update empirical means $\hat{w}_{t+1}(p_t)$
17:   Update the number of samples $T_{t+1}(p_t)$
18: **end for**

---

We further develop an improvement BLUCB-Parallel in high confidence regime, whose sample complexity drops the dependence on unnecessary base arms for small enough $\delta$. Both algorithms achieve the optimal sample complexity for a family of instances (within a logarithmic factor).

### 3.1   Algorithm BLUCB with Base-arm-gap Dependent Results

Algorithm 1 illustrates the proposed algorithm BLUCB for CPE-B in the FC setting. Here $\mathcal{S}(M_t)$ denotes the set of all supersets of super arm $M_t$ (Line 8). Since the bottleneck reward function is monotonically decreasing, for any $M' \in \mathcal{S}(M_t)$, we have $\mathtt{MinW}(M', \boldsymbol{w}) \leq \mathtt{MinW}(M_t, \boldsymbol{w})$. Hence, to verify the optimality of $M_t$, we only need to compare $M_t$ against super arms in $\mathcal{M} \setminus \mathcal{S}(M_t)$, and this property will also be used in the later algorithm BLUCB-Parallel.

BLUCB is allowed to access an efficient *bottleneck maximization oracle* $\mathtt{MaxOracle}(\mathcal{F}, \boldsymbol{v})$, which returns an optimal super arm from $\mathcal{F}$ with respect to $\boldsymbol{v}$, i.e., $\mathtt{MaxOracle}(\mathcal{F}, \boldsymbol{v}) \in \mathrm{argmax}_{M \in \mathcal{F}} \mathtt{MinW}(M, \boldsymbol{v})$. For $\mathcal{F} = \mathcal{M}$ (Line 7), such an efficient oracle exists for many decision classes, such as the bottleneck shortest path [27], bottleneck bipartite matching [28] and minimum bottleneck spanning tree [8] algorithms. For $\mathcal{F} = \mathcal{M} \setminus \mathcal{S}(M_t)$ (Line 8), we can also efficiently find the best super arm (excluding the supersets of $M_t$) by repeatedly removing each base arm in $M_t$ and calling the basic maximization oracle, and then selecting the one with the maximum bottleneck value.

We describe the procedure of BLUCB as follows: at each timestep $t$, we calculate the lower and upper confidence bounds of base arm rewards, denoted by $\underline{w}_t$ and $\bar{w}_t$, respectively. Then, we call MaxOracle to find the super arm $M_t$ with the maximum pessimistic bottleneck value from $\mathcal{M}$ using $\underline{w}_t$ (Line 7), and the super arm $\tilde{M}_t$ with the maximum optimistic bottleneck value from $\mathcal{M} \setminus \mathcal{S}(M_t)$ using $\bar{w}_t$ (Line 8). $M_t$ and $\tilde{M}_t$ are two critical super arms that determine when the algorithm should stop or not. If the pessimistic bottleneck value of $M_t$ is higher than the optimistic bottleneck value of $\tilde{M}_t$ (Line 9), we can determine that $M_t$ has the higher bottleneck value than any other super arm with high confidence, and then the algorithm can stop and output $M_t$. Otherwise, we select two base arms $c_t$ and $d_t$ with the minimum lower reward confidence bounds in $M_t$ and $\tilde{M}_t$ respectively, and play the one with the larger confidence radius (Lines 12-14).

**Bottleneck-adaptive sample strategy.** The "select-minimum" sample strategy in Lines 12-14 comes from an *insight* for the bottleneck problem: to determine that $M_t$ has a higher bottleneck value than $\tilde{M}_t$, it suffices to find a base arm from $\tilde{M}_t$ which is worse than any base arm (the bottleneck base arm) in $M_t$. To achieve this, base arms $c_t$ and $d_t$, which have the most potential to be the bottlenecks of $M_t$ and $\tilde{M}_t$, are the most necessary ones to be sampled. This bottleneck-adaptive sample strategy is crucial for BLUCB to achieve the tight base-arm-gap dependent sample complexity. In contrast, the sample strategy of prior CPE algorithms [11, 12, 16] treats all base arms in critical super arms ($M_t$ and $\tilde{M}_t$) equally and does a uniform choice. If one naively adapts those algorithms with the current reward function $\mathtt{MinW}(M, \boldsymbol{w})$, a loose super-arm-gap dependent sample complexity is incurred.

To formally state the sample complexity of BLUCB, we introduce some notation and gap definition. Let $N = \{e \mid e \notin M_*, w(e) < \mathtt{OPT}\}$ and $\tilde{N} = \{e \mid e \notin M_*, w(e) \geq \mathtt{OPT}\}$, which stand for the necessary and *unnecessary* base arms contained in the sub-optimal super arms, respectively. We define the reward gap for the FC setting as

---

**Algorithm 2** `BLUCB-Parallel`, an improved algorithm for the FC setting under small $\delta$

---

1: **Input:** $\delta \in (0, 0.01)$ and sub-algorithm `BLUCB-Verify`.
2: For $k = 0, 1, \ldots$, let `BLUCB-Verify`$_k$ be the sub-algorithm `BLUCB-Verify` with $\delta_k = \frac{\delta}{2^{k+1}}$
3: **for** $t = 1, 2, \ldots$ **do**
4:    **for** each $k = 0, 1, \ldots$ such that $t \bmod 2^k = 0$ **do**
5:       Start or resume `BLUCB-Verify`$_k$ with one sample, and then suspend `BLUCB-Verify`$_k$
6:       **if** `BLUCB-Verify`$_k$ returns an answer $M_{\text{out}}$, then **return** $M_{\text{out}}$
7:    **end for**
8: **end for**

---

**Algorithm 3** `BLUCB-Verify`, sub-algorithm of `BLUCB-Parallel`

---

1: **Input:** $\mathcal{M}, \delta^V \in (0, 0.01)$ and `MaxOracle`.
2: $\kappa \leftarrow 0.01$
3: $\hat{M}_*, \hat{B}_{\text{sub}} \leftarrow$ `BLUCB-Explore`$(\mathcal{M}, \kappa, $`MaxOracle`$)$
4: Initialize: play each $e \in [n]$ once, and update empirical means $\hat{w}_{n+1}$ and $T_{n+1}$
5: **for** $t = n + 1, n + 2, \ldots$ **do**
6:    $\text{rad}_t(e) \leftarrow R\sqrt{2\ln(\frac{4nt^3}{\delta^V})/T_t(e)}, \forall e \in [n]$
7:    $\underline{w}_t(e) \leftarrow \hat{w}_t(e) - \text{rad}_t(e), \ \forall e \in [n]$
8:    $\bar{w}_t(e) \leftarrow \hat{w}_t(e) + \text{rad}_t(e), \ \forall e \in [n]$
9:    $\tilde{M}_t = $ `MaxOracle`$(\mathcal{M} \setminus \mathcal{S}(\hat{M}_*), \bar{w}_t)$
10:    **if** `MinW`$(\hat{M}_*, \underline{w}_t) \geq$ `MinW`$(\tilde{M}_t, \bar{w}_t)$ **then**
11:       **return** $\hat{M}_*$
12:    **end if**
13:    $c_t \leftarrow \text{argmin}_{e \in \hat{M}_*} \underline{w}_t(e)$
14:    $F_t \leftarrow \{e \in \hat{B}_{\text{sub}} : \bar{w}_t(e) > \underline{w}_t(c_t)\}$
15:    $p_t \leftarrow \text{argmax}_{e \in F_t \cup \{c_t\}} \text{rad}_t(e)$
16:    Play $p_t$, and observe the reward
17:    Update empirical means $\hat{w}_{t+1}(p_t)$
18:    Update the number of samples $T_{t+1}(p_t)$
19: **end for**

---

**Definition 1** (Fixed-confidence Gap).

$$\Delta_e^{\text{C}} = \begin{cases} w(e) - \max_{M \neq M_*} \text{MinW}(M, \boldsymbol{w}), & \text{if } e \in M_*, & \text{(a)} \\ w(e) - \max_{M \in \mathcal{M}:e \in M} \text{MinW}(M, \boldsymbol{w}), & \text{if } e \in \tilde{N}, & \text{(b)} \\ \text{OPT} - \max_{M \in \mathcal{M}:e \in M} \text{MinW}(M, \boldsymbol{w}), & \text{if } e \in N. \end{cases}$$

Now we present the sample complexity upper bound of BLUCB.

**Theorem 1** (Fixed-confidence Upper Bound). *With probability at least $1 - \delta$, algorithm* BLUCB *(Algorithm 1) for CPE-B in the FC setting returns the optimal super arm with sample complexity*

$$O\left(\sum_{e \in [n]} \frac{R^2}{(\Delta_e^{\text{C}})^2} \ln\left(\sum_{e \in [n]} \frac{R^2 n}{(\Delta_e^{\text{C}})^2 \delta}\right)\right).$$

**Base-arm-gap dependent sample complexity.** Owing to the bottleneck-adaptive sample strategy, the reward gap $\Delta_e^{\text{C}}$ (Definition 1(a)(b)) is just defined as the difference between some critical bottleneck value and $w(e)$ itself, instead of the bottleneck gap between two super arms, and thus our result depends on the tight base-arm-level (instead of super-arm-level) gaps. For example, in Figure 1, BLUCB only spends $\tilde{O}((\frac{2}{\Delta_{e_2,e_1}^2} + \sum_{i=3,4,5,6} \frac{1}{\Delta_{e_i,e_1}^2}) \ln \delta^{-1})$ samples, while a naive adaptation of prior CPE algorithms [11, 12, 16] with the bottleneck reward function will cause a loose super-arm-gap dependent result $\tilde{O}(\sum_{e_i,i \in [6]} \frac{1}{\Delta_{M_*,M_{\text{sub}}}^2} \ln \delta^{-1})$. Regarding the optimality, Theorem 1 matches the lower bound (presented in Section 4) for some family of instances (up to a logarithmic factor). However, in general cases there still exists a gap on those needless base arms $\tilde{N}$ ($e_3, e_5, e_6$ in Figure 1), which are not contained in the lower bound. Next, we show how to bridge this gap.

### 3.2 Remove Dependence on Unnecessary Base Arms under Small $\delta$

**Challenges of avoiding unnecessary base arms.** Under the bottleneck reward function, in each sub-optimal super arm $M_{\text{sub}}$, only the base arms with rewards lower than OPT (base arms in $N$) can determine the relationship of bottleneck values between $M_*$ and $M_{\text{sub}}$ (the bottleneck of $M_{\text{sub}}$ is the most efficient choice to do this), and the others (base arms in $\tilde{N}$) are useless for revealing the sub-optimality of $M_{\text{sub}}$. Hence, to determine $M_*$, all we need is to sample the base arms in $M_*$ and the *bottlenecks from all sub-optimal super arms*, denoted by $B_{\text{sub}}$, to see that each sub-optimal super

---

**Algorithm 4** `BLUCB-Explore`, sub-algorithm of `BLUCB-Verify`, the *key algorithm*

---

1: **Input:** $\mathcal{M}$, $\kappa = 0.01$ and `MaxOracle`.
2: Initialize: play each $e \in [n]$ once, and update empirical means $\hat{w}_{n+1}$ and $T_{n+1}$
3: **for** $t = n+1, n+2, \dots$ **do**
4:    $\text{rad}_t(e) \leftarrow R\sqrt{2\ln(\frac{4nt^3}{\kappa})/T_t(e)}, \ \forall e \in [n]$
5:    $\underline{w}_t(e) \leftarrow \hat{w}_t(e) - \text{rad}_t(e), \ \forall e \in [n]$
6:    $\bar{w}_t(e) \leftarrow \hat{w}_t(e) + \text{rad}_t(e), \ \forall e \in [n]$
7:    $M_t \leftarrow \text{MaxOracle}(\mathcal{M}, \underline{w}_t)$
8:    $\hat{B}_{\text{sub},t} \leftarrow \text{BottleneckSearch}(\mathcal{M}, M_t, \underline{w}_t)$
9:    **if** $\bar{w}_t(e) \le \frac{1}{2}(\text{MinW}(M_t, \underline{w}_t) + \underline{w}_t(e))$ for all $e \in \hat{B}_{\text{sub},t}$ **then**
10:        **return** $M_t, \hat{B}_{\text{sub},t}$
11:    **end if**
12:    $c_t \leftarrow \text{argmin}_{e \in M_t} \underline{w}_t(e)$
13:    $\hat{B}'_{\text{sub},t} \leftarrow \{e \in \hat{B}_{\text{sub},t} : \bar{w}_t(e) > \frac{1}{2}(\text{MinW}(M_t, \underline{w}_t) + \underline{w}_t(e))\}$
14:    $p_t \leftarrow \text{argmax}_{e \in \hat{B}'_{\text{sub},t} \cup \{c_t\}} \text{rad}_t(e)$
15:    Play $p_t$, and observe the reward
16:    Update empirical means $\hat{w}_{t+1}(p_t)$
17:    Update the number of samples $T_{t+1}(p_t)$
18: **end for**

---

arm contains at least one base arm that is worse than anyone in $M_*$. However, before sampling, (i) we do not know which is $M_*$ that should be taken as the comparison benchmark, and in each $M_{\text{sub}}$, which base arm is its bottleneck (included in $B_{\text{sub}}$). Also, (ii) under combinatorial setting, how to efficiently collect $B_{\text{sub}}$ from all sub-optimal super arms is another challenge.

To handle these challenges, we propose algorithm `BLUCB-Parallel` based on the explore-verify-parallel framework [22, 10]. `BLUCB-Parallel` (Algorithm 2) simultaneously simulates multiple `BLUCB-Verify`$_k$ (Algorithm 3) with confidence $\delta_k^V = \delta/2^{k+1}$ for $k \in \mathbb{N}$. `BLUCB-Verify`$_k$ first calls `BLUCB-Explore` (Algorithm 4) to guess an optimal super arm $\hat{M}_*$ and collect a *near bottleneck set* $\hat{B}_{\text{sub}}$ with *constant confidence* $\kappa$, and then uses the required confidence $\delta_k^V$ to verify the correctness of $\hat{M}_*$ by only sampling base arms in $\hat{M}_*$ and $\hat{B}_{\text{sub}}$. Through parallel simulations, `BLUCB-Parallel` guarantees the $1 - \delta$ correctness.

The *key component* of this framework is `BLUCB-Explore` (Algorithm 4), which provides a hypothesized answer $\hat{M}_*$ and critical base arms $\hat{B}_{\text{sub}}$ for verification to accelerate its identification process. Below we first describe the procedure of `BLUCB-Explore`, and then explicate its two *innovative techniques*, i.e. offline subroutine and stopping condition, developed to handle the challenges (i),(ii). `BLUCB-Explore` employs the subroutine $\text{BottleneckSearch}(\mathcal{M}, M_{\text{ex}}, \boldsymbol{v})$ to return the set of bottleneck base arms from all super arms in $\mathcal{M} \setminus \mathcal{S}(M_{\text{ex}})$ with respect to weight vector $\boldsymbol{v}$. At each timestep, we first calculate the best super arm $M_t$ under lower reward confidence bound $\underline{w}_t$, and call $\text{BottleneckSearch}$ to collect the bottlenecks $\hat{B}_{\text{sub},t}$ from all super arms in $\mathcal{M} \setminus \mathcal{S}(M_t)$ with respect to $\underline{w}_t$ (Line 8). Then, we use a stopping condition (Line 9) to examine if $M_t$ is correct and $\hat{B}_{\text{sub},t}$ is close enough to $\hat{B}_{\text{sub}}$ (with confidence $\kappa$). If so, $M_t$ and $\hat{B}_{\text{sub},t}$ are eligible for verification and returned; otherwise, we play a base arm from $M_t$ and $\hat{B}_{\text{sub},t}$, which is most necessary for achieving the stopping condition. In the following, we explicate the two innovative techniques in `BLUCB-Explore`.

**Efficient "bottleneck-searching" offline subroutine.** $\text{BottleneckSearch}(\mathcal{M}, M_{\text{ex}}, \boldsymbol{v})$ (Line 8) serves as an efficient offline procedure to collect bottlenecks from all super arms in given decision class $\mathcal{M} \setminus \mathcal{S}(M_{\text{ex}})$ with respect to $\boldsymbol{v}$. To achieve efficiency, the main idea behind $\text{BottleneckSearch}$ is to avoid enumerating super arms in the combinatorial space, but only enumerate base arms $e \in [n]$ to check if $e$ is the bottleneck of some super arm in $\mathcal{M} \setminus \mathcal{S}(M_{\text{ex}})$. We achieve this by removing all base arms with rewards lower than $v(e)$ and examining whether there exists a feasible super arm $M$ that contains $e$ in the remaining decision class. If so, $e$ is the bottleneck of $M$ and added to the output (more procedures are designed to exclude $\mathcal{S}(M_{\text{ex}})$). This efficient offline subroutine solves challenge (ii) on computation complexity (see the supplementary material for its pseudo-codes and details).

**Delicate "check-near-bottleneck" stopping condition.** The stopping condition (Line 9) aims to ensure the returned $\hat{B}_{\text{sub},t} = \hat{B}_{\text{sub}}$ to satisfy the following Property (1): for each sub-optimal super arm $M_{\text{sub}}$, some base arm $e$ such that $w(e) \le \frac{1}{2}(\text{MinW}(M_*, \boldsymbol{w}) + \text{MinW}(M_{\text{sub}}, \boldsymbol{w}))$ is included in $\hat{B}_{\text{sub}}$, which implies that $e$ is near to the actual bottleneck of $M_{\text{sub}}$ within $\frac{1}{2}\Delta_{M_*, M_{\text{sub}}}$, and cannot be anyone in $\tilde{N}$. Property (1) is crucial for `BLUCB-Verify` to achieve the optimal sample complexity, since it guarantees that in verification using $\hat{B}_{\text{sub}}$ to verify $M_*$ just costs the same order of samples as

using $B_{\text{sub}}$, which matches the lower bound. In the following, we explain why this stopping condition can guarantee Property (1).

If the stopping condition (Line 9) holds, i.e., $\forall e \in \hat{B}_{\text{sub},t}, \bar{w}_t(e) \leq \frac{1}{2}(\text{MinW}(M_t, \underline{\boldsymbol{w}}_t) + \underline{w}_t(e))$, using the definition of BottleneckSearch, we have that for any $M' \in \mathcal{M} \setminus \mathcal{S}(M_t)$, its bottleneck $e'$ with respect to $\underline{\boldsymbol{w}}_t$ is included in $\hat{B}_{\text{sub},t}$ and satisfies that

$$w(e') \leq \bar{w}_t(e') \overset{(a)}{\leq} \frac{1}{2}(\text{MinW}(M_t, \underline{\boldsymbol{w}}_t) + \underline{w}_t(e')) \leq \frac{1}{2}(\text{MinW}(M_t, \boldsymbol{w}) + \text{MinW}(M', \boldsymbol{w})),$$

where inequality (a) comes from $\bar{w}_t(e') \leq \frac{1}{2}(\text{MinW}(M_t, \underline{\boldsymbol{w}}_t) + \underline{w}_t(e'))$ and $\underline{w}_t(e') = \text{MinW}(M', \underline{\boldsymbol{w}}_t)$. Hence, we can defer that $\text{MinW}(M', \boldsymbol{w}) \leq w(e') \leq \frac{1}{2}(\text{MinW}(M_t, \boldsymbol{w}) + \text{MinW}(M', \boldsymbol{w}))$ for any $M' \in \mathcal{M} \setminus \mathcal{S}(M_t)$, and thus $M_t = M_*$ (with confidence $\kappa$). In addition, the returned $\hat{B}_{\text{sub},t}$ satisfies Property (1). This stopping condition offers knowledge of a hypothesized optimal super arm $\hat{M}_*$ and a near bottleneck set $\hat{B}_{\text{sub}}$ for verification, which solves the challenge (i) and enables the overall sample complexity to achieve the optimality for small enough $\delta$. Note that these two techniques are new in the literature, which are specially designed for handling the unique challenges of CPE-B.

We formally state the sample complexity of BLUCB-Parallel in Theorem 2.

**Theorem 2** (Improved Fixed-confidence Upper Bound). *For any $\delta < 0.01$, with probability at least $1 - \delta$, algorithm BLUCB-Parallel (Algorithm 2) for CPE-B in the FC setting returns $M_*$ and takes the expected sample complexity*

$$O\left( \sum_{e \in M_* \cup N} \frac{R^2}{(\Delta_e^C)^2} \ln\left( \frac{1}{\delta} \sum_{e \in M_* \cup N} \frac{R^2 n}{(\Delta_e^C)^2} \right) + \sum_{e \in \tilde{N}} \frac{R^2}{(\Delta_e^C)^2} \ln\left( \sum_{e \in \tilde{N}} \frac{R^2 n}{(\Delta_e^C)^2} \right) \right).$$

**Results without dependence on $\tilde{N}$ in the dominant term.** Let $H_V = \sum_{e \in M_* \cup N} \frac{R^2}{(\Delta_e^C)^2}$ and $H_E = \sum_{e \in [n]} \frac{R^2}{(\Delta_e^C)^2}$ denote the verification and exploration hardness, respectively. Compared to BLUCB (Theorem 1), the sample complexity of BLUCB-Parallel removes the redundant dependence on $\tilde{N}$ in the $\ln \delta^{-1}$ term, which guarantees better performance when $\ln \delta^{-1} \geq \frac{H_E}{H_E - H_V}$, i.e., $\delta \leq \exp(-\frac{H_E}{H_E - H_V})$. This sample complexity matches the lower bound (within a logarithmic factor) under small enough $\delta$. For the example in Figure 1, BLUCB-Parallel only requires $\tilde{O}((\frac{2}{\Delta_{e_2,e_1}^2} + \frac{1}{\Delta_{e_4,e_1}^2}) \ln \delta^{-1})$ samples, which are just enough efforts (optimal) for identifying $M_*$.

The condition $\delta < 0.01$ in Theorem 2 is due to that the used explore-verify-parallel framework [22, 10] needs a small $\delta$ to guarantee that BLUCB-Parallel can maintain the same order of sample complexity as its sub-algorithm BLUCB-Verify$_k$. Prior pure exploration works [22, 10] also have such condition on $\delta$.

**Time Complexity.** All our algorithms can run in polynomial time, and the running time mainly depends on the offline oracles. For example, on $s$-$t$ path instances with $E$ edges and $V$ vertices, the used offline procedures MaxOracle and BottleneckSearch only spend $O(E)$ and $O(E^2(E + V))$ time, respectively. See the supplementary material for more time complexity analysis.

## 4 Lower Bound for the Fixed-Confidence Setting

In this section, we establish a matching sample complexity lower bound for CPE-B in the FC setting. To formally state our results, we first define the notion of $\delta$-*correct algorithm* as follows. For any confidence parameter $\delta \in (0, 1)$, we call an algorithm $\mathcal{A}$ a $\delta$-correct algorithm if for the fixed-confidence CPE-B problem, $\mathcal{A}$ returns the optimal super arm with probability at least $1 - \delta$.

**Theorem 3** (Fixed-confidence Lower Bound). *There exists a family of instances for the fixed-confidence CPE-B problem, for which given any $\delta \in (0, 0.1)$, any $\delta$-correct algorithm has the expected sample complexity*

$$\Omega\left( \sum_{e \in M_* \cup N} \frac{R^2}{(\Delta_e^C)^2} \ln\left( \frac{1}{\delta} \right) \right).$$

This lower bound demonstrates that the sample complexity of BLUCB-Parallel (Theorem 2) is optimal (within a logarithmic factor) under small enough $\delta$, since its $\ln \delta^{-1}$ (dominant) term does

**Algorithm 5** BSAR, algorithm for CPE-B in the FB setting

1: **Input:** budget $T$, $\mathcal{M}$, and $\mathtt{AR - Oracle}$.
2: $\tilde{\log}(n) \leftarrow \sum_{i=1}^{n} \frac{1}{i}$. $\tilde{T}_0 \leftarrow 0$. $A_1, R_1 \leftarrow \varnothing$.
3: **for** $t = 1, \dots, n$ **do**
4: $\quad \tilde{T}_t \leftarrow \left\lceil \frac{T-n}{\tilde{\log}(n)(n-t+1)} \right\rceil$
5: $\quad U_t \leftarrow [n] \setminus (A_t \cup R_t)$
6: $\quad$ Play each $e \in U_t$ for $\tilde{T}_t - \tilde{T}_{t-1}$ times
7: $\quad$ Update empirical mean $\hat{w}_t(e), \forall e \in U_t$
8: $\quad \hat{w}_t(e) \leftarrow \infty$ for all $e \in A_t$
9: $\quad M_t \leftarrow \mathtt{AR\text{-}Oracle}(\bot, R_t, \hat{\boldsymbol{w}}_t)$
10: $\quad$ **for** each $e \in U_t$ **do**
11: $\quad\quad$ **if** $e \in M_t$ **then**
12: $\quad\quad\quad \tilde{M}_{t,e} \leftarrow \mathtt{AR\text{-}Oracle}(\bot, R_t \cup \{e\}, \hat{\boldsymbol{w}}_t)$
13: $\quad\quad$ **else**
14: $\quad\quad\quad \tilde{M}_{t,e} \leftarrow \mathtt{AR\text{-}Oracle}(e, R_t, \hat{\boldsymbol{w}}_t)$
15: $\quad\quad$ **end if**
16: $\quad\quad$ // $\mathtt{AR\text{-}Oracle}$ returns $\bot$ if the calculated feasible set is empty
17: $\quad$ **end for**
18: $\quad p_t \leftarrow \underset{e \in U_t}{\mathrm{argmax}}\ \mathtt{MinW}(M_t, \hat{\boldsymbol{w}}_t) - \mathtt{MinW}(\tilde{M}_{t,e}, \hat{\boldsymbol{w}}_t)$
19: $\quad$ // $\mathtt{MinW}(\bot, \hat{\boldsymbol{w}}_t) = -\infty$
20: $\quad$ **if** $p_t \in M_t$ **then**
21: $\quad\quad A_{t+1} \leftarrow A_t \cup \{p_t\}, R_{t+1} \leftarrow R_t$
22: $\quad$ **else**
23: $\quad\quad A_{t+1} \leftarrow A_t, R_{t+1} \leftarrow R_t \cup \{p_t\}$
24: $\quad$ **end if**
25: **end for**
26: **return** $A_{n+1}$

not depend on unnecessary base arms $\tilde{N}$ either. In addition, if we impose some constraint on the constructed instances, the sample complexity of BLUCB (Theorem 1) can also match the lower bound up to a logarithmic factor (see the supplementary material for details). The condition $\delta < 0.1$ comes from the lower bound analysis, which ensures that the binary entropy of finding a correct or wrong answer can be lower bounded by $\ln \delta^{-1}$. Existing pure exploration works [11, 10] also have such condition on $\delta$ in their lower bounds.

Notice that, both our lower and upper bounds depend on the tight base-arm-level (instead of super-arm-level) gaps, and capture the *bottleneck insight*: different base arms in one super arm play distinct roles in determining its (sub-)optimality and impose different influences on the problem hardness.

## 5 Algorithm for the Fixed-Budget Setting

For CPE-B in the FB setting, we design a novel algorithm BSAR that adopts a special acceptance scheme for bottleneck identification. We allow BSAR to access an efficient accept-reject oracle AR-Oracle, which takes an accepted base arm $e$ or $\bot$, a rejected base arm set $R$ and a weight vector $\boldsymbol{v}$ as inputs, and returns an optimal super arm from the decision class $\mathcal{M}(e, R) = \{M \in \mathcal{M} : e \in M, R \cap M = \varnothing\}$ with respect to $\boldsymbol{v}$, i.e., $\mathtt{AR\text{-}Oracle} \in \mathrm{argmax}_{M \in \mathcal{M}(e,R)} \mathtt{MinW}(M, \boldsymbol{w})$. If $\mathcal{M}(e, R)$ is empty, AR-Oracle simply returns $\bot$. Such an efficient oracle exists for many decision classes, e.g., paths, matchings and spanning trees (see the supplementary material for implementation details).

BSAR allocates the sample budget $T$ to $n$ phases adaptively, and maintains the accepted set $A_t$, rejected set $R_t$ and undetermined set $U_t$. In each phase, we only sample base arms in $U_t$ and set the empirical rewards of base arms in $A_t$ to infinity (Line 8). Then, we call AR-Oracle to compute the empirical best super arm $M_t$. For each $e \in U_t$, we forbid $R_t$ and constrain $e$ inside/outside the calculated super arms and find the empirical best super arm $\tilde{M}_{t,e}$ from the restricted decision class (Lines 12,14). Then, we accept or reject the base arm $p_t$ that maximizes the empirical reward gap between $M_t$ and $\tilde{M}_{t,e}$, i.e., the one that is most likely to be in or out of $M_*$ (Line 18).

**Special acceptance scheme for bottleneck and polynomial running time.** The acceptance scheme $\hat{w}_t(e) \leftarrow \infty$ for all $e \in A_t$ (Line 8) is critical to the correctness and computation efficiency of BSAR. Since $A_t$ and $R_t$ are not pulled in phase $t$ and their estimated rewards are not accurate enough, we need to avoid them to disturb the following calculation of empirical bottleneck values (Lines 9-18). By setting the empirical rewards of $A_t$ to infinity, the estimation of bottleneck values for sub-optimal super arms $M_{\text{sub}}$ avoids the disturbance of $A_t$, because each $M_{\text{sub}}$ has at least one base arm with reward lower than OPT and this base arm will never be included in $A_t$ (conditioned on high probability events). As for $M_*$, its empirical bottleneck value can be raised, but this only enlarges the empirical gap between $M_*$ and $M_{\text{sub}}$ and does not affect the correctness of the choice $p_t$ (Line 18). Hence, this acceptance scheme guarantees the correctness of BSAR in bottleneck identification task.

Compared to existing CPE-L algorithm CSAR [11], they force the whole set $A_t$ inside the calculated super arms in the oracle, i.e., replacing Lines 12,14 with $\mathtt{AR\text{-}Oracle}(A_t, R_t \cup \{e\}, \hat{\boldsymbol{w}}_t)$ and $\mathtt{AR\text{-}Oracle}(A_t \cup \{e\}, R_t, \hat{\boldsymbol{w}}_t)$, and deleting Line 8. Such acceptance strategy incurs *exponential-time*

complexity on $s$-$t$ path instances,[4] and *only works* for the linear reward function, where the common part $A_t$ between two compared super arms can be canceled out. If one naively applies their acceptance strategy to our bottleneck problem, the common part $A_t$ is possible to drag down (dominate) the empirical bottleneck values of all calculated super arms (Lines 9,12,14) and their empirical gaps will become all zeros (Line 18), which destroys the correctness of the choice $p_t$ in theoretical analysis.

BSAR is the first to run in *polynomial time* on fixed-budget $s$-$t$ path instances among existing CPE algorithms, owing to its skillful acceptance scheme and the simplified AR-Oracle (only work with one accepted base arm instead of $A_t$). Specifically, for $E$ edges and $V$ vertices, the time complexity of AR-Oracle is $O(E(E+V))$ and BSAR only spends $O(E^2(E+V))$ time in decision making.

Now we give the definitions of fixed-budget reward gap and problem hardness, and then formally state the error probability result of BSAR. For $e \in M_*$, $\Delta_e^{\mathrm{B}} = \mathrm{OPT} - \max_{M \in \mathcal{M}: e \notin M} \mathrm{MinW}(M, \boldsymbol{w})$, and for $e \notin M_*$, $\Delta_e^{\mathrm{B}} = \mathrm{OPT} - \max_{M \in \mathcal{M}: e \in M} \mathrm{MinW}(M, \boldsymbol{w})$. Let $\Delta_{(1)}^{\mathrm{B}}, \ldots, \Delta_{(n)}^{\mathrm{B}}$ be the permutation of $\Delta_1^{\mathrm{B}}, \ldots, \Delta_n^{\mathrm{B}}$ such that $\Delta_{(1)}^{\mathrm{B}} \leq \cdots \leq \Delta_{(n)}^{\mathrm{B}}$, and the fixed-budget problem hardness is defined as $H^{\mathrm{B}} = \max_{i \in [n]} \frac{i}{(\Delta_{(i)}^{\mathrm{B}})^2}$. Let $\widetilde{\log}(n) = \sum_{i=1}^{n} \frac{1}{i}$.

**Theorem 4** (Fixed-budget Upper Bound). *For any $T > n$, algorithm BSAR (Algorithm 5) for CPE-B in the FB setting uses at most $T$ samples and returns the optimal super arm with the error probability bounded by*

$$O\left(n^2 \exp\left(-\frac{T-n}{\widetilde{\log}(n) R^2 H^{\mathrm{B}}}\right)\right).$$

Compared to the uniform sampling algorithm, which plays all base arms equally and has $O(n \exp(-\frac{T}{R^2 n \Delta_{\min}^{-2}}))$ error probability with $\Delta_{\min} = \mathrm{OPT} - \max_{M \neq M_*} \mathrm{MinW}(M, \boldsymbol{w})$, Theorem 4 achieves a significantly better correctness guarantee (when $\Delta_e^B > \Delta_{\min}$ for most $e \in [n]$). In addition, when our CPE-B problem reduces to conventional $K$-armed pure exploration problem [7], Theorem 4 matches existing state-of-the-art result in [7]. To our best knowledge, the lower bound for the fixed-budget setting in the CPE literature [11, 20, 23, 16] remains open.

Our error probability analysis falls on taking advantage of the bottleneck property to handle the disturbance from the accepted arm set (which are not pulled sufficiently) and guaranteeing the estimation accuracy of bottleneck rewards. The differences between our analysis and prior analysis for CSAR [11] are highlighted as follows: (i) Prior analysis [11] relies on the linear property to cancel out the common part between two super arms when calculating their reward gap, in order to avoid the disturbance of accepted arms. In contrast, to achieve this goal, we utilize the special acceptance scheme of BSAR to exclude all accepted arms in the calculation of bottleneck rewards, which effectively addresses the perturbation of inaccurate estimation on accepted arms. (ii) Prior analysis [11] mainly uses the "exchange sets" technique, which only works for the linear reward function and leads to the dependence on the parameter of decision class structures. Instead, our analysis exploits the bottleneck property to establish confidence intervals in the base arm level, and effectively avoids the dependence on the parameter of decision class structures.

## 6   Experiments

In this section, we conduct experiments for CPE-B in FC/FB settings on synthetic and real-world datasets. The synthetic dataset consists of the $s$-$t$ path and matching instances. For the $s$-$t$ path instance, the number of edges (base arms) $n = 85$, and the expected reward of edges $w(e) = [0, 10.5]$ ($e \in [n]$). The minimum reward gap of any two edges (which is also the minimum gap of bottleneck values between two super arms) is denoted by $\Delta_{\min} \in [0.4, 0.7]$. For the matching instances, we use a $5 \times 3$ complete bipartite graph, where $n = 15$, $w(e) = [0.1, 1.08]$ and $\Delta_{\min} \in [0.03, 0.07]$. We change $\Delta_{\min}$ to generate a series of instances with different hardness (plotted points in Figures 2(a),2(b),2(e)). In terms of the real-world dataset, we use the data of American airports and the number of available seats of flights in 2002, provided by the International Air Transportation Association database (www.iata.org) [6]. Here we regard an airport as a vertex and a direct flight connecting two airports as an edge (base arm), and also consider the number of available seats of a flight as the expected reward of an edge. Our objective is to find an air route connecting the starting and destination airports which maximizes the minimum number of available seats among its passing

---

[4]Finding a $s$-$t$ path which contains a given edge set is NP-hard. See the supplementary material for its proof.

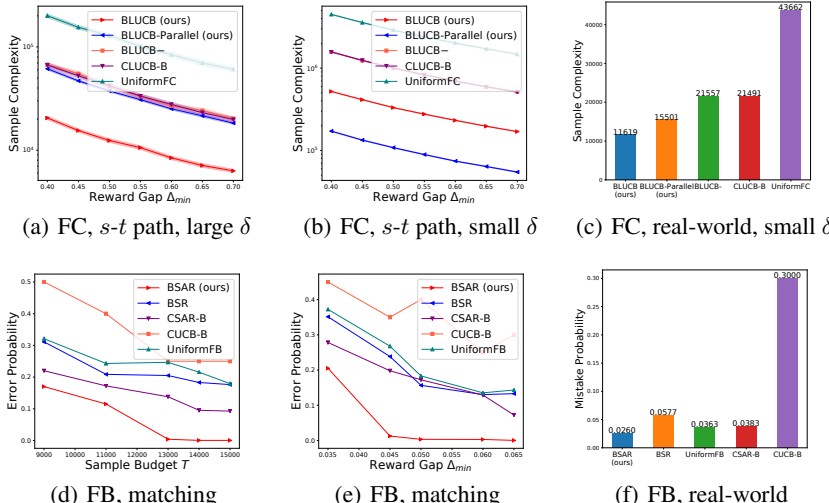

| (a) FC, $s$-$t$ path, large $\delta$ | (b) FC, $s$-$t$ path, small $\delta$ | (c) FC, real-world, small $\delta$ |
|---|---|---|
| (d) FB, matching | (e) FB, matching | (f) FB, real-world |

Figure 2: Experiments for CPE-B in the FC/FB setting on synthetic and real-world datasets.

flights. In this instance, $n = 9$ and $w(e) \in [0.62, 1.84]$. We present the detailed graphs with specific values of $w(e)$ for the $s$-$t$ path, matching and real-world air route instances in the supplementary material.

In the FC setting, we set a large $\delta = 0.005$ and a small $\delta = \exp(-1000)$, and perform $50$ independent runs to plot average sample complexity with $95\%$ confidence intervals. In the FB setting, we set sample budget $T \in [6000, 15000]$, and perform $3000$ independent runs to show the error probability across runs. For all experiments, the random reward of each edge $e \in [n]$ is i.i.d. drawn from Gaussian distribution $\mathcal{N}(w(e), 1)$.

**Experiments for the FC setting.** We compare our BLUCB/BLUCB-Parallel with three baselines. BLUCB− is an ablation variant of BLUCB, which replaces the sample strategy (Lines 12-14) with the one that uniformly samples a base arm in critical super arms. CLUCB-B [11] is the state-of-the-art fixed-confidence CPE-L algorithm run with bottleneck reward function. UniformFC is a fixed-confidence uniform sampling algorithm. As shown in Figures 2(a)-2(c), BLUCB and BLUCB-Parallel achieve better performance than the three baselines, which validates the statistical efficiency of our bottleneck-adaptive sample strategy. Under small $\delta$, BLUCB-Parallel enjoys lower sample complexity than BLUCB due to its careful algorithmic design to avoid playing unnecessary base arms, which matches our theoretical results.

**Experiments for the FB setting.** Our BSAR is compared with four baselines. As an ablation variant of BSAR, BSR removes the special acceptance scheme of BSAR. CSAR-B [11] is the state-of-the-art fixed-budget CPE-L algorithm implemented with bottleneck reward function. CUCB-B [14] is a regret minimization algorithm allowing nonlinear reward functions, and in pure exploration experiments we let it return the empirical best super arm after $T$ (sample budget) timesteps. UniformFB is a fixed-budget uniform sampling algorithm. One sees from Figures 2(d)-2(f) that, BSAR achieves significantly better error probability than all the baselines, which demonstrates that its special acceptance scheme effectively guarantees the correctness for the bottleneck identification task.

## 7 Conclusion and Future Work

In this paper, we study the Combinatorial Pure Exploration with the Bottleneck reward function (CPE-B) problem in FC/FB settings. For the FC setting, we propose two novel algorithms, which achieve the optimal sample complexity for a broad family of instances (within a logarithmic factor), and establish a matching lower bound to demonstrate their optimality. For the FB setting, we propose an algorithm whose error probability matches the state-of-the-art result, and it is the first to run efficiently on fixed-budget path instances among existing CPE algorithms. The empirical evaluation also validates the superior performance of our algorithms. There are several interesting directions worth further research. One direction is to derive a lower bound for the FB setting, and another direction is to investigate the general nonlinear reward functions.

## Acknowledgments and Disclosure of Funding

The work of Yihan Du is supported in part by the Technology and Innovation Major Project of the Ministry of Science and Technology of China under Grant 2020AAA0108400 and 2020AAA0108403. Yuko Kuroki is supported by Microsoft Research Asia and JST ACT-X 1124477.

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
