10:     **if** $\text{MinW}(\hat{M}_*, \underline{\boldsymbol{w}}_t) \geq \text{MinW}(\tilde{M}_t, \bar{\boldsymbol{w}}_t)$ **then**
11:         **return** $\hat{M}_*$
12:     **end if**
13:     $c_t \leftarrow \text{argmin}_{e \in \hat{M}_*} \underline{w}_t(e)$
14:     $F_t \leftarrow \{e \in \hat{B}_{\text{sub}} : \bar{w}_t(e) > \underline{w}_t(c_t)\}$
15:     $p_t \leftarrow \text{argmax}_{e \in F_t \cup \{c_t\}} \text{rad}_t(e)$
16:     Play $p_t$, and observe the reward
17:     Update empirical means $\hat{w}_{t+1}(p_t)$
18:     Update the number of samples $T_{t+1}(p_t)$
19: **end for**

---

**Definition 1** (Fixed-confidence Gap).

$$\Delta_e^{\text{C}} = \begin{cases} w(e) - \max_{M \neq M_*} \text{MinW}(M, \boldsymbol{w}), & \text{if } e \in M_*, & \text{(a)} \\ w(e) - \max_{M \in \mathcal{M}: e \in M} \text{MinW}(M, \boldsymbol{w}), & \text{if } e \in \tilde{N}, & \text{(b)} \\ \text{OPT} - \max_{M \in \mathcal{M}: e \in M} \text{MinW}(M, \boldsymbol{w}), & \text{if } e \in N. \end{cases}$$

Now we present the sample complexity upper bound of BLUCB.

**Theorem 1** (Fixed-confidence Upper Bound). *With probability at least $1 - \delta$, algorithm* BLUCB *(Algorithm 1) for CPE-B in the FC setting returns the optimal super arm with sample complexity*

$$O\left(\sum_{e \in [n]} \frac{R^2}{(\Delta_e^{\text{C}})^2} \ln\left(\sum_{e \in [n]} \frac{R^2 n}{(\Delta_e^{\text{C}})^2 \delta}\right)\right).$$

**Base-arm-gap dependent sample complexity.** Owing to the bottleneck-adaptive sample strategy, the reward gap $\Delta_e^{\text{C}}$ (Definition 1(a)(b)) is just defined as the difference between some critical bottleneck value and $w(e)$ itself, instead of the bottleneck gap between two super arms, and thus our result depends on the tight base-arm-level (instead of super-arm-level) gaps. For example, in Figure 1, BLUCB only spends $\tilde{O}((\frac{2}{\Delta_{e_2, e_1}^2} + \sum_{i=3,4,5,6} \frac{1}{\Delta_{e_i, e_1}^2}) \ln \delta^{-1})$ samples, while a naive adaptation of prior CPE algorithms [11, 12, 16] with the bottleneck reward function will cause a loose super-arm-gap dependent result $\tilde{O}(\sum_{e_i, i \in [6]} \frac{1}{\Delta_{M_*, M_{\text{sub}}}^2} \ln \delta^{-1})$. Regarding the optimality, Theorem 1 matches the lower bound (presented in Section 4) for some family of instances (up to a logarithmic factor). However, in general cases there still exists a gap on those needless base arms $\tilde{N}$ ($e_3, e_5, e_6$ in Figure 1), which are not contained in the lower bound. Next, we show how to bridge this gap.

### 3.2 Remove Dependence on Unnecessary Base Arms under Small $\delta$

**Challenges of avoiding unnecessary base arms.** Under the bottleneck reward function, in each sub-optimal super arm $M_{\text{sub}}$, only the base arms with rewards lower than OPT (base arms in $N$) can determine the relationship of bottleneck values between $M_*$ and $M_{\text{sub}}$ (the bottleneck of $M_{\text{sub}}$ is the most efficient choice to do this), and the others (base arms in $\tilde{N}$) are useless for revealing the sub-optimality of $M_{\text{sub}}$. Hence, to determine $M_*$, all we need is to sample the base arms in $M_*$ and the *bottlenecks from all sub-optimal super arms*, denoted by $B_{\text{sub}}$, to see that each sub-optimal super

---

**Algorithm 4** BLUCB-Explore, sub-algorithm of BLUCB-Verify, the *key algorithm*

---

1: **Input:** $\mathcal{M}$, $\kappa = 0.01$ and MaxOracle.
2: Initialize: play each $e \in [n]$ once, and update empirical means $\hat{\boldsymbol{w}}_{n+1}$ and $T_{n+1}$
3: **for** $t = n+1, n+2, \dots$ **do**
4:    $\mathrm{rad}_t(e) \leftarrow R\sqrt{2\ln(\frac{4nt^3}{\kappa})/T_t(e)}$, $\forall e \in [n]$
5:    $\underline{w}_t(e) \leftarrow \hat{w}_t(e) - \mathrm{rad}_t(e)$, $\forall e \in [n]$
6:    $\bar{w}_t(e) \leftarrow \hat{w}_t(e) + \mathrm{rad}_t(e)$, $\forall e \in [n]$
7:    $M_t \leftarrow$ MaxOracle$(\mathcal{M}, \underline{\boldsymbol{w}}_t)$
8:    $\hat{B}_{\mathrm{sub},t} \leftarrow$ BottleneckSearch$(\mathcal{M}, M_t, \underline{\boldsymbol{w}}_t)$
9:    **if** $\bar{w}_t(e) \leq \frac{1}{2}(\mathrm{MinW}(M_t, \underline{\boldsymbol{w}}_t) + \underline{w}_t(e))$ for all $e \in \hat{B}_{\mathrm{sub},t}$ **then**

10:       **return** $M_t, \hat{B}_{\mathrm{sub},t}$
11:    **end if**
12:    $c_t \leftarrow \mathrm{argmin}_{e \in M_t} \underline{w}_t(e)$
13:    $\hat{B}'_{\mathrm{sub},t} \leftarrow \{e \in \hat{B}_{\mathrm{sub},t} : \bar{w}_t(e) > \frac{1}{2}(\mathrm{

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

# Appendix

## A More Details of Experimental Setups

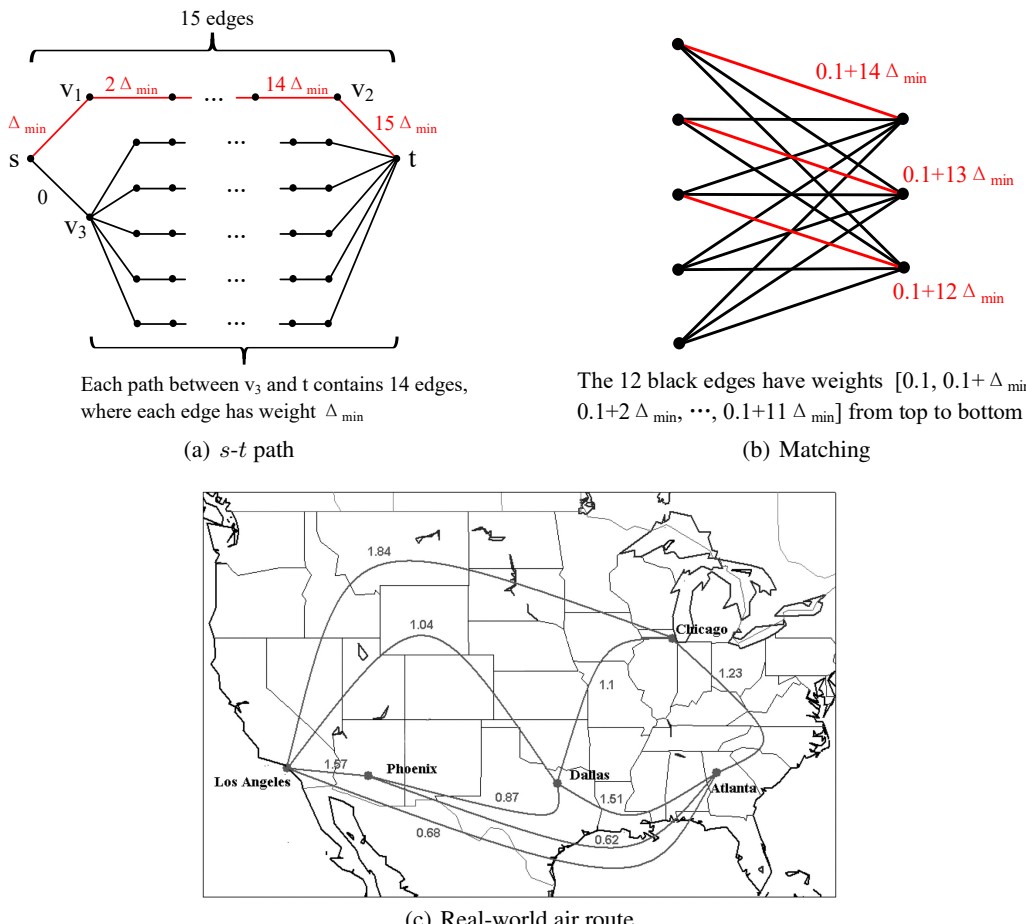

(a) $s$-$t$ path

(b) Matching

(c) Real-world air route

Figure 3: Graphs of the $s$-$t$ path, matching and real-world air route instances in our experiments.

In this section, we supplement more details of graphs and the expected rewards of edges (base arms) for the $s$-$t$ path, matching and real-world air route instances in our experiments.

Figure 3(a) shows the graph of $s$-$t$ path instance. The red path contains 15 edges with weights $[\Delta_{\min}, 2\Delta_{\min}, \ldots, 15\Delta_{\min}]$ and is the optimal $s$-$t$ path with the maximum bottleneck value. There are 5 paths connecting $v_3$ and $t$, and each of them contains 14 edges with weights $\Delta_{\min}$. In this instance, we set $\Delta_{\min} \in [0.4, 0.7]$.

As shown in Figure 3(b), the matching instance uses a $5 \times 3$ complete bipartite graph with $n = 15$ edges. The red matching is the optimal one, which contains three edges with weights $[0.1 + 14\Delta_{\min}, 0.1 + 13\Delta_{\min}, 0.1 + 12\Delta_{\min}]$. The remaining 12 black edges have weights $[0.1, 0.1 + \Delta_{\min}, \ldots, 0.1 + 11\Delta_{\min}]$ from top to bottom. In this instance, $\Delta_{\min} \in [0.03, 0.07]$.

Figure 3(c) illustrates the graph of real-world air route instance, which is originated from [6]. We regard an airport (e.g., Los Angeles) as a vertex and a direct flight connecting two airports (e.g., Los Angeles $\leftrightarrow$ Chicago) as an edge. The number marked on each edge denotes the number of available seats of this flight, i.e., the expected reward of this edge. Our objective is to find an air route connecting Los Angeles and Atlanta, and the optimal route is [Los Angeles $\leftrightarrow$ Chicago $\leftrightarrow$ Atlanta].

## B  CPE-B in the Fixed-Confidence Setting

### B.1  Proof for Algorithm BLUCB

In this subsection, we prove the sample complexity of Algorithm BLUCB (Theorem 1).

In order to prove Theorem 1, we first introduce the following Lemmas 1-5. For ease of notation, we define a function $\texttt{MinE}(M, \boldsymbol{v})$ to return the base arm with the minimum reward in $M$ with respect to weight vector $\boldsymbol{v}$, i.e., $\texttt{MinE}(M, \boldsymbol{v}) \in \arg\min_{e \in M} v(e)$.

**Lemma 1** (Concentration). *For any $t > 0$ and $e \in [n]$, defining the confidence radius $\mathrm{rad}_t(e) = R\sqrt{\frac{2\ln(\frac{4nt^3}{\delta})}{T_t(e)}}$ and the events*

$$\xi_t = \{\forall e \in [n], \ |w(e) - \hat{w}_t(e)| < \mathrm{rad}_t(e)\}$$

*and*

$$\xi = \bigcap_{t=1}^{\infty} \xi_t,$$

*then, we have*

$$\Pr[\xi] \geq 1 - \delta.$$

*Proof.* Since for any $e \in [n]$, the reward distribution of base arm $e$ has an R-sub-Gaussian tail and the mean of $w(e)$, according to the Hoeffding's inequality, we have that for any $t > 0$ and $e \in [n]$,

$$\Pr\left[|w(e) - \hat{w}_t(e)| \geq R\sqrt{\frac{2\ln(\frac{4nt^3}{\delta})}{T_t(e)}}\right] = \sum_{s=1}^{t-1} \Pr\left[|w(e) - \hat{w}_t(e)| \geq R\sqrt{\frac{2\ln(\frac{4nt^3}{\delta})}{T_t(e)}}, \ T_t(e) = s\right]$$

$$\leq \sum_{s=1}^{t-1} \frac{\delta}{2nt^3}$$

$$\leq \frac{\delta}{2nt^2}$$

Using a union bound over $e \in [n]$, we have

$$\Pr[\xi_t] \leq \frac{\delta}{2t^2}$$

and thus

$$\Pr[\xi] \geq 1 - \sum_{t=1}^{\infty} \Pr[\neg\xi_t]$$

$$\geq 1 - \sum_{t=1}^{\infty} \frac{\delta}{2t^2}$$

$$\geq 1 - \delta$$

$\square$

**Lemma 2.** *Assume that event $\xi$ occurs. Then, if algorithm BLUCB (Algorithm 1) terminates at round $t$, we have $M_t = M_*$.*

*Proof.* According to the stop condition (Line 9 of Algorithm 1), when algorithm BLUCB terminates at round $t$, we have that for any $M \in \mathcal{M} \setminus \mathcal{S}(M_t)$,

$$\texttt{MinW}(M_t, \boldsymbol{w}) \geq \texttt{MinW}(M_t, \underline{\boldsymbol{w}}_t) \geq \texttt{MinW}(M, \bar{\boldsymbol{w}}_t) \geq \texttt{MinW}(M, \boldsymbol{w}).$$

For any $M \in \mathcal{S}(M_t)$, according to the property of the bottleneck reward function, we have

$$\texttt{MinW}(M_t, \boldsymbol{w}) \geq \texttt{MinW}(M, \boldsymbol{w}).$$

Thus, we have $\texttt{MinW}(M_t, \boldsymbol{w}) \geq \texttt{MinW}(M, \boldsymbol{w})$ for any $M \neq M_t$ and according to the unique assumption of $M_*$, we obtain $M_t = M_*$. $\square$

**Lemma 3.** *Assume that event $\xi$ occurs. For any $e \in M_*$, if $\mathrm{rad}_t(e) < \frac{\Delta_e^{\mathrm{C}}}{4} = \frac{1}{4}(w(e) - \max_{M \neq M_*} \mathrm{MinW}(M, \boldsymbol{w}))$, then, base arm $e$ will not be pulled at round $t$, i.e., $p_t \neq e$.*

*Proof.* Suppose that for some $e \in M_*$, $\mathrm{rad}_t(e) < \frac{\Delta_e^{\mathrm{C}}}{4} = \frac{1}{4}(w(e) - \max_{M \neq M_*} \mathrm{MinW}(M, \boldsymbol{w}))$ and $p_t = e$. According to the selection strategy of $p_t$, we have that $\mathrm{rad}_t(c_t) < \frac{\Delta_e^{\mathrm{C}}}{4}$ and $\mathrm{rad}_t(d_t) < \frac{\Delta_e^{\mathrm{C}}}{4}$.

Case (i): If $e$ is selected from $M_*$, then one of $M_t$ and $\tilde{M}_t$ is $M_*$ such that $e = \mathrm{MinE}(M_*, \underline{\boldsymbol{w}}_t)$, and the other is a sub-optimal super arm $M'$. Let $e' = \mathrm{MinE}(M', \underline{\boldsymbol{w}}_t)$. $\underline{w}(e') \leq \underline{w}(\mathrm{MinE}(M', \boldsymbol{w})) \leq w(\mathrm{MinE}(M', \boldsymbol{w})) = \mathrm{MinW}(M', \boldsymbol{w})$. $\{e, e'\} = \{c_t, d_t\}$. Then,

$$\underline{w}(e) - \bar{w}(e') \geq w(e) - \underline{w}(e') - 2\mathrm{rad}_t(e) - 2\mathrm{rad}_t(e')$$
$$> w(e) - \mathrm{MinW}(M', \boldsymbol{w}) - \Delta_e^{\mathrm{C}}$$
$$\geq 0.$$

Then, we have

$$\mathrm{MinW}(M_*, \underline{\boldsymbol{w}}_t) = \underline{w}(e) > \bar{w}(e') \geq \mathrm{MinW}(M', \bar{\boldsymbol{w}}_t),$$

and algorithm BLUCB must have stopped, which gives a contradiction.

Case (ii): If $e$ is selected from a sub-optimal super arm $M$, then one of $M_t$ and $\tilde{M}_t$ is $M$ such that $e = \mathrm{MinE}(M, \underline{\boldsymbol{w}}_t)$. Since $e \in M_*$, we have $w(e) \geq \mathrm{MinW}(M_*, \boldsymbol{w}) > \mathrm{MinW}(M, \boldsymbol{w})$ and thus $w(e) - \mathrm{MinW}(M, \boldsymbol{w}) = w(e) - w(\mathrm{MinE}(M, \boldsymbol{w})) > 0$. Then,

$$\underline{w}(e) - \underline{w}(\mathrm{MinE}(M, \boldsymbol{w})) \geq w(e) - 2\mathrm{rad}_t(e) - \mathrm{MinW}(M, \boldsymbol{w})$$
$$> w(e) - \mathrm{MinW}(M, \boldsymbol{w}) - \frac{\Delta_e^{\mathrm{C}}}{2}$$
$$> 0,$$

which contradicts $e = \mathrm{MinE}(M, \underline{\boldsymbol{w}}_t)$. $\qquad\square$

**Lemma 4.** *Assume that event $\xi$ occurs. For any $e \notin M_*, w(e) \geq \mathrm{MinW}(M_*, \boldsymbol{w})$, if $\mathrm{rad}_t(e) < \frac{\Delta_e^{\mathrm{C}}}{2} = \frac{1}{2}(w(e) - \max_{M \in \mathcal{M}: e \in M} \mathrm{MinW}(M, \boldsymbol{w}))$, then, base arm $e$ will not be pulled at round $t$, i.e., $p_t \neq e$.*

*Proof.* Suppose that for some $e \notin M_*, w(e) \geq \mathrm{MinW}(M_*, \boldsymbol{w})$, $\mathrm{rad}_t(e) < \frac{\Delta_e^{\mathrm{C}}}{2} = \frac{1}{2}(w(e) - \max_{M \in \mathcal{M}: e \in M} \mathrm{MinW}(M, \boldsymbol{w}))$ and $p_t = e$. According to the selection strategy of $p_t$, we have that $\mathrm{rad}_t(c_t) < \frac{\Delta_e^{\mathrm{C}}}{2}$ and $\mathrm{rad}_t(d_t) < \frac{\Delta_e^{\mathrm{C}}}{2}$.

Since $e \notin M_*$, $e$ is selected from a sub-optimal super arm $M$. One of $M_t$ and $\tilde{M}_t$ is $M$ such that $e = \mathrm{MinE}(M, \underline{\boldsymbol{w}}_t)$. Since $w(e) \geq \mathrm{MinW}(M_*, \boldsymbol{w}) > \mathrm{MinW}(M, \boldsymbol{w})$, we have $w(e) - \mathrm{MinW}(M, \boldsymbol{w}) = w(e) - w(\mathrm{MinE}(M, \boldsymbol{w})) > 0$. Then,

$$\underline{w}(e) - \underline{w}(\mathrm{MinE}(M, \boldsymbol{w})) \geq w(e) - 2\mathrm{rad}_t(e) - \mathrm{MinW}(M, \boldsymbol{w})$$
$$> w(e) - \mathrm{MinW}(M, \boldsymbol{w}) - \Delta_e^{\mathrm{C}}$$
$$\geq 0,$$

which contradicts $e = \mathrm{MinE}(M, \underline{\boldsymbol{w}}_t)$. $\qquad\square$

**Lemma 5.** *Assume that event $\xi$ occurs. For any $e \notin M_*, w(e) < \mathrm{MinW}(M_*, \boldsymbol{w})$, if $\mathrm{rad}_t(e) < \frac{\Delta_e^{\mathrm{C}}}{4} = \frac{1}{4}(\mathrm{MinW}(M_*, \boldsymbol{w}) - \max_{M \in \mathcal{M}: e \in M} \mathrm{MinW}(M, \boldsymbol{w}))$, then, base arm $e$ will not be pulled at round $t$, i.e., $p_t \neq e$.*

*Proof.* Suppose that for some $e \notin M_*, w(e) < \mathrm{MinW}(M_*, \boldsymbol{w})$, $\mathrm{rad}_t(e) < \frac{\Delta_e^{\mathrm{C}}}{4} = \frac{1}{4}(\mathrm{MinW}(M_*, \boldsymbol{w}) - \max_{M \in \mathcal{M}: e \in M} \mathrm{MinW}(M, \boldsymbol{w}))$ and $p_t = e$. According to the selection strategy of $p_t$, we have that $\mathrm{rad}_t(c_t) < \frac{\Delta_e^{\mathrm{C}}}{4}$ and $\mathrm{rad}_t(d_t) < \frac{\Delta_e^{\mathrm{C}}}{4}$.

Case (i): If one of $M_t$ and $\tilde{M}_t$ is $M_*$, then the other is a sub-optimal super arm $M$ such that $e = \mathrm{MinE}(M, \underline{\boldsymbol{w}}_t)$. Let $f = \mathrm{MinE}(M_*, \underline{\boldsymbol{w}}_t)$. $\{e, f\} = \{c_t, d_t\}$. Then, we have

$$\underline{w}(f) - \bar{w}(e) \geq w(f) - 2\mathrm{rad}_t(f) - \underline{w}(e) - 2\mathrm{rad}_t(e)$$

$$>w(f) - \underline{w}(e) - \Delta_e^{\mathsf{C}}$$
$$\geq \mathtt{MinW}(M_*, \boldsymbol{w}) - \underline{w}(\mathtt{MinE}(M, \boldsymbol{w})) - \Delta_e^{\mathsf{C}}$$
$$\geq \mathtt{MinW}(M_*, \boldsymbol{w}) - \mathtt{MinW}(M, \boldsymbol{w}) - \Delta_e^{\mathsf{C}}$$
$$\geq 0.$$

Thus,

$$\mathtt{MinW}(M_*, \boldsymbol{w}) \geq \underline{w}(\mathtt{MinE}(M_*, \boldsymbol{w})) \geq \underline{w}(f) > \bar{w}(e) \geq \mathtt{MinW}(M, \bar{\boldsymbol{w}}_t)$$

and algorithm BLUCB must have stopped, which gives a contradiction.

Case (ii): If neither $M_t$ nor $\tilde{M}_t$ is $M_*$ and $e = c_t$, i.e., $e = \mathtt{MinE}(M_t, \underline{\boldsymbol{w}}_t)$, we have

$$\bar{w}(d_t) \geq \mathtt{MinW}(\tilde{M}_t, \bar{\boldsymbol{w}}_t) \geq \mathtt{MinW}(M_*, \bar{\boldsymbol{w}}_t) \geq \mathtt{MinW}(M_*, \boldsymbol{w})$$

and

$$\underline{w}(d_t) = \mathtt{MinW}(\tilde{M}_t, \underline{\boldsymbol{w}}_t) \leq \mathtt{MinW}(M_t, \underline{\boldsymbol{w}}_t) \leq \mathtt{MinW}(M_t, \boldsymbol{w}),$$

and thus

$$2\mathrm{rad}_t(d_t) = \bar{w}(d_t) - \underline{w}(d_t)$$
$$\geq \mathtt{MinW}(M_*, \boldsymbol{w}) - \mathtt{MinW}(M_t, \boldsymbol{w}),$$

which contradicts $\mathrm{rad}_t(d_t) < \frac{\Delta_e^{\mathsf{C}}}{4} < \frac{\Delta_e^{\mathsf{C}}}{2} \leq \frac{\mathtt{MinW}(M_*, \boldsymbol{w}) - \mathtt{MinW}(M_t, \boldsymbol{w})}{2}$.

Case (iii): If neither $M_t$ nor $\tilde{M}_t$ is $M_*$ and $e = d_t$, i.e., $e = \mathtt{MinE}(\tilde{M}_t, \underline{\boldsymbol{w}}_t)$. Let $c(M_*, \tilde{M}_t) = \frac{1}{2}(\mathtt{MinW}(M_*, \boldsymbol{w}) + \mathtt{MinW}(\tilde{M}_t, \boldsymbol{w}))$. If $c(M_*, \tilde{M}_t) < w(e) < \mathtt{MinW}(M_*, \boldsymbol{w})$, we have

$$\underline{w}(e) \geq w(e) - 2\mathrm{rad}_t(e)$$
$$> c(M_*, \tilde{M}_t) - \frac{\Delta_e^{\mathsf{C}}}{2}$$
$$\geq \frac{1}{2}(\mathtt{MinW}(M_*, \boldsymbol{w}) + \mathtt{MinW}(\tilde{M}_t, \boldsymbol{w})) - \frac{1}{2}(\mathtt{MinW}(M_*, \boldsymbol{w}) - \mathtt{MinW}(\tilde{M}_t, \boldsymbol{w}))$$
$$= \mathtt{MinW}(\tilde{M}_t, \boldsymbol{w})$$
$$\geq \underline{w}(\mathtt{MinE}(\tilde{M}_t, \boldsymbol{w})),$$

which contradicts $e = \mathtt{MinE}(\tilde{M}_t, \underline{\boldsymbol{w}}_t)$.

If $\mathtt{MinW}(\tilde{M}_t, \boldsymbol{w}) \leq w(e) \leq c(M_*, \tilde{M}_t)$, we have

$$\mathtt{MinW}(\tilde{M}_t, \bar{\boldsymbol{w}}_t) \leq \bar{w}(e)$$
$$\leq w(e) + 2\mathrm{rad}_t(e)$$
$$< c(M_*, \tilde{M}_t) + \frac{\Delta_e^{\mathsf{C}}}{2}$$
$$\leq \frac{1}{2}(\mathtt{MinW}(M_*, \boldsymbol{w}) + \mathtt{MinW}(\tilde{M}_t, \boldsymbol{w})) + \frac{1}{2}(\mathtt{MinW}(M_*, \boldsymbol{w}) - \mathtt{MinW}(\tilde{M}_t, \boldsymbol{w}))$$
$$= \mathtt{MinW}(M_*, \boldsymbol{w})$$
$$\leq \mathtt{MinW}(M_*, \bar{\boldsymbol{w}}_t).$$

In addition, from the uniqueness of $M_*$, we have $M_* \notin \mathcal{S}(\tilde{M}_t)$. Thus, the inequality $\mathtt{MinW}(\tilde{M}_t, \bar{\boldsymbol{w}}_t) < \mathtt{MinW}(M_*, \bar{\boldsymbol{w}}_t)$ violates the optimality of $\tilde{M}_t$ with respect to $\bar{\boldsymbol{w}}_t$. $\square$

Next, we prove Theorem 1.

*Proof.* For any $e \in [n]$, let $T(e)$ denote the number of samples for base arm $e$, and $t_e$ denote the last timestep at which $e$ is pulled. Then, we have $T_{t_e} = T(e) - 1$. Let $T$ denote the total number of samples. According to Lemmas 3-5, we have

$$R\sqrt{\frac{2\ln(\frac{4nt_e^3}{\delta})}{T(e) - 1}} \geq \frac{1}{4}\Delta_e^{\mathsf{C}}$$

Thus, we obtain

$$T(e) \leq \frac{32R^2}{(\Delta_e^{\mathsf{C}})^2} \ln\left(\frac{4nt_e^3}{\delta}\right) + 1 \leq \frac{32R^2}{(\Delta_e^{\mathsf{C}})^2} \ln\left(\frac{4nT^3}{\delta}\right) + 1$$

Summing over $e \in [n]$, we have

$$T \leq \sum_{e\in[n]} \frac{32R^2}{(\Delta_e^{\mathsf{C}})^2} \ln\left(\frac{4nT^3}{\delta}\right) + n \leq \sum_{e\in[n]} \frac{96R^2}{(\Delta_e^{\mathsf{C}})^2} \ln\left(\frac{2nT}{\delta}\right) + n,$$

where $\sum_{e\in[n]} \frac{R^2}{(\Delta_e^{\mathsf{C}})^2} \geq n$. Then, applying Lemma 20, we have

$$\begin{aligned}
T &\leq \sum_{e\in[n]} \frac{576R^2}{(\Delta_e^{\mathsf{C}})^2} \ln\left(\frac{2n^2}{\delta} \sum_{e\in[n]} \frac{96R^2}{(\Delta_e^{\mathsf{C}})^2}\right) + n \\
&= O\left(\sum_{e\in[n]} \frac{R^2}{(\Delta_e^{\mathsf{C}})^2} \ln\left(\sum_{e\in[n]} \frac{R^2n^2}{(\Delta_e^{\mathsf{C}})^2\delta}\right) + n\right) \\
&= O\left(\sum_{e\in[n]} \frac{R^2}{(\Delta_e^{\mathsf{C}})^2} \ln\left(\sum_{e\in[n]} \frac{R^2n}{(\Delta_e^{\mathsf{C}})^2\delta}\right)\right)
\end{aligned}$$

$\square$

## B.2 Details for the Improved Algorithm BLUCB-Parallel

In this subsection, we describe algorithm BLUCB-Parallel and its sub-algorithms in details, present the pseudo-code of the offline subroutine BottleneckSearch in Algorithm 6 and give the proofs of theoretical results.

### B.2.1 Detailed Algorithm Description

BLUCB-Parallel simulates multiple sub-algorithms BLUCB-Verify (Algorithm 3) with different confidence parameters in parallel. For $k \in \mathbb{N}$, BLUCB-Verify$_k$ denotes the sub-algorithm BLUCB-Verify with confidence parameter $\delta_k^V = \frac{\delta}{2^{k+1}}$ At each timestep $t$, we start or resume sub-algorithms BLUCB-Verify$_k$ such that $t$ is divisible by $2^k$ with only one sample, and then suspend these sub-algorithms. Such parallel simulation is performed until there exist some sub-algorithm which terminates and returns the answer $M_{\text{out}}$, and then we output $M_{\text{out}}$ as the answer of BLUCB-Parallel.

Algorithm BLUCB-Verify (Algorithm 3) calls Algorithm BLUCB-Explore (Algorithm 4) as its preparation procedure. Algorithm BLUCB-Explore uses a big constant confidence parameter $\kappa$ to guesses an optimal super arm and an advice set $\hat{B}_{\text{sub}}$ which contains the bottleneck base arms for the sub-optimal super arms, and then algorithm BLUCB-Verify verifies the correctness of the answer provided by BLUCB-Explore with the given confidence parameter $\delta^V$.

BLUCB-Explore first calculates an super arm $M_t$ with the maximum pessimistic bottleneck value, and then uses a subroutine BottleneckSearch (Algorithm 6) to find the set of bottleneck base arms $\hat{B}_{\text{sub},t}$ from all super arms in $\mathcal{M} \setminus \mathcal{S}(M_t)$ with respect to the lower reward confidence bound $\underline{\boldsymbol{w}}_t$. Then, we check whether for any base arm $e \in \hat{B}_{\text{sub},t}$, the optimistic reward of $e$ is lower than a half of its pessimistic reward plus the pessimistic bottleneck value of $M_t$. If this stopping condition holds, we simply return $M_t$ as the hypothesized optimal super arm and $\hat{B}_{\text{sub},t}$ as the advice set. Otherwise, we find the bottleneck $c_t$ from $M_t$ with respect to $\underline{\boldsymbol{w}}_t$, and collect the set of base arms $\hat{B}'_{\text{sub},t}$ which violate the stopping condition from $\hat{B}_{\text{sub},t}$. Let $P_t^E \stackrel{\text{def}}{=} \hat{B}'_{\text{sub},t} \cup \{c_t\}$ denote the sampling set of BLUCB-Explore. We plays the base arm with the maximum confidence radius in $P_t^E$.

BLUCB-Verify first calculates the super arm $\tilde{M}_t$ with the maximum optimistic bottleneck value in $\mathcal{M} \setminus \mathcal{S}(\hat{M}_*)$, and checks whether the pessimistic bottleneck value of $\hat{M}_*$ is higher than the optimistic bottleneck value of $\tilde{M}_t$. If so, we can determine that the answer $\hat{M}_*$ is correct, and simply stop

---

**Algorithm 6** `BottleneckSearch`, offline subroutine of `BLUCB-Explore`

1: **Input:** $\mathcal{M}$, $M_{\text{ex}}$, $\boldsymbol{v}$ and existence oracle `ExistOracle`$(\mathcal{F}, e)$: check if there exists a feasible super arm $M \in \mathcal{F}$ such that $e \in M$, and return $M$ if there exists and $\perp$ otherwise.

2: $A_{\text{out}} \leftarrow \varnothing$
3: **for** $e \in [n]$ **do**
4:     Remove all base arms with rewards lower than $v(e)$ from $\mathcal{M}$, and obtain $\mathcal{M}_{\geq v(e)}$
5:     **if** $e \in M_{\text{ex}}$ and $v(e) = \texttt{MinW}(M_{\text{ex}}, \boldsymbol{v})$ **then**
6:         **for** each $e_0 \in M_{\text{ex}} \setminus \{e\}$ **do**
7:             Remove $e_0$ from $\mathcal{M}_{\geq v(e)}$, and obtain $\mathcal{M}_{\geq v(e), -e_0}$
8:             **if** `ExistOracle`$(\mathcal{M}_{\geq v(e), -e_0}, e) \neq \perp$, then $A_{\text{out}} \leftarrow A_{\text{out}} \cup \{e\}$ and **break**
9:         **end for**
10:     **else if** `ExistOracle`$(\mathcal{M}_{\geq v(e)}, e) \neq \perp$ **then**
11:         $A_{\text{out}} \leftarrow A_{\text{out}} \cup \{e\}$
12:     **end if**
13: **end for**
14: **return** $A_{\text{out}}$

---

and return $\hat{M}_*$. Otherwise, we find the bottleneck $c_t$ from $M_t$ with respect to the lower reward confidence bound $\underline{\boldsymbol{w}}_t$, and collect the set of base arms $F_t$ whose optimistic rewards are higher than the pessimistic bottleneck value of $\hat{M}_*$ from $\hat{B}_{\text{sub}}$. Let $P_t^V \overset{\text{def}}{=} F_t \cup \{c_t\}$ denote the sampling set of `BLUCB-Verify`. We samples the base arm with the maximum confidence radius in $P_t^V$.

Now, we discuss the skillful subroutine `BottleneckSearch`, which is formally defined as follows.

**Definition 2** (`BottleneckSearch`). *We define* `BottleneckSearch`$(\mathcal{M}, M, \boldsymbol{v})$ *as an algorithm that takes decision class $\mathcal{M}$, super arm $M \in \mathcal{M}$ and weight vector $\boldsymbol{v} \in \mathbb{R}^d$ as inputs and returns a set of base arms $A_{\text{out}}$ that satisfies (i) for any $M' \in \mathcal{M} \setminus \mathcal{S}(M)$, there exists a base arm $e \in A_{\text{out}} \cap M'$ such that $\boldsymbol{v}(e) = \texttt{MinW}(M', \boldsymbol{v})$, and (ii) for any $e \in A_{\text{out}}$, there exists a super arm $M' \in \mathcal{M} \setminus \mathcal{S}(M)$ such that $e \in M'$ and $\boldsymbol{v}(e) = \texttt{MinW}(M', \boldsymbol{v})$.*

Algorithm 6 illustrates the implementation procedure of `BottleneckSearch`. We access the subroutine `BottleneckSearch` an efficient existence oracle `ExistOracle`$(\mathcal{F}, e)$, which returns a feasible super arm that contains base arm $e$ from decision class $\mathcal{F}$ if there exists, and otherwise returns $\perp$, i.e., `ExistOracle`$(\mathcal{F}, e) \in \{M \in \mathcal{F} : e \in M\}$. Such efficient oracles exist for a wide family of decision classes. For example, for $s$-$t$ paths, this problem can be reduced to the well-studied 2-vertex connectivity problem [20], which is polynomially tractable (see Section E.1 for the proof of reduction). For maximum cardinality matchings, we just need to remove $e$ and its two end vertices, and then find a feasible maximum cardinality matching in the remaining graph. For spanning trees, we can just merge the vertices of $e$ and find a feasible spanning tree in the remaining graph, which can also be solved efficiently.

In the subroutine `BottleneckSearch`, we enumerate all the base arms to collect the bottlenecks. For each enumerated base arm $e$, we first remove all base arms with the rewards lower than $w(e)$ from $\mathcal{M}$ and obtain a new decision class $\mathcal{M}_{\geq w(e)}$, in which the super arms only contain the base arms with the rewards at least $w(e)$. Then, we call `ExistOracle` to check whether there exists a feasible super arm $M_e$ that contains $e$ in $\mathcal{M}_{\geq w(e)}$. If there exist, then we obtain that $e$ is the bottleneck of $M_e$ with respect to $\boldsymbol{v}$, i.e., $e \in M_e$ and $w(e) = \texttt{MinW}(M_e, \boldsymbol{v})$, and add $e$ to the output set $A_{\text{out}}$. Otherwise, we can determine that $e$ is not the bottleneck for any super arm in $\mathcal{M} \setminus \mathcal{S}(M_t)$ with respect to $\boldsymbol{v}$. However, for the particular $e$ such that $e \in M_t$ and $w(e) = \texttt{MinW}(M_t, \boldsymbol{v})$, directly calling `ExistOracle` can obtain the output $M_t$, which should be excluded. We solve this problem by repeatedly removing each base arm in $M_t \setminus \{e\}$ and then calling `ExistOracle` on the new decision classes, which can check whether $e$ is some super arm's bottleneck apart from $\mathcal{M} \setminus \mathcal{S}(M_t)$.

### B.2.2   Proof for Algorithm `BLUCB-Parallel`

Below we give the theoretical results for the proposed algorithms.

For algorithm `BLUCB-Explore`, define the events $\xi_{0,t} = \{\forall e \in [n], |w(e) - \hat{w}_t(e)| < \text{rad}_t(e)\}$ and $\xi_0 = \bigcap_{t=1}^{\infty} \xi_{0,t}$. Then, similar to Lemma 1, we have $\Pr[\xi_0] \geq 1 - \kappa$. For algorithm `BLUCB-Verify`, define the events $\xi_t = \{\forall e \in [n], |w(e) - \hat{w}_t(e)| < \text{rad}_t(e)\}$ and $\xi = \bigcap_{t=1}^{\infty} \xi_t$. Then, applying

Lemma 1, we have $\Pr[\xi] \geq 1 - \delta^V$. Let $M_{\text{second}} = \text{argmax}_{M \in \mathcal{M} \setminus \{M_*\}} \text{MinW}(M, w)$ denote the second best super arm.

**Lemma 6** (Correctness of BLUCB-Explore). *For algorithm* BLUCB-Explore, *assume that event $\xi_0$ occurs. Then, if algorithm* BLUCB-Explore *(Algorithm 4) terminates at round $t$, we have that (i) $M_t = M_*$, (ii) for any $M \in \mathcal{M} \setminus \mathcal{S}(M_*)$, there exists a base arm $e \in \hat{B}_{\text{sub},t} \cap M$ satisfying $w(e) \leq \frac{1}{2}(\text{MinW}(M_*, \boldsymbol{w}) + \text{MinW}(M, \boldsymbol{w}))$, i.e., $\Delta^{\text{C}}_{\text{MinW}(M_*,\boldsymbol{w}),e} \geq \frac{1}{2}\Delta^{\text{C}}_{M_*,M}$ and (iii) for any $e \in \hat{B}_{\text{sub},t}$, there exists a sub-optimal super arm $M \in \mathcal{M} \setminus \mathcal{S}(M_*)$ such that $e \in M$ and $w(e) \leq \frac{1}{2}(\text{MinW}(M_*, \boldsymbol{w}) + \text{MinW}(M)) \leq \frac{1}{2}(\text{MinW}(M_*, \boldsymbol{w}) + \text{MinW}(M_{\text{second}}, \boldsymbol{w}))$.*

*Proof.* According to the stop condition (Lines 9 of Algorithm 4) and the definition of BottleneckSearch (Definition 2), when algorithm BLUCB-Explore (Algorithm 4) terminates at round $t$, we have that for any $\mathcal{M} \setminus \mathcal{S}(M_t)$, there exists a base arm $e \in \hat{B}_{\text{sub},t} \cap M$ satisfying

$$\bar{w}_t(e) \leq \frac{1}{2}(\text{MinW}(M_t, \underline{\boldsymbol{w}}_t) + \underline{w}_t(e)) = \frac{1}{2}(\text{MinW}(M_t, \underline{\boldsymbol{w}}_t) + \text{MinW}(M, \underline{\boldsymbol{w}}_t))$$

and thus,

$$w(e) \leq \frac{1}{2}(\text{MinW}(M_t, \boldsymbol{w}) + \text{MinW}(M, \boldsymbol{w})).$$

Then, we can obtain $M_t = M_*$. Otherwise, we cannot find any base arm $e \in M_*$ satisfying $w(e) \leq \frac{1}{2}(\text{MinW}(M_t, \boldsymbol{w}) + \text{MinW}(M_*, \boldsymbol{w})) < \text{MinW}(M_*, \boldsymbol{w})$, where $M_t$ is a sub-optimal super arm. Thus, we have (i) and (ii).

Now we prove (iii). According to the stop condition (Lines 9 of Algorithm 4), the definition of BottleneckSearch (Definition 2) and $M_t = M_*$, we have that for any $e \in \hat{B}_{\text{sub},t}$, there exists a sub-optimal super arm $\mathcal{M} \setminus \mathcal{S}(M_*)$ such that $e \in M$ and $\underline{w}_t(e) = \text{MinW}(M, \underline{\boldsymbol{w}}_t)$, and thus

$$\bar{w}_t(e) \leq \frac{1}{2}(\text{MinW}(M_*, \underline{\boldsymbol{w}}_t) + \underline{w}_t(e)) = \frac{1}{2}(\text{MinW}(M_*, \underline{\boldsymbol{w}}_t) + \text{MinW}(M, \underline{\boldsymbol{w}}_t)),$$

Then, for any $e \in \hat{B}_{\text{sub},t}$,

$$w(e) \leq \frac{1}{2}(\text{MinW}(M_*, \boldsymbol{w}) + \text{MinW}(M)) \leq \frac{1}{2}(\text{MinW}(M_*, \boldsymbol{w}) + \text{MinW}(M_{\text{second}}, \boldsymbol{w})),$$

which completes the proof. $\qquad\square$

**Lemma 7.** *For algorithm* BLUCB-Explore, *assume that event $\xi_0$ occurs. For any two base arms $e_1, e_2 \in [n]$ s.t. $w(e_1) < w(e_2)$, if $\text{rad}_t(e_1) < \frac{1}{6}\Delta^{\text{C}}_{e_2,e_1}$ and $\text{rad}_t(e_2) < \frac{1}{6}\Delta^{\text{C}}_{e_2,e_1}$, we have $\bar{w}_t(e_1) < \frac{1}{2}(\underline{w}_t(e_1) + \underline{w}_t(e_2))$.*

*Proof.* if $\text{rad}_t(e_1) < \frac{1}{6}\Delta^{\text{C}}_{e_2,e_1}$ and $\frac{1}{6}\Delta^{\text{C}}_{e_2,e_1}$, we have

$$\bar{w}_t(e_1) - \frac{1}{2}(\underline{w}_t(e_1) + \underline{w}_t(e_2)) = \underline{w}_t(e_1) + 2\text{rad}_t(e_1) - \frac{1}{2}(\underline{w}_t(e_1) + \underline{w}_t(e_2))$$

$$= \frac{1}{2}\underline{w}_t(e_1) - \frac{1}{2}\underline{w}_t(e_2) + 2\text{rad}_t(e_1)$$

$$\leq \frac{1}{2}w_t(e_1) - \frac{1}{2}(w_t(e_2) - 2\text{rad}_t(e_2)) + 2\text{rad}_t(e_1)$$

$$= \frac{1}{2}w_t(e_1) - \frac{1}{2}w_t(e_2) + \text{rad}_t(e_2) + 2\text{rad}_t(e_1)$$

$$< \frac{1}{2}w_t(e_1) - \frac{1}{2}w_t(e_2) + \frac{1}{2}\Delta^{\text{C}}_{e_2,e_1}$$

$$= 0.$$

$\qquad\square$

**Lemma 8.** *For algorithm* BLUCB-Explore, *assume that event $\xi_0$ occurs. For any $e \in M_*$, if $\text{rad}_t(e) < \frac{\Delta^{\text{C}}_e}{12} = \frac{1}{12}(w(e) - \max_{M \neq M_*} \text{MinW}(M, \boldsymbol{w}))$, then, base arm $e$ will not be pulled at round $t$, i.e., $p_t \neq e$.*

*Proof.* Suppose that for some $e \in M_*$, $\mathrm{rad}_t(e) < \frac{\Delta_e^{\mathsf{C}}}{12} = \frac{1}{12}(w(e) - \max_{M \neq M_*} \mathrm{MinW}(M, \boldsymbol{w}))$ and $p_t = e$.

According to the selection strategy of $p_t$, we have that for any $i \in P_t^E$, $\mathrm{rad}_t(i) \leq \mathrm{rad}_t(e) < \frac{\Delta_e^{\mathsf{C}}}{12}$. From the definition of $\mathtt{BottleneckSearch}$ and $c_t$, we have that for any $i \in P_t^E$, there exists a super arm $M \in \mathcal{M}$ such that $i \in M$ and $\underline{w}_t(i) = \mathrm{MinW}(M, \underline{\boldsymbol{w}}_t)$.

Case (i): Suppose that $e$ is selected from a sub-optimal super arm $M$, i.e., $e \in M$ and $\underline{w}_t(e) = \mathrm{MinW}(M, \underline{\boldsymbol{w}}_t)$. Then, using $\mathrm{rad}_t(e) < \frac{\Delta_e^{\mathsf{C}}}{12} < \frac{\Delta_e^{\mathsf{C}}}{2}$, we have

$$
\begin{aligned}
\underline{w}_t(e) \geq & w(e) - 2\mathrm{rad}_t(e) \\
> & w(e) - (w(e) - \mathrm{MinW}(M_{\mathrm{second}}, \boldsymbol{w})) \\
= & \mathrm{MinW}(M_{\mathrm{second}}, \boldsymbol{w}) \\
\geq & \mathrm{MinW}(M, \boldsymbol{w}) \\
\geq & \mathrm{MinW}(M, \underline{\boldsymbol{w}}_t)
\end{aligned}
$$

which gives a contradiction.

Case (ii): Suppose that $e$ is selected from $M_*$, i.e., $\underline{w}_t(e) = \mathrm{MinW}(M_*, \underline{\boldsymbol{w}}_t)$. We can obtain $M_* = M_t$. Otherwise,

$$
\begin{aligned}
\mathrm{MinW}(M_*, \underline{\boldsymbol{w}}_t) = & \underline{w}_t(e) \\
\geq & w(e) - 2\mathrm{rad}_t(e) \\
> & w(e) - (w(e) - \mathrm{MinW}(M_{\mathrm{second}}, \boldsymbol{w})) \\
= & \mathrm{MinW}(M_{\mathrm{second}}, \boldsymbol{w}) \\
\geq & \mathrm{MinW}(M_t, \boldsymbol{w}) \\
\geq & \mathrm{MinW}(M_t, \underline{\boldsymbol{w}}_t)
\end{aligned}
$$

Thus, We have $M_* = M_t$. From the definition of $\mathtt{BottleneckSearch}$, for any $e' \in \hat{B}'_{\mathrm{sub},t}$, there exists a super arm $M' \in \mathcal{M} \setminus \{M_*\}$ such that $e' \in M'$ and $\underline{w}_t(e') = \mathrm{MinW}(M', \underline{\boldsymbol{w}}_t)$. If $w(e') \geq \frac{1}{2}(w(e) + \mathrm{MinW}(M', \boldsymbol{w}))$, using $\mathrm{rad}_t(e') < \frac{\Delta_e^{\mathsf{C}}}{12} < \frac{\Delta_e^{\mathsf{C}}}{4}$, we have

$$
\begin{aligned}
\underline{w}_t(e') \geq & w(e') - 2\mathrm{rad}_t(e') \\
> & \frac{1}{2}(w(e) + \mathrm{MinW}(M', \boldsymbol{w})) - \frac{1}{2}(w(e) - M_{\mathrm{second}}) \\
\geq & \frac{1}{2}(w(e) + \mathrm{MinW}(M', \boldsymbol{w})) - \frac{1}{2}(w(e) - \mathrm{MinW}(M', \boldsymbol{w})) \\
= & \mathrm{MinW}(M', \boldsymbol{w}),
\end{aligned}
$$

which gives a contradiction.

If $w(e') < \frac{1}{2}(w(e) + \mathrm{MinW}(M', \boldsymbol{w}))$, we have

$$
\begin{aligned}
\Delta_{e,e'}^{\mathsf{C}} = & w(e) - w(e') \\
> & w(e) - \frac{1}{2}(w(e) + \mathrm{MinW}(M', \boldsymbol{w})) \\
= & \frac{1}{2}(w(e) - \mathrm{MinW}(M', \boldsymbol{w}))
\end{aligned}
$$

Since $\mathrm{rad}_t(e) < \frac{\Delta_e^{\mathsf{C}}}{12} = \frac{1}{12}(w(e) - \mathrm{MinW}(M_{\mathrm{second}})) \leq \frac{1}{12}(w(e) - \mathrm{MinW}(M', \boldsymbol{w})) \leq \frac{1}{6}\Delta_{e,e'}^{\mathsf{C}}$ and $\mathrm{rad}_t(e') \leq \mathrm{rad}_t(e) < \frac{1}{6}\Delta_{e,e'}^{\mathsf{C}}$, according to Lemma 7, we have

$$
\begin{aligned}
\bar{w}_t(e') < & \frac{1}{2}(\underline{w}_t(e) + \underline{w}_t(e')) \\
= & \frac{1}{2}(\mathrm{MinW}(M_t, \underline{\boldsymbol{w}}_t) + \underline{w}_t(e')),
\end{aligned}
$$

which contradicts the definition of $\hat{B}'_{\mathrm{sub},t}$. $\qquad \square$

**Lemma 9.** *For algorithm* BLUCB-Explore, *assume that event $\xi_0$ occurs. For any $e \notin M_*, w(e) \geq$* MinW$(M_*, \boldsymbol{w})$, *if* $\mathrm{rad}_t(e) < \frac{\Delta_e^C}{2} = \frac{1}{2}(w(e) - \max_{M \in \mathcal{M}:e \in M} \mathrm{MinW}(M, \boldsymbol{w}))$, *then, base arm $e$ will not be pulled at round $t$, i.e., $p_t \neq e$.*

*Proof.* Suppose that for some $e \notin M_*, w(e) \geq \mathrm{MinW}(M_*, \boldsymbol{w})$, $\mathrm{rad}_t(e) < \frac{\Delta_e^C}{2} = \frac{1}{2}(w(e) - \max_{M \in \mathcal{M}:e \in M} \mathrm{MinW}(M, \boldsymbol{w}))$ and $p_t = e$.

According to the selection strategy of $p_t$, we have that for any $i \in P_t^E$, $\mathrm{rad}_t(i) \leq \mathrm{rad}_t(e) < \frac{\Delta_e^C}{12}$. From the definition of BottleneckSearch and $c_t$, we have that for any $i \in P_t^E$, there exists a super arm $M \in \mathcal{M}$ such that $i \in M$ and $\underline{w}_t(i) = \mathrm{MinW}(M, \underline{\boldsymbol{w}}_t)$.

Since $e \notin M_*$, $e$ is selected from a sub-optimal super arm $M'$, i.e., $e \in M'$ and $\underline{w}_t(e) = \mathrm{MinW}(M', \underline{\boldsymbol{w}}_t)$. Then,

$$
\begin{aligned}
\underline{w}_t(e) \geq{}& w(e) - 2\mathrm{rad}_t(e) \\
>{}& w(e) - (w(e) - \max_{M \in \mathcal{M}:e \in M} \mathrm{MinW}(M, \boldsymbol{w})) \\
={}& \max_{M \in \mathcal{M}:e \in M} \mathrm{MinW}(M, \boldsymbol{w}) \\
\geq{}& \mathrm{MinW}(M', \boldsymbol{w}) \\
\geq{}& \mathrm{MinW}(M', \underline{\boldsymbol{w}}_t)
\end{aligned}
$$

which contradicts $\underline{w}_t(e) = \mathrm{MinW}(M', \underline{\boldsymbol{w}}_t)$. $\qquad\square$

**Lemma 10.** *For algorithm* BLUCB-Explore, *assume that event $\xi_0$ occurs. For any $e \notin M_*, w(e) <$* MinW$(M_*, \boldsymbol{w})$, *if* $\mathrm{rad}_t(e) < \frac{\Delta_e^C}{12} = \frac{1}{12}(\mathrm{MinW}(M_*, \boldsymbol{w}) - \max_{M \in \mathcal{M}:e \in M} \mathrm{MinW}(M, \boldsymbol{w}))$, *then, base arm $e$ will not be pulled at round $t$, i.e., $p_t \neq e$.*

*Proof.* Suppose that for some $e \notin M_*, w(e) < \mathrm{MinW}(M_*, \boldsymbol{w}), \mathrm{rad}_t(e) < \frac{\Delta_e^C}{12} = \frac{1}{12}(\mathrm{MinW}(M_*, \boldsymbol{w}) - \max_{M \in \mathcal{M}:e \in M} \mathrm{MinW}(M, \boldsymbol{w}))$ and $p_t = e$.

According to the selection strategy of $p_t$, we have that for any $i \in P_t^E$, $\mathrm{rad}_t(i) \leq \mathrm{rad}_t(e) < \frac{\Delta_e^C}{12}$. From the definition of BottleneckSearch and $c_t$, we have that for any $i \in P_t^E$, there exists a super arm $M \in \mathcal{M}$ such that $i \in M$ and $\underline{w}_t(i) = \mathrm{MinW}(M, \underline{\boldsymbol{w}}_t)$. Thus, there exists a sub-optimal super arm $M' \in \mathcal{M}$ such that $e \in M'$ and $\underline{w}_t(e) = \mathrm{MinW}(M', \underline{\boldsymbol{w}}_t)$.

Case (i) Suppose that $w(e) \geq \frac{1}{2}(\mathrm{MinW}(M_*, \boldsymbol{w}) + \mathrm{MinW}(M', \boldsymbol{w}))$. Using $\mathrm{rad}_t(e) < \frac{\Delta_e^C}{12} < \frac{\Delta_e^C}{4}$, we have

$$
\begin{aligned}
\underline{w}_t(e) \geq{}& w(e) - 2\mathrm{rad}_t(e) \\
>{}& \frac{1}{2}(\mathrm{MinW}(M_*, \boldsymbol{w}) + \mathrm{MinW}(M', \boldsymbol{w})) - \frac{1}{2}(\mathrm{MinW}(M_*, \boldsymbol{w}) - \max_{M \in \mathcal{M}:e \in M} \mathrm{MinW}(M, \boldsymbol{w})) \\
={}& \frac{1}{2}\mathrm{MinW}(M', \boldsymbol{w}) + \frac{1}{2} \max_{M \in \mathcal{M}:e \in M} \mathrm{MinW}(M, \boldsymbol{w}) \\
\geq{}& \mathrm{MinW}(M', \boldsymbol{w}),
\end{aligned}
$$

which contradicts $\underline{w}_t(e) = \mathrm{MinW}(M', \underline{\boldsymbol{w}}_t)$.

Case (ii) Suppose that $w(e) < \frac{1}{2}(\mathrm{MinW}(M_*, \boldsymbol{w}) + \mathrm{MinW}(M', \boldsymbol{w}))$. According to the definition of BottleneckSearch (Definition 2) and $c_t$, there exists a base arm $\tilde{e} \in \hat{B}_{\mathrm{sub},t} \cap \{c_t\}$ satisfying $\tilde{e} \in M_*$ and $\underline{w}_t(\tilde{e}) = \mathrm{MinW}(M_*, \underline{\boldsymbol{w}}_t)$.

First, we prove $\tilde{e} \in P_t^E$ and thus $\mathrm{rad}_t(\tilde{e}) \leq \mathrm{rad}_t(e) < \frac{\Delta_e^C}{12}$. If $M_* = M_t$, then $\tilde{e} = c_t$ and the claim holds. If $M_* \neq M_t$, then $\tilde{e} \in \hat{B}_{\mathrm{sub},t}$. We can obtain that $\tilde{e}$ will be put into $\hat{B}'_{\mathrm{sub},t} \subseteq P_t^E$. Otherwise, we have

$$
w(\tilde{e}) \leq \bar{w}_t(\tilde{e}) \leq \frac{1}{2}(\mathrm{MinW}(M_t, \underline{\boldsymbol{w}}_t) + \underline{w}_t(\tilde{e})) \leq \frac{1}{2}(\mathrm{MinW}(M_t, \boldsymbol{w}) + w(\tilde{e})).
$$

Since $w(\tilde{e}) \geq \mathrm{MinW}(M_*, \boldsymbol{w})$ and $\mathrm{MinW}(M_t, \boldsymbol{w}) < \mathrm{MinW}(M_*, \boldsymbol{w})$, the above inequality cannot hold. Thus, we obtain that $\tilde{e}$ will be put into $\hat{B}'_{\mathrm{sub},t} \subseteq P_t^E$ and thus $\mathrm{rad}_t(\tilde{e}) \leq \mathrm{rad}_t(e) < \frac{\Delta_e^C}{12}$.

Next, we discuss the following two cases: (a) $e = c_t$ and (b) $e \neq c_t$.

(a) if $e = c_t$, then $M_t \neq M_*$. Using $\text{rad}_t(\tilde{e}) \leq \text{rad}_t(e) < \frac{\Delta_e^{\text{C}}}{12} < \frac{\Delta_e^{\text{C}}}{2}$, we have

$$
\begin{aligned}
\text{MinW}(M_*, \underline{\boldsymbol{w}}_t) =& \underline{w}_t(\tilde{e}) \\
\geq & w(e) - 2\text{rad}_t(e) \\
> & w(e) - (\text{MinW}(M_*, \boldsymbol{w}) - \max_{M \in \mathcal{M}: e \in M} \text{MinW}(M, \boldsymbol{w})) \\
\geq & \text{MinW}(M_*, \boldsymbol{w}) - (\text{MinW}(M_*, \boldsymbol{w}) - \text{MinW}(M_t, \boldsymbol{w})) \\
\geq & \text{MinW}(M_t, \boldsymbol{w}) \\
\geq & \text{MinW}(M_t, \underline{\boldsymbol{w}}_t),
\end{aligned}
$$

which contradicts the definition of $M_t$.

(b) if $e \neq c_t$, i.e., $e \in \hat{B}'_{\text{sub},t}$ we have

$$
\begin{aligned}
\Delta_{\tilde{e},e}^{\text{C}} =& w(\tilde{e}) - w(e) \\
> & \text{MinW}(M_*, \boldsymbol{w}) - \frac{1}{2}(\text{MinW}(M_*, \boldsymbol{w}) + \text{MinW}(M', \boldsymbol{w})) \\
= & \frac{1}{2}(\text{MinW}(M_*, \boldsymbol{w}) - \text{MinW}(M', \boldsymbol{w}))
\end{aligned}
$$

Since $\text{rad}_t(e) < \frac{\Delta_e^{\text{C}}}{12} = \frac{1}{12}(\text{MinW}(M_*, \boldsymbol{w}) - \max_{M \in \mathcal{M}: e \in M} \text{MinW}(M, \boldsymbol{w})) \leq \frac{1}{12}(\text{MinW}(M_*, \boldsymbol{w}) - \text{MinW}(M', \boldsymbol{w})) < \frac{1}{6}\Delta_{\tilde{e},e}^{\text{C}}$ and $\text{rad}_t(\tilde{e}) \leq \text{rad}_t(e) < \frac{1}{6}\Delta_{\tilde{e},e}^{\text{C}}$, according to Lemma 7, we have

$$
\begin{aligned}
\bar{w}_t(e) < & \frac{1}{2}(\underline{w}_t(\tilde{e}) + \underline{w}_t(e)) \\
= & \frac{1}{2}(\text{MinW}(M_*, \underline{\boldsymbol{w}}_t) + \underline{w}_t(e)) \\
\leq & \frac{1}{2}(\text{MinW}(M_t, \underline{\boldsymbol{w}}_t) + \underline{w}_t(e'))
\end{aligned}
$$

which contradicts the definition of $\hat{B}'_{\text{sub},t}$. $\qquad\square$

**Theorem 5** (Sample Complexity of BLUCB-Explore). *With probability at least $1 - \kappa$, the BLUCB-Explore algorithm (Algorithm 4) will return $M_*$ with sample complexity*

$$
O\left(\sum_{e \in [n]} \frac{R^2}{(\Delta_e^{\text{C}})^2} \ln \left(\sum_{e \in [n]} \frac{R^2 n}{(\Delta_e^{\text{C}})^2 \kappa}\right)\right).
$$

*Proof.* For any $e \in [n]$, let $T(e)$ denote the number of samples for base arm $e$, and $t_e$ denote the last timestep at which $e$ is pulled. Then, we have $T_{t_e} = T(e) - 1$. Let $T$ denote the total number of samples. According to Lemmas 8-10, we have

$$
R\sqrt{\frac{2\ln(\frac{4nt_e^3}{\kappa})}{T(e) - 1}} \geq \frac{1}{12}\Delta_e^{\text{C}}
$$

Thus, we obtain

$$
T(e) \leq \frac{288R^2}{(\Delta_e^{\text{C}})^2} \ln \left(\frac{4nt_e^3}{\kappa}\right) + 1 \leq \frac{288R^2}{(\Delta_e^{\text{C}})^2} \ln \left(\frac{4nT^3}{\kappa}\right) + 1
$$

Summing over $e \in [n]$, we have

$$
T \leq \sum_{e \in [n]} \frac{288R^2}{(\Delta_e^{\text{C}})^2} \ln \left(\frac{4nT^3}{\kappa}\right) + n \leq \sum_{e \in [n]} \frac{864R^2}{(\Delta_e^{\text{C}})^2} \ln \left(\frac{2nT}{\kappa}\right) + n,
$$

where $\sum_{e \in [n]} \frac{R^2}{(\Delta_e^{\text{C}})^2} \geq n$. Then, applying Lemma 20, we have

$$
T \leq \sum_{e \in [n]} \frac{5184R^2}{(\Delta_e^{\text{C}})^2} \ln \left(\frac{2n^2}{\kappa} \sum_{e \in [n]} \frac{864R^2}{(\Delta_e^{\text{C}})^2}\right) + n
$$

$$=O\left(\sum_{e\in[n]}\frac{R^2}{(\Delta_e^{\mathsf{C}})^2}\ln\left(\sum_{e\in[n]}\frac{R^2n^2}{(\Delta_e^{\mathsf{C}})^2\kappa}\right)+n\right)$$

$$=O\left(\sum_{e\in[n]}\frac{R^2}{(\Delta_e^{\mathsf{C}})^2}\ln\left(\sum_{e\in[n]}\frac{R^2n}{(\Delta_e^{\mathsf{C}})^2\kappa}\right)\right).$$

$\square$

**Lemma 11.** *For algorithm* `BLUCB-Verify`, *assume that event* $\xi_0\cap\xi$ *occurs. For any* $e\in M_*$, *if* $\mathrm{rad}_t(e)<\frac{\Delta_e^{\mathsf{C}}}{8}=\frac{1}{8}(w(e)-\max_{M\neq M_*}\mathtt{MinW}(M,\boldsymbol{w}))$, *then, base arm* $e$ *will not be pulled at round t, i.e.,* $p_t\neq e$.

*Proof.* Suppose that for some $e\in M_*$, $\mathrm{rad}_t(e)<\frac{\Delta_e^{\mathsf{C}}}{8}=\frac{1}{8}(w(e)-\max_{M\neq M_*}\mathtt{MinW}(M,\boldsymbol{w}))$ and $p_t=e$.

Since event $\xi_0$ occurs, according to Lemma 6, we have that (i) $\hat{M}_*=M_*$, (ii) for any $M\neq M_*$, there exists a base arm $e\in\hat{B}_{\mathrm{sub}}\cap M$ satisfying $w(e)\leq\frac{1}{2}(\mathtt{MinW}(M_*,\boldsymbol{w})+\mathtt{MinW}(M,\boldsymbol{w}))$, i.e., $\Delta_{\mathtt{MinW}(M_*,\boldsymbol{w}),e}^{\mathsf{C}}\geq\frac{1}{2}\Delta_{M_*,M}^{\mathsf{C}}$ and (iii) for any $i\in\hat{B}_{\mathrm{sub}}$, there exists a sub-optimal super arm $M\neq M_*$ such that $i\in M$ and $w(i)\leq\frac{1}{2}(\mathtt{MinW}(M_*,\boldsymbol{w})+\mathtt{MinW}(M))\leq\frac{1}{2}(\mathtt{MinW}(M_*,\boldsymbol{w})+\mathtt{MinW}(M_{\mathrm{second}},\boldsymbol{w}))$. According to the selection strategy of $p_t$, we have that for any $i\in P_t^V$, $\mathrm{rad}_t(i)\leq\mathrm{rad}_t(e)<\frac{\Delta_e^{\mathsf{C}}}{8}$.

First, since for any $i\in\hat{B}_{\mathrm{sub}}$, $w(i)\leq\frac{1}{2}(\mathtt{MinW}(M_*,\boldsymbol{w})+\mathtt{MinW}(M_{\mathrm{second}},\boldsymbol{w}))<\mathtt{MinW}(M_*,\boldsymbol{w})$ and $w(e)\geq\mathtt{MinW}(M_*,\boldsymbol{w})$, we can obtain that $e=c_t$.

Then, for any $i\in F_t$, we have

$$\begin{aligned}\Delta_{e,i}^{\mathsf{C}}&=w(e)-w(i)\\&\geq w(e)-\frac{1}{2}(\mathtt{MinW}(M_*,\boldsymbol{w})+\mathtt{MinW}(M_{\mathrm{second}},\boldsymbol{w}))\\&\geq w(e)-\frac{1}{2}(w(e)+\mathtt{MinW}(M_{\mathrm{second}},\boldsymbol{w}))\\&=\frac{1}{2}(w(e)-\mathtt{MinW}(M_{\mathrm{second}},\boldsymbol{w}))\\&=\frac{1}{2}\Delta_e^{\mathsf{C}}\end{aligned}$$

and

$$\begin{aligned}\underline{w}_t(c_t)-\bar{w}_t(i)&=\underline{w}_t(e)-\bar{w}_t(i)\\&\geq w(e)-2\mathrm{rad}_t(e)-(w(i)+2\mathrm{rad}_t(i))\\&=\Delta_{e,i}^{\mathsf{C}}-2\mathrm{rad}_t(e)-2\mathrm{rad}_t(i)\\&>\frac{1}{2}\Delta_e^{\mathsf{C}}-\frac{\Delta_e^{\mathsf{C}}}{4}-\frac{\Delta_e^{\mathsf{C}}}{4}\\&=0,\end{aligned}$$

which implies $F_t=\varnothing$. Thus, for any $i\in\hat{B}_{\mathrm{sub}}$, $\underline{w}_t(c_t)\geq\bar{w}_t(i)$.

According to Lemma 6(ii), for any $M'\neq\hat{M}_*$, there exists a base arm $e'\in\hat{B}_{\mathrm{sub}}\cap M'$, we have

$$\begin{aligned}\mathtt{MinW}(\hat{M}_*,\underline{\boldsymbol{w}}_t)&=\underline{w}_t(c_t)\\&\geq\bar{w}_t(e')\\&\geq\mathtt{MinW}(M',\bar{\boldsymbol{w}}_t)\end{aligned}$$

which implies that algorithm 3 has already stopped. $\square$

**Lemma 12.** *For algorithm* `BLUCB-Verify`, *assume that event* $\xi_0\cap\xi$ *occurs. For any* $e\notin M_*$, $w(e)<\mathtt{MinW}(M_*,\boldsymbol{w})$, *if* $\mathrm{rad}_t(e)<\frac{\Delta_e^{\mathsf{C}}}{8}=\frac{1}{8}(\mathtt{MinW}(M_*,\boldsymbol{w})-\max_{M\in\mathcal{M}:e\in M}\mathtt{MinW}(M,\boldsymbol{w}))$, *then, base arm* $e$ *will not be pulled at round t, i.e.,* $p_t\neq e$.

*Proof.* Suppose that for some $e \notin M_*, w(e) < \text{MinW}(M_*, \boldsymbol{w}), \text{rad}_t(e) < \frac{\Delta_e^{\text{C}}}{8} = \frac{1}{8}(\text{MinW}(M_*, \boldsymbol{w}) - \max_{M \in \mathcal{M}: e \in M} \text{MinW}(M, \boldsymbol{w}))$ and $p_t = e$.

Since event $\xi_0$ occurs, according to Lemma 6, we have that (i) $\hat{M}_* = M_*$, (ii) for any $M \neq M_*$, there exists a base arm $e \in \hat{B}_{\text{sub}} \cap M$ satisfying $w(e) \leq \frac{1}{2}(\text{MinW}(M_*, \boldsymbol{w}) + \text{MinW}(M, \boldsymbol{w}))$, i.e., $\Delta_{\text{MinW}(M_*, \boldsymbol{w}), e}^{\text{C}} \geq \frac{1}{2}\Delta_{M_*, M}^{\text{C}}$ and (iii) for any $i \in \hat{B}_{\text{sub}}$, there exists a sub-optimal super arm $M \neq M_*$ such that $i \in M$ and $w(i) \leq \frac{1}{2}(\text{MinW}(M_*, \boldsymbol{w}) + \text{MinW}(M)) \leq \frac{1}{2}(\text{MinW}(M_*, \boldsymbol{w}) + \text{MinW}(M_{\text{second}}, \boldsymbol{w}))$. According to the selection strategy of $p_t$, we have that for any $i \in P_t^V$, $\text{rad}_t(i) \leq \text{rad}_t(e) < \frac{\Delta_e^{\text{C}}}{8}$.

Since $e \notin M_* = \hat{M}_*$, we have that $e \in F_t$ and there exists a sub-optimal super arm $M'$ such that $e \in M'$ and $w(e) \leq \frac{1}{2}(\text{MinW}(M_*, \boldsymbol{w}) + \text{MinW}(M'))$. Then, we have

$$
\begin{aligned}
\underline{w}_t(c_t) - \bar{w}_t(e) \geq & w(c_t) - 2\text{rad}_t(c_t) - (w(e) + 2\text{rad}_t(e)) \\
\geq & \text{MinW}(M_*, \boldsymbol{w}) - \frac{1}{2}(\text{MinW}(M_*, \boldsymbol{w}) + \text{MinW}(M', \boldsymbol{w})) - 2\text{rad}_t(c_t) - 2\text{rad}_t(e) \\
> & \frac{1}{2}\Delta_{M_*, M'}^{\text{C}} - \frac{\Delta_e^{\text{C}}}{4} - \frac{\Delta_e^{\text{C}}}{4} \\
= & \frac{1}{2}\Delta_{M_*, M'}^{\text{C}} - \frac{1}{2}(\text{MinW}(M_*, \boldsymbol{w}) - \max_{M \in \mathcal{M}: e \in M} \text{MinW}(M, \boldsymbol{w})) \\
\geq & \frac{1}{2}\Delta_{M_*, M'}^{\text{C}} - \frac{1}{2}(\text{MinW}(M_*, \boldsymbol{w}) - \text{MinW}(M', \boldsymbol{w})) \\
= & 0,
\end{aligned}
$$

which contradicts $e \in F_t$. $\qquad\square$

Recall that $B = \{e \mid e \notin M_*, w(e) < \text{MinW}(M_*, \boldsymbol{w})\}$ and $B^c = \{e \mid e \notin M_*, w(e) \geq \text{MinW}(M_*, \boldsymbol{w})\}$.

**Theorem 6** (Sample Complexity of BLUCB-Verify). *With probability at least $1 - \kappa - \delta^V$, the BLUCB-Verify algorithm (Algorithm 3) will return $M_*$ with sample complexity*

$$
O\left(\sum_{e \in B^c} \frac{R^2}{(\Delta_e^{\text{C}})^2} \ln\left(\sum_{e \in B^c} \frac{R^2 n}{(\Delta_e^{\text{C}})^2}\right) + \sum_{e \in M_* \cup B} \frac{R^2}{(\Delta_e^{\text{C}})^2} \ln\left(\sum_{e \in M_* \cup B} \frac{R^2 n}{(\Delta_e^{\text{C}})^2 \delta^V}\right)\right).
$$

*Proof.* Assume that event $\xi_0 \cap \xi$ occurs. $\Pr[\xi_0 \cap \xi] \geq 1 - \kappa - \delta^V$.

First, we prove the correctness. According to Lemma 6, the hypothesized $\hat{M}_*$ outputted by the preparation procedure BLUCB-Explore is exactly the optimal super arm, and thus if BLUCB-Verify terminates, it returns the correct answer.

Next, we prove the sample complexity upper bound. According to Theorem 5, the preparation procedure BLUCB-Explore costs sample complexity

$$
O\left(\sum_{e \in [n]} \frac{R^2}{(\Delta_e^{\text{C}})^2} \ln\left(\sum_{e \in [n]} \frac{R^2 n}{(\Delta_e^{\text{C}})^2 \kappa}\right)\right).
$$

Then, we bound the sample complexity of the following verification part. Following the analysis procedure of Theorem 5 with Lemmas 11,12, we can obtain that the verification part cost sample complexity

$$
\sum_{e \in M_* \cup B} \frac{R^2}{(\Delta_e^{\text{C}})^2} \ln\left(\sum_{e \in M_* \cup B} \frac{R^2 n}{(\Delta_e^{\text{C}})^2 \delta^V}\right).
$$

Combining both parts, we obtain that the sample complexity is bounded by

$$
O\left(\sum_{e \in B^c} \frac{R^2}{(\Delta_e^{\text{C}})^2} \ln\left(\sum_{e \in B^c} \frac{R^2 n}{(\Delta_e^{\text{C}})^2 \kappa}\right) + \sum_{e \in M_* \cup B} \frac{R^2}{(\Delta_e^{\text{C}})^2} \ln\left(\sum_{e \in M_* \cup B} \frac{R^2 n}{(\Delta_e^{\text{C}})^2 \delta^V}\right)\right)
$$

$$=O\left(\sum_{e\in B^c}\frac{R^2}{(\Delta_e^{\mathrm{C}})^2}\ln\left(\sum_{e\in B^c}\frac{R^2n}{(\Delta_e^{\mathrm{C}})^2}\right)+\sum_{e\in M_*\cup B}\frac{R^2}{(\Delta_e^{\mathrm{C}})^2}\ln\left(\sum_{e\in M_*\cup B}\frac{R^2n}{(\Delta_e^{\mathrm{C}})^2\delta^V}\right)\right).$$

$\square$

**Lemma 13** (Correctness of `BLUCB-Verify`). *With probability at least $1-\delta$, if algorithm `BLUCB-Verify` (Algorithm 3) terminates, it returns the optimal super arm $M_*$.*

*Proof.* Assume that event $\xi$ occurs, where $\Pr[\xi]\geq 1-\delta^V$. If algorithm `BLUCB-Verify` terminates by returning $\hat{M}_*$, we have that for any $M\in\mathcal{M}\setminus\mathcal{S}(\hat{M}_*)$,

$$\texttt{MinW}(\hat{M}_*,\boldsymbol{w})\geq\texttt{MinW}(\hat{M}_*,\underline{\boldsymbol{w}}_t)\geq\texttt{MinW}(M,\bar{\boldsymbol{w}}_t)\geq\texttt{MinW}(M,\boldsymbol{w}).$$

For any $M\in\mathcal{S}(\hat{M}_*)$, according to the property of the bottleneck reward function, we have

$$\texttt{MinW}(\hat{M}_*,\boldsymbol{w})\geq\texttt{MinW}(M,\boldsymbol{w}).$$

Thus, we have $\texttt{MinW}(\hat{M}_*,\boldsymbol{w})\geq\texttt{MinW}(M,\boldsymbol{w})$ for any $M\neq\hat{M}_*$ and according to the unique assumption of $M_*$, we obtain $\hat{M}_*=M_*$. In other words, algorithm `BLUCB-Verify` (Algorithm 3) will never return a wrong answer. $\square$

Now, we prove Theorem 2.

*Proof.* Using Theorem 6, Lemma 13 and Lemma 4.8 in [10], we can obtain this theorem. $\square$

### B.3 PAC Learning

In this subsection, we further study the fixed-confidence CPE-B problem in the PAC learning setting, where the learner's objective is to identify a super arm $M_{\mathrm{pac}}$ such that $\texttt{MinW}(M_{\mathrm{pac}},\boldsymbol{w})\geq \mathrm{OPT}-\varepsilon$, and the uniqueness assumption of the optimal super arm is dropped. We propose two algorithms `BLUCB-PAC` and `BLUCB-Parallel-PAC` for the PAC learning setting, based on `BLUCB` and `BLUCB-Parallel`, respectively.

The PAC algorithms and their theoretical guarantees do not require the uniqueness assumption of the optimal super arm. Compared to the PAC lower bound, both proposed PAC algorithms achieve the optimal sample complexity for some family of instances. Similar to the exact case, when $\delta$ is small enough, the dominant term of sample complexity for algorithm `BLUCB-Parallel-PAC` does not depend on the reward gaps of unnecessary base arms, and thus `BLUCB-Parallel-PAC` achieves better theoretical guarantee than `BLUCB-PAC` and matches the lower bound for a broader family of instances.

#### B.3.1 Algorithm `BLUCB-PAC`

`BLUCB-PAC` simply replaces the stopping condition of `BLUCB` (Line 9 in Algorithm 1) with $\texttt{MinW}(\tilde{M}_t,\bar{\boldsymbol{w}}_t)-\texttt{MinW}(M_t,\underline{\boldsymbol{w}}_t)\leq\varepsilon$ to allow an $\varepsilon$ deviation between the returned answer and the optimal one. The sample complexity of algorithm `BLUCB-PAC` is given as follows.

**Theorem 7** (Fixed-confidence Upper Bound for PAC). *With probability at least $1-\delta$, the `BLUCB-PAC` algorithm will return $M_{\mathrm{out}}$ such that $\texttt{MinW}(M_{\mathrm{out}},\boldsymbol{w})\geq\texttt{MinW}(M_*,\boldsymbol{w})-\varepsilon$, with sample complexity*

$$O\left(\sum_{e\in[n]}\frac{R^2}{\max\{(\Delta_e^{\mathrm{C}})^2,\varepsilon^2\}}\ln\left(\sum_{e\in[n]}\frac{R^2n}{\max\{(\Delta_e^{\mathrm{C}})^2,\varepsilon^2\}\delta}\right)\right).$$

Now we prove the sample complexity of algorithm `BLUCB-PAC` (Theorem 7).

*Proof.* First, we prove the correctness. When the stop condition of `BLUCB-PAC` is satisfied, we have that for any $M\neq M_t$,

$$\texttt{MinW}(M,\boldsymbol{w})-\texttt{MinW}(M_t,\boldsymbol{w})\leq\texttt{MinW}(M,\bar{\boldsymbol{w}}_t)-\texttt{MinW}(M_t,\underline{\boldsymbol{w}}_t)\leq\texttt{MinW}(\tilde{M}_t,\bar{\boldsymbol{w}}_t)-\texttt{MinW}(M_t,\underline{\boldsymbol{w}}_t)\leq\varepsilon.$$

If $M_t = M_*$, then the correctness holds. If $M_t \neq M_*$, the returned super arm $M_t$ satisfies

$$\texttt{MinW}(M_*, \boldsymbol{w}) - \texttt{MinW}(M_t, \boldsymbol{w}) \leq \varepsilon,$$

which guarantees the correctness.

Next, we prove the sample complexity. When inheriting the proof of Theorem 1 for the baseline algorithm BLUCB, to prove Theorem 7 for PAC leaning, it suffices to prove that for any $e \in [n]$, if $\text{rad}_t(e) < \frac{\varepsilon}{2}$, base arm $e$ will not be pulled at round $t$, i.e., $p_t \neq e$.

Suppose that $\text{rad}_t(e) < \frac{\varepsilon}{2}$ and $p_t = e$. According to the selection strategy of $p_t$, we have $\text{rad}_t(c_t) < \frac{\varepsilon}{2}$ and $\text{rad}_t(d_t) < \frac{\varepsilon}{2}$. According to the definition of $d_t$, we have

$$
\begin{aligned}
\texttt{MinW}(\tilde{M}_t, \bar{\boldsymbol{w}}_t) - \texttt{MinW}(M_t, \underline{\boldsymbol{w}}_t) \leq & \bar{w}(d_t) - \texttt{MinW}(\tilde{M}_t, \underline{\boldsymbol{w}}_t) \\
= & \bar{w}(d_t) - \underline{w}(d_t) \\
= & 2\text{rad}_t(d_t) \\
< & \varepsilon,
\end{aligned}
$$

which contradicts the stop condition. $\qquad\square$

### B.3.2 Algorithm BLUCB-Parallel-PAC

BLUCB-Parallel-PAC is obtained by simply replacing the stopping condition of BLUCB-Verify (Line 10 in Algorithm 3) with $\texttt{MinW}(\tilde{M}_t, \bar{\boldsymbol{w}}_t) - \texttt{MinW}(M_t, \underline{\boldsymbol{w}}_t) \leq \varepsilon$.

Theorem 8 presents the sample complexity of algorithm BLUCB-Parallel-PAC.

**Theorem 8** (Improved Fixed-confidence Upper Bound for PAC). *For any $\delta < 0.01$, with probability at least $1 - \delta$, algorithm BLUCB-Parallel-PAC returns $M_*$ and takes the expected sample complexity*

$$
O\left( \sum_{e \in M_* \cup N} \frac{R^2}{\max\{(\Delta_e^{\mathsf{C}})^2, \varepsilon^2\}} \ln\left( \sum_{e \in M_* \cup N} \frac{R^2 n}{\max\{(\Delta_e^{\mathsf{C}})^2, \varepsilon^2\}\delta} \right) + \sum_{e \in \tilde{N}} \frac{R^2}{(\Delta_e^{\mathsf{C}})^2} \ln\left( \sum_{e \in \tilde{N}} \frac{R^2 n}{(\Delta_e^{\mathsf{C}})^2} \right) \right).
$$

*Proof.* First, we prove the correctness. When the stop condition of BLUCB-Verify is satisfied, we have that for any $M \neq \hat{M}_*$,

$$\texttt{MinW}(M, \boldsymbol{w}) - \texttt{MinW}(\hat{M}_*, \boldsymbol{w}) \leq \texttt{MinW}(M, \bar{\boldsymbol{w}}_t) - \texttt{MinW}(\hat{M}_*, \underline{\boldsymbol{w}}_t) \leq \texttt{MinW}(\tilde{M}_t, \bar{\boldsymbol{w}}_t) - \texttt{MinW}(\hat{M}_*, \underline{\boldsymbol{w}}_t) \leq \varepsilon.$$

If $\hat{M}_* = M_*$, then the correctness holds. If $\hat{M}_* \neq M_*$, the returned super arm $\hat{M}_*$ satisfies

$$\texttt{MinW}(M_*, \boldsymbol{w}) - \texttt{MinW}(\hat{M}_*, \boldsymbol{w}) \leq \varepsilon,$$

which guarantees the correctness.

Next, we prove the sample complexity. We inherit the proofs of Theorems 2,6. Then, to prove Theorem 8 for PAC leaning, it suffices to prove that conditioning on $\xi_0 \cap \xi$, for any $e \in [n]$, if $\text{rad}_t(e) < \frac{\varepsilon}{4}$, base arm $e$ will not be pulled at round $t$, i.e., $p_t \neq e$.

Suppose that $\text{rad}_t(e) < \frac{\varepsilon}{4}$ and $p_t = e$. According to the selection strategy of $p_t$, we have $\text{rad}_t(c_t) < \frac{\varepsilon}{4}$ and for any $e \in F_t$ $\text{rad}_t(e) < \frac{\varepsilon}{4}$. Using $F_t \subseteq \hat{B}_{\text{sub}}$ and the definition of $\hat{B}_{\text{sub}}$, we have that for any $e \in F_t$

$$
\begin{aligned}
\bar{w}(e) - \texttt{MinW}(\hat{M}_*, \underline{\boldsymbol{w}}_t) \leq & w(e) + 2\text{rad}_t(e) - (w(c_t) - 2\text{rad}_t(c_t)) \\
< & \texttt{MinW}(\hat{M}_*, \boldsymbol{w}) + 2\text{rad}_t(e) - (\texttt{MinW}(\hat{M}_*, \boldsymbol{w}) - 2\text{rad}_t(c_t)) \\
< & 4 \cdot \frac{\varepsilon}{4} \\
= & \varepsilon
\end{aligned}
$$

and thus

$$\texttt{MinW}(\tilde{M}_t, \bar{\boldsymbol{w}}_t) - \texttt{MinW}(\hat{M}_*, \underline{\boldsymbol{w}}_t) \leq \varepsilon,$$

which contradicts the stop condition. $\qquad\square$

# C  Lower Bounds for the Fixed-Confidence Setting

In this section, we present the proof of lower bound for the exact fixed-confidence CPE-B problem. Then, we also provide a lower bound for the PAC fixed-confidence CPE-B problem and give its proof.

First, we prove the lower bound for the exact fixed-confidence CPE-B problem (Theorem 3). Notice that, the sample complexity of algorithm BLUCB also matches the lower bound within a logarithmic factor if we replace condition (iii) below with that each sub-optimal super arm only has a single base arm.

*Proof.* Consider an instance $\mathcal{I}$ of the fixed-confidence CPE-B problem such that: (i) the reward distribution of each base arm $e \in [n]$ is $\mathcal{N}(w(e), R)$; (ii) both $M_*$ and the second best super arms are unique, and the second best super arm has no overlapped base arm with $M_*$; (iii) in each sub-optimal super arm, there is a single base arm with reward below $\texttt{MinW}(M_*, \boldsymbol{w})$.

Fix an arbitrary $\delta$-correct algorithm $\mathbb{A}$. For an arbitrary base arm $e \in M_*$, we construct an instance $\mathcal{I}'$ by changing its reward distribution to $\mathcal{N}(w'(e), R)$ where $w'(e) = w(e) - 2\Delta_e^\mathsf{C}$. Recall that $M_\text{second} = \text{argmax}_{M \neq M_*} \texttt{MinW}(M, \boldsymbol{w})$. For instance $\mathcal{I}'$, from the definition of $\Delta_e^\mathsf{C}$ (Definition 1),

$$
\begin{aligned}
w'(e) =& w(e) - 2\Delta_e^\mathsf{C} \\
=& w(e) - (w(e) - \texttt{MinW}(M_\text{second}, \boldsymbol{w})) - \Delta_e^\mathsf{C} \\
=& \texttt{MinW}(M_\text{second}, \boldsymbol{w}) - \Delta_e^\mathsf{C} \\
<& \texttt{MinW}(M_\text{second}, \boldsymbol{w})
\end{aligned}
$$

and $\texttt{MinW}(M_*, \boldsymbol{w}') = w'(e) < \texttt{MinW}(M_\text{second}, \boldsymbol{w})$. Thus, $M_\text{second}$ becomes the optimal super arm.

Let $T_e$ denote the number of samples drawn from base arm $e$ when algorithm $\mathbb{A}$ runs on instance $\mathcal{I}$. Let $d(x, y) = x \ln(x/y) + (1 - x) \ln[(1 - x)/(1 - y)]$ denote the binary relative entropy function. Define $\mathcal{H}$ as the event that algorithm $\mathbb{A}$ returns $M_*$. Since $\mathbb{A}$ is $\delta$-correct, we have $\Pr_{\mathbb{A}, \mathcal{I}}[\mathcal{H}] \geq 1 - \delta$ and $\Pr_{\mathbb{A}, \mathcal{I}'}[\mathcal{H}] \leq \delta$. Thus, $d(\Pr_{\mathbb{A}, \mathcal{I}}[\mathcal{H}], \Pr_{\mathbb{A}, \mathcal{I}'}[\mathcal{H}]) \geq d(1 - \delta, \delta)$. Using Lemma 1 in [27], we can obtain

$$
\mathbb{E}[T_e]\text{KL}(\mathcal{N}(w(e), R), \mathcal{N}(w'(e), R)) \geq d(1 - \delta, \delta),
$$

Since the reward distribution of each base arm is Gaussian distribution, we have $\text{KL}(\mathcal{N}(w(e), R), \mathcal{N}(w(e'), R)) = \frac{1}{2R^2}(w(e) - w'(e))^2 = \frac{2}{R^2}(\Delta_e^\mathsf{C})^2$. Since $\delta \in (0, 0.1)$, $d(1 - \delta, \delta) \geq 0.4 \ln(1/\delta)$. Thus, we have

$$
\frac{2}{R^2}(\Delta_e^\mathsf{C})^2 \cdot \mathbb{E}[T_e] \geq 0.4 \ln(\frac{1}{\delta}).
$$

Then,

$$
\mathbb{E}[T_e] \geq 0.2 \frac{R^2}{(\Delta_e^\mathsf{C})^2} \ln(\frac{1}{\delta}).
$$

For an arbitrary base arm $e \notin M_*, w(e) < \texttt{MinW}(M_*, \boldsymbol{w})$, we can construct another instance $\mathcal{I}'$ by changing its reward distribution to $\mathcal{N}(w'(e), R)$ where $w'(e) = w(e) + 2\Delta_e^\mathsf{C}$. Let $M_e$ denote the sub-optimal super arm that contains $e$.

For instance $\mathcal{I}'$, from the definition of $\Delta_e^\mathsf{C}$ (Definition 1),

$$
\begin{aligned}
w'(e) =& w(e) + 2\Delta_e^\mathsf{C} \\
=& w(e) + (\texttt{MinW}(M_*, \boldsymbol{w}) - \texttt{MinW}(M_e)) + \Delta_e^\mathsf{C} \\
=& w(e) + (\texttt{MinW}(M_*, \boldsymbol{w}) - w(e)) + \Delta_e^\mathsf{C} \\
=& \texttt{MinW}(M_*, \boldsymbol{w}) + \Delta_e^\mathsf{C} \\
>& \texttt{MinW}(M_*, \boldsymbol{w}).
\end{aligned}
$$

Thus, $M_e$ become the optimal super arm. Similarly, using Lemma 1 in [27] we can obtain

$$
\frac{2}{R^2}(\Delta_e^\mathsf{C})^2 \cdot \mathbb{E}[T_e] \geq 0.4 \ln(\frac{1}{\delta}).
$$

---
**Algorithm 7** AR-Oracle
---
1: **Input:** decision class $\mathcal{M}$, accepted base arm $a$, set of the rejected base arms $R$ and weight vector $\boldsymbol{v}$.
2: Remove the base arms in $R$ from $\mathcal{M}$ and obtain a new decision class $\mathcal{M}_{-R}$.
3: Sort the remaining base arms by descending rewards and denote them by $e_{(1)}, \ldots, e_{(n-|R|)}$
4: **for** $e = e_{(1)}, \ldots, e_{(n-|R|)}$ **do**
5:     Remove all base arms with the rewards lower than $w(e)$ from $\mathcal{M}$ and obtain a new decision class $\mathcal{M}_{-R,\geq w(e)}$
6:     $M_{\text{out}} \leftarrow \texttt{ExistOracle}(\mathcal{M}_{-R,\geq w(e)}, a)$
7:     **if** $M_{\text{out}} \neq \perp$ **then**
8:         **return** $M_{\text{out}}$
9:     **end if**
10: **end for**
---

Then,

$$\mathbb{E}[T_e] \geq 0.2 \frac{R^2}{(\Delta_e^{\mathsf{C}})^2} \ln(\frac{1}{\delta}).$$

Summing over all $e \in M_*$ and $e \notin M_*, w(e) < \texttt{MinW}(M_*, \boldsymbol{w})$, we can obtain that any $\delta$-correct algorithm has sample complexity

$$\Omega\left(\sum_{e \in M_* \cup B} \frac{R^2}{(\Delta_e^{\mathsf{C}})^2} \ln\left(\frac{1}{\delta}\right)\right).$$

$\square$

Next, we present the lower bound for the PAC fixed-confidence CPE-B problem, where we can relax condition (ii) in the proof of the exact lower bound (Theorem 3). To formally state our result, we first introduce the notion of $(\delta, \varepsilon)$-*correct algorithm* as follows. For any confidence parameter $\delta \in (0, 1)$ and accuracy parameter $\varepsilon > 0$, we call an algorithm $\mathcal{A}$ a $(\delta, \varepsilon)$-correct algorithm if for the fixed-confidence CPE-B in PAC learning, $\mathcal{A}$ returns a super arm $M_{\text{pac}}$ such that $\texttt{MinW}(M_{\text{pac}}, \boldsymbol{w}) \geq \texttt{MinW}(M_*, \boldsymbol{w}) - \varepsilon$ with probability at least $1 - \delta$.

**Theorem 9** (Fixed-confidence Lower Bound for PAC). *There exists a family of instances for the fixed-confidence CPE-B problem, where for any $\delta \in (0, 0.1)$, any $(\delta, \varepsilon)$-correct algorithm has the expected sample complexity*

$$\Omega\left(\sum_{e \in M_* \cup B} \frac{R^2}{\max\{(\Delta_e^{\mathsf{C}})^2, \varepsilon^2\}} \ln\left(\frac{1}{\delta}\right)\right).$$

*Proof.* Consider the instance $\mathcal{I}$ for the PAC fixed-confidence CPE-B problem, where $\varepsilon < \texttt{MinW}(M_*, \boldsymbol{w}) - \texttt{MinW}(M_{\text{second}}, \boldsymbol{w})$ (to guarantee that $M_*$ is unique) and (i) the reward distribution of each base arm $e \in [n]$ is $\mathcal{N}(w(e), 1)$; (ii) the PAC solution $M_{\text{pac}}$ is unique and the second best super arm has no overlapped base arm with $M_{\text{pac}}$; (iii) in each sub-optimal super arm, there is a single base arm with reward below $\texttt{MinW}(M_*, \boldsymbol{w})$. Then, following the proof procedure of Theorem 3, we can obtain Theorem 9. $\square$

## D   CPE-B in the Fixed-Budget Setting

In this section, we present the implementation details of AR-Oracle and error probability proof for algorithm BSAR.

### D.1   Implementation Details of AR-Oracle

First, we discuss AR-Oracle. Recall that AR-Oracle $\in \operatorname{argmax}_{M \in \mathcal{M}(e,R)} \texttt{MinW}(M, \boldsymbol{w})$, where $\mathcal{M}(e, R) = \{M \in \mathcal{M} : e \in M, R \cap M = \varnothing\}$. If $\mathcal{M}(e, R) = \varnothing$, AR-Oracle $= \perp$. Algorithm 7

gives the algorithm pseudo-code of `AR-Oracle`. As `BottleneckSearch`, `AR-Oracle` also uses the existence oracle `ExistOracle` to find a feasible super arm that contains some base arm from the given decision class if there exists, and otherwise return $\perp$. We explain the procedure of `AR-Oracle` as follows: we first remove all base arms in $R$ from the decision class $\mathcal{M}$ to disable the super arms that contain the rejected base arms. Then, we enumerate the remaining base arms by descending rewards. For each enumerated base arm $e$, we remove the base arms with rewards lower than $w(e)$ and obtain a new decision class $\mathcal{M}_{-R,\geq w(e)}$, and then use `ExistOracle` to find a feasible super arm that contains the accepted base arm $a$ from $\mathcal{M}_{-R,\geq w(e)}$. Once such a feasible super arm is found, the procedure terminates and returns this super arm. Since the enumeration of base arms is performed according to descending rewards and the computed decision class only contains base arms no worse than the enumerated one, `AR-Oracle` guarantees to return an optimal super arm from $\mathcal{M}(e, R)$.

As for the computational efficiency, the time complexity of `AR-Oracle` mainly depends on the step of finding a feasible super arm containing some base arm $e$. Fortunately, this existence problem can be solved in polynomial time for a wide family of decision classes. For example, for $s$-$t$ paths, this problem can be reduced to the well-studied 2-vertex connectivity problem [20], which is polynomially tractable (see Section E.1 for the proof of reduction). For maximum cardinality matchings, we just need to remove $e$ and its two end vertices, and then find a feasible maximum cardinality matching in the remaining graph; and for spanning trees, we can just merge the vertices of $e$ and find a feasible spanning tree in the remaining graph. All of the above cases can be solved efficiently.

### D.2 Proof for Algorithm `BSAR`

Below we present the proof of error probability for algorithm `BSAR`. To prove Theorem 4, we first introduce the lowing Lemmas 14-17.

**Lemma 14.** *For phase $t = 1, \ldots, n$, define events*

$$\mathcal{E}_t = \left\{ \forall i \in [n] \setminus (A_t \cup R_t), \ |\hat{w}_t(i) - w(i)| < \frac{\Delta^{\mathrm{B}}_{(n+1-t)}}{8} \right\}.$$

*and $\mathcal{E} \triangleq \bigcap_{t=1}^{n} \mathcal{E}_t$. Then, we have*

$$\Pr[\mathcal{E}] \geq 1 - 2n^2 \exp\left( -\frac{(T-n)}{128 \tilde{\log}(n) R^2 H^B} \right).$$

*Proof.* For any $t \in [n]$ and $e \in [n] \setminus (A_t \cup R_t)$, according to the Hoeffding's inequality,

$$\left\{ |\hat{w}_t(i) - w(i)| \geq \frac{\Delta^{\mathrm{B}}_{(n+1-t)}}{8} \right\} \leq 2 \exp\left( -\frac{\tilde{T}(\Delta^{\mathrm{B}}_{n-t+1})^2}{128 R^2} \right).$$

From the definition of $\tilde{T}$ and $H^B$, we have

$$
\begin{aligned}
\Pr\left[ |\hat{w}_t(i) - w(i)| \geq \frac{\Delta^{\mathrm{B}}_{(n+1-t)}}{8} \right] &\leq 2 \exp\left( -\frac{\tilde{T}(\Delta^{\mathrm{B}}_{n-t+1})^2}{128 R^2} \right) \\
&\leq 2 \exp\left( -\frac{\frac{T-n}{\tilde{\log}(n)(n-t+1)}(\Delta^{\mathrm{B}}_{n-t+1})^2}{128 R^2} \right) \\
&= 2 \exp\left( -\frac{(T-n)}{128 \tilde{\log}(n) R^2 \frac{n-t+1}{(\Delta^{\mathrm{B}}_{n-t+1})^2}} \right) \\
&\leq 2 \exp\left( -\frac{(T-n)}{128 \tilde{\log}(n) R^2 H^B} \right)
\end{aligned}
$$

By a union bound over $t \in [n]$ and $e \in [n] \setminus (A_t \cup R_t)$, we have

$$\Pr[\mathcal{E}] \geq 1 - n^2 \Pr\left[ |\hat{w}_t(i) - w(i)| \geq \frac{\Delta^{\mathrm{B}}_{(n+1-t)}}{8} \right]$$

$$\geq 1 - 2n^2 \exp\left(-\frac{(T-n)}{128 \tilde{\log}(n) R^2 H^B}\right).$$

$\square$

**Lemma 15.** *Fix any phase $t > 0$. Assume that event $\mathcal{E}_t$ occurs and algorithm BSAR does not make any mistake before phase $t$, i.e., $A_t \subseteq M_*$ and $R_t \cap M_* = \varnothing$. Then, for any $e \in [n] \setminus (A_t \cup R_t)$ s.t. $\Delta_e^B \geq \Delta_{(n+1-t)}^B$, we have $e \in (M_* \cap M_t) \cup (\neg M_* \cap \neg M_t)$.*

*Proof.* Suppose that $e \in (M_* \cap \neg M_t) \cup (\neg M_* \cap M_t)$.

Case (I). If $e \in M_*, e \notin M_t$, then $M_t$ is a sub-optimal super arm and $\Delta_{M_*, M_t}^B \geq \Delta_e^B \geq \Delta_{(n+1-t)}^B$. Then, we have

$$
\begin{aligned}
\text{MinW}(M_*, \hat{\boldsymbol{w}}_t) - \text{MinW}(M_t, \hat{\boldsymbol{w}}_t) &> \text{MinW}(M_*, \boldsymbol{w}) - \frac{1}{8}\Delta_{(n+1-t)}^B - \left(\text{MinW}(M_t, \boldsymbol{w}) + \frac{1}{8}\Delta_{(n+1-t)}^B\right) \\
&= \text{MinW}(M_*, \boldsymbol{w}) - \text{MinW}(M_t, \boldsymbol{w}) - \frac{1}{4}\Delta_{(n+1-t)}^B \\
&\geq \Delta_e^B - \frac{1}{4}\Delta_{(n+1-t)}^B \\
&\geq \frac{3}{4}\Delta_{(n+1-t)}^B \\
&> 0,
\end{aligned}
$$

which contradicts the definition of $M_t$.

Case (II). If $e \in M_t, e \notin M_*$, then $M_t$ is a sub-optimal super arm and $\Delta_{M_*, M_t}^B \geq \Delta_e^B \geq \Delta_{(n+1-t)}^B$. Then, we have

$$
\begin{aligned}
\text{MinW}(M_*, \hat{\boldsymbol{w}}_t) - \text{MinW}(M_t, \hat{\boldsymbol{w}}_t) &> \text{MinW}(M_*, \boldsymbol{w}) - \frac{1}{8}\Delta_{(n+1-t)}^B - \left(\text{MinW}(M_t, \boldsymbol{w}) + \frac{1}{8}\Delta_{(n+1-t)}^B\right) \\
&= \text{MinW}(M_*, \boldsymbol{w}) - \text{MinW}(M_t, \boldsymbol{w}) - \frac{1}{4}\Delta_{(n+1-t)}^B \\
&\geq \Delta_e^B - \frac{1}{4}\Delta_{(n+1-t)}^B \\
&\geq \frac{3}{4}\Delta_{(n+1-t)}^B \\
&> 0,
\end{aligned}
$$

which contradicts the definition of $M_t$.

Thus, the supposition does not hold and we obtain $e \in (M_* \cap M_t) \cup (\neg M_* \cap \neg M_t)$. $\square$

**Lemma 16.** *Fix any phase $t > 0$. Assume that event $\mathcal{E}_t$ occurs and algorithm BSAR does not make any mistake before phase $t$, i.e., $A_t \subseteq M_*$ and $R_t \cap M_* = \varnothing$. Then, there exists some base arm $e \in [n] \setminus (A_t \cup R_t)$ s.t. $\Delta_e^B \geq \Delta_{(n+1-t)}^B$ and this base arm $e$ satisfies*

$$\text{MinW}(M_t, \hat{\boldsymbol{w}}_t) - \text{MinW}(\tilde{M}_{t,e}, \hat{\boldsymbol{w}}_t) > \frac{3}{4}\Delta_{(n+1-t)}^B.$$

*Proof.* Since $e \in [n] \setminus (A_t \cup R_t)$ and $\Delta_e^B \geq \Delta_{(n+1-t)}^B$, according to Lemma 15, we have $e \in (M_* \cap M_t) \cup (\neg M_* \cap \neg M_t)$.

Case (I). If $e \in (M_* \cap M_t)$, then $e \notin \tilde{M}_{t,e}$ (if $\tilde{M}_{t,e} = \perp$ then the lemma trivially holds) and $\Delta_{M_*, \tilde{M}_{t,e}}^B \geq \Delta_e^B$. We have

$$
\begin{aligned}
\text{MinW}(M_*, \hat{\boldsymbol{w}}_t) - \text{MinW}(\tilde{M}_{t,e}, \hat{\boldsymbol{w}}_t) &> \text{MinW}(M_*, \boldsymbol{w}) - \frac{1}{8}\Delta_{(n+1-t)}^B - \left(\text{MinW}(\tilde{M}_{t,e}, \boldsymbol{w}) + \frac{1}{8}\Delta_{(n+1-t)}^B\right) \\
&= \text{MinW}(M_*, \boldsymbol{w}) - \text{MinW}(\tilde{M}_{t,e}, \boldsymbol{w}) - \frac{1}{4}\Delta_{(n+1-t)}^B \\
&\geq \Delta_e^C - \frac{1}{4}\Delta_{(n+1-t)}^B
\end{aligned}
$$

$$\geq \frac{3}{4}\Delta^{\mathtt{B}}_{(n+1-t)}.$$

Case (II). If $e \in (\neg M_* \cap \neg M_t)$, then $e \in \tilde{M}_{t,e}$ (if $\tilde{M}_{t,e} = \perp$ then the lemma trivially holds) and $\Delta^{\mathtt{B}}_{M_*, \tilde{M}_{t,e}} \geq \Delta^{\mathtt{B}}_e$. We have

$$\mathtt{MinW}(M_*, \hat{\boldsymbol{w}}_t) - \mathtt{MinW}(\tilde{M}_{t,e}, \hat{\boldsymbol{w}}_t) > \mathtt{MinW}(M_*, \boldsymbol{w}) - \frac{1}{8}\Delta^{\mathtt{B}}_{(n+1-t)} - \left(\mathtt{MinW}(\tilde{M}_{t,e}, \boldsymbol{w}) + \frac{1}{8}\Delta^{\mathtt{B}}_{(n+1-t)}\right)$$

$$= \mathtt{MinW}(M_*, \boldsymbol{w}) - \mathtt{MinW}(\tilde{M}_{t,e}, \boldsymbol{w}) - \frac{1}{4}\Delta^{\mathtt{B}}_{(n+1-t)}$$

$$\geq \Delta^{\mathtt{C}}_e - \frac{1}{4}\Delta^{\mathtt{B}}_{(n+1-t)}$$

$$\geq \frac{3}{4}\Delta^{\mathtt{B}}_{(n+1-t)}.$$

Combining cases (I) and (II), we obtain the lemma. $\qquad \square$

**Lemma 17.** *Fix any phase $t > 0$. Assume that event $\mathcal{E}_t$ occurs and algorithm* BSAR *does not make any mistake before phase $t$, i.e., $A_t \subseteq M_*$ and $R_t \cap M_* = \varnothing$. Then, for any $p \in [n] \setminus (A_t \cup R_t)$ s.t. $p \in (M_* \cap \neg M_t) \cup (\neg M_* \cap M_t)$, we have*

$$\mathtt{MinW}(M_t, \hat{\boldsymbol{w}}_t) - \mathtt{MinW}(\tilde{M}_{t,p}, \hat{\boldsymbol{w}}_t) < \frac{1}{4}\Delta^{\mathtt{B}}_{(n+1-t)}$$

*Proof.* Since $p \in (M_* \cap \neg M_t) \cup (\neg M_* \cap M_t)$, then $M_t$ is a sub-optimal super arm and $\Delta^{\mathtt{B}}_{M_t, M_*} < 0$. We have

$$\mathtt{MinW}(M_t, \hat{\boldsymbol{w}}_t) - \mathtt{MinW}(M_*, \hat{\boldsymbol{w}}_t) < \mathtt{MinW}(M_t, \boldsymbol{w}) + \frac{1}{8}\Delta^{\mathtt{B}}_{(n+1-t)} - \left(\mathtt{MinW}(M_*, \boldsymbol{w}) - \frac{1}{8}\Delta^{\mathtt{B}}_{(n+1-t)}\right)$$

$$= \mathtt{MinW}(M_t, \boldsymbol{w}) - \mathtt{MinW}(M_*, \boldsymbol{w}) + \frac{1}{4}\Delta^{\mathtt{B}}_{(n+1-t)}$$

$$< \frac{1}{4}\Delta^{\mathtt{B}}_{(n+1-t)}.$$

Since $p \in (M_* \cap \neg M_t) \cup (\neg M_* \cap M_t)$, according to the definition of $\tilde{M}_{t,p}$, we have $\mathtt{MinW}(\tilde{M}_{t,p}, \hat{\boldsymbol{w}}_t) \geq \mathtt{MinW}(M_*, \hat{\boldsymbol{w}}_t)$. Then, we have

$$\mathtt{MinW}(M_t, \hat{\boldsymbol{w}}_t) - \mathtt{MinW}(\tilde{M}_{t,p}, \hat{\boldsymbol{w}}_t) \leq \mathtt{MinW}(M_t, \hat{\boldsymbol{w}}_t) - \mathtt{MinW}(M_*, \hat{\boldsymbol{w}}_t)$$

$$< \frac{1}{4}\Delta^{\mathtt{B}}_{(n+1-t)}.$$

$\qquad \square$

Now, we prove Theorem 4.

*Proof.* First, we prove that the number of samples for algorithm BSAR is bounded by $T$. Summing the number of samples for each phase, we have

$$\sum_{t=1}^{n} \tilde{T}_t \leq \sum_{t=1}^{n} \left(\frac{T-n}{\tilde{\log}(n)(n-t+1)} + 1\right)$$

$$= \frac{T-n}{\tilde{\log}(n)}\tilde{\log}(n) + n$$

$$= T.$$

Next, we prove the mistake probability. According to Lemma 14, in order to prove Theorem 4, it suffices to prove that conditioning on $\mathcal{E}$, algorithm BSAR returns $M_*$.

Assuming that $\mathcal{E}$ occurs, we prove by induction. Fix a phase $t \in [n]$. Suppose that algorithm BSAR does not make any mistake before phase $t$, i.e., $A_t \subseteq M_*$ and $R_t \cap M_* = \varnothing$. We show that algorithm BSAR does not make any mistake in phase $t$ either.

According to Lemma 16, there exists some base arm $e \in [n] \setminus (A_t \cup R_t)$ s.t. $\Delta_e^{\mathrm{B}} \geq \Delta_{(n+1-t)}^{\mathrm{B}}$ and this base arm $e$ satisfies $\mathtt{MinW}(M_t, \hat{\boldsymbol{w}}_t) - \mathtt{MinW}(\tilde{M}_{t,e}, \hat{\boldsymbol{w}}_t) > \frac{3}{4}\Delta_{(n+1-t)}^{\mathrm{B}}$. Suppose that algorithm BSAR makes a mistake in phase $t$, i.e., $p_t \in (M_* \cap \neg M_t) \cup (\neg M_* \cap M_t)$. According to Lemma 17, we have $\mathtt{MinW}(M_t, \hat{\boldsymbol{w}}_t) - \mathtt{MinW}(\tilde{M}_{t,p_t}, \hat{\boldsymbol{w}}_t) < \frac{1}{4}\Delta_{(n+1-t)}^{\mathrm{B}}$. Then,

$$
\begin{aligned}
\mathtt{MinW}(M_t, \bar{\boldsymbol{w}}_t) - \mathtt{MinW}(\tilde{M}_{t,e}, \bar{\boldsymbol{w}}_t) &> \frac{3}{4}\Delta_{(n+1-t)}^{\mathrm{B}} \\
&> \frac{1}{4}\Delta_{(n+1-t)}^{\mathrm{B}} \\
&> \mathtt{MinW}(M_t, \bar{\boldsymbol{w}}_t) - \mathtt{MinW}(\tilde{M}_{t,p_t}, \bar{\boldsymbol{w}}_t),
\end{aligned}
$$

which contradicts the selection strategy of $p_t$. Thus, $p_t \in (M_* \cap M_t) \cup (\neg M_* \cap \neg M_t)$, i.e., algorithm BSAR does not make any mistake in phase $t$, which completes the proof. $\square$

### D.3 Exponential-time Complexity of the Accept-reject Oracle used in Prior Work [11]

The accept-reject oracle used in prior CPE-L work [11], which returns the optimal super arm with a given base arm set $A_t$ contained in it, costs exponential running time on $s$-$t$ path instances. This is because the problem $\mathcal{P}$ of finding an $s$-$t$ path which contains a given edge set is NP-hard. In the following, we prove the NP-hardness of problem $\mathcal{P}$ by building a reduction from the Hamiltonian Path Problem [22] to $\mathcal{P}$.

*Proof.* Given any Hamiltonian Path instance $G = (V, E)$ with start and end nodes $s, t \in V$, we need to find an $s$-$t$ path that passes through every vertex in $V$ once ($s$-$t$ Hamiltonian path). We construct a new graph $G'$ as follows: for each vertex $u \in V \setminus \{s, t\}$, we split $u$ into two vertices $u_1, u_2$ and add an "internal" edge $(u_1, u_2)$. For each edge $(u, v) \in E$ such that $u, v \in V \setminus \{s, t\}$, we change the original $(u, v)$ to two edges $(u_1, v_2)$ and $(u_2, v_1)$. For each edge $(s, u) \in E$ such that $u \in V \setminus \{s, t\}$, we change the original $(s, u)$ to edge $(s, u_1)$. For each edge $(u, t) \in E$ such that $u \in V \setminus \{s, t\}$, we change the original $(u, t)$ to edge $(u_2, t)$.

Then, the following two statements are equivalent: (i) there exists an $s$-$t$ Hamiltonian path in $G$, and (ii) there exists an $s$-$t$ path in $G'$, which contains all internal edges $(u_1, u_2)$ for $u \in V \setminus \{s, t\}$. If there is a polynomial-time oracle to find an $s$-$t$ path which contains a given edge set, then this oracle can solve the given Hamiltonian path instance in polynomial time. However, the Hamiltonian Path Problem is NP-hard, and thus the problem of finding an $s$-$t$ path which contains a given edge set is also NP-hard. $\square$

## E  Time Complexity

In this paper, all our algorithms run in polynomial time. Since the running time of our algorithms mainly depends on their used offline procedures, here we present the time complexity of used offline procedures on three common decision classes, e.g., $s$-$t$ paths, maximum cardinality matchings and spanning trees. Let $E$ and $V$ denote the numbers of edges and vertices in the graph, respectively.

| | MaxOracle | ExistOracle | BottleneckSearch | AR-Oracle |
|---|---|---|---|---|
| $s$-$t$ paths | $O(E)$ | $O(E+V)$ | $O(E^2(E+V))$ | $O(E(E+V))$ |
| matchings | $O(V\sqrt{V}E)$ | $O(E\sqrt{V})$ | $O(E^3\sqrt{V})$ | $O(E^2\sqrt{V})$ |
| spanning trees | $O(E)$ | $O(E)$ | $O(E^3)$ | $O(E^2)$ |

Table 1: Time complexity of the offline procedures used in our algorithms.

### E.1  Reduction of ExistOracle to $2$-vertex Connectivity

In this subsection, we show how to reduce *the problem of finding a $s$-$t$ path that contains a given edge $(u, v)$ (ExistOracle)* to *the $2$-vertex connectivity problem [20]* as follows.

First, we formally define these two problems.

**Problem A** (`ExistOracle`). Given a graph $G$ with vertices $s, t, u, v$, check if there exists a $s$-$t$ simple path that contains $(u, v)$, and output such a path if it exists.

**Problem B** (2-**vertex connectivity**). Given a graph $G$ with vertices $w, z$, check if there exist two vertex-disjoint paths connecting $w$ and $z$, and output such two vertex-disjoint paths if they exist.

Now we present the proof of reduction from Problem A to Problem B.

*Proof.* The reduction starts from a given instance of Problem A. Given a graph $G$ with vertices $s, t, u, v$, we divide edge $(u, v)$ into two edges $(u, w), (w, v)$ with an added virtual vertex $w$. Similarly, we also divide edge $(s, t)$ into two edges $(s, z), (z, t)$ with an added virtual vertex $z$. Now, we show that finding a $s$-$t$ simple path that contains $(u, v)$ is equivalent to finding two vertex-disjoint paths connecting $w$ and $z$.

(i) If we have a $s$-$t$ simple path $p$ that contains $(u, v)$, then $p$ has two subpaths $p_1, p_2$ connecting $s, w$ and $w, t$, respectively, where $p_1, p_2$ do not have overlapped vertices. We concatenate $p_1$ and $(s, z)$, and concatenate $p_2$ and $(t, z)$. Then, we obtain two vertex-disjoint paths connecting $w$ and $z$.

(ii) If we have two vertex-disjoint paths connecting $w$ and $z$, then using the facts that $w$ is only connected to vertices $u, v$ and $z$ is only connected to vertices $s, t$, we can obtain two vertex-disjoint paths $q_1, q_2$ connecting $s, u$ and $t, v$, respectively (or connecting $s, v$ and $t, u$, respectively). We concatenate $q_1, q_2$ and $(u, v)$. Then, we obtain a $s$-$t$ simple path that contains $(u, v)$.

Therefore, we showed that for any given instance of Problem A, we can transform it to an instance of Problem B (by the above construction), and then use an existing oracle of Problem B [20] to solve the given instance of Problem A. □

# F    Extension to General Reward Functions

## F.1    Problem Setting

In this section, we study the extension of CPE-B to general reward functions (CPE-G) in the fixed-confidence setting. Let $f(M, \boldsymbol{w})$ denote the expected reward function of super arm $M$, which only depends on $\{w(e)\}_{e \in M}$. Different from previous CPE works [11, 10, 24] which either study the linear reward function or impose strong assumptions (continuous and separable [24]) on nonlinear reward functions, we only make the following two standard assumptions:

**Assumption 1** (Monotonicity). *For any $M \in \mathcal{M}$ and $\boldsymbol{v}, \boldsymbol{v}' \in \mathbb{R}^n$ such that $\forall e \in [n]$, $v'(e) \leq v(e)$, we have $f(M, \boldsymbol{v}') \leq f(M, \boldsymbol{v})$.*

**Assumption 2** (Lipschitz continuity with $\infty$-norm). *For any $M \in \mathcal{M}$ and $\boldsymbol{v}, \boldsymbol{v}' \in \mathbb{R}^n$, there exists a universal constant $U > 0$ such that $|f(M, \boldsymbol{v}) - f(M, \boldsymbol{v}')| \leq U \max_{e \in M} |v(e) - v'(e)|$.*

A wide family of reward functions satisfy these two mild assumptions, with the linear reward function (CPE-L) [11, 10], bottleneck reward function (CPE-B) and continuous and separable reward functions (CPE-CS) [24] as its special cases. In addition, many other interesting problems, such as the quadratic network flow [37], quadratic network allocation [23] and the densest subgraph [21], are encompassed by CPE-G.

## F.2    Algorithm for CPE-G

For CPE-G, we propose a novel algorithm `GenLUCB` as in Algorithm 8. We allow `GenLUCB` to access an efficient maximization oracle `MaxOracle` for reward function $f$ to find an optimal super arm from the given decision class and weight vector. Formally, `MaxOracle`$(\mathcal{F}, \boldsymbol{v}) \in \operatorname{argmax}_{M \in \mathcal{F}} f(M, \boldsymbol{v})$. We describe the procedure of `GenLUCB` as follows: at each timestep, we compute the lower and upper confidence bounds of the base arm rewards, and use the maximization oracle `MaxOracle` to find the super arm $M_t$ with the maximum pessimistic reward from $\mathcal{M}$ and super arm $\tilde{M}_t$ with the maximum optimistic reward from $\mathcal{M} \setminus \{M_t\}$. Then, we play the base arm $p_t$ with the maximum confidence radius from $M_t \cup \tilde{M}_t$. When we see that the pessimistic reward of $M_t$ is higher than the optimistic reward of $\tilde{M}_t$, which implies that $M_t$ has a higher reward than any other super arm with high confidence, we stop the algorithm and return $M_t$.

---
**Algorithm 8** GenLUCB
---
1: **Input:** decision class $\mathcal{M}$, confidence $\delta \in (0, 1)$, reward function $f$ and maximization oracle
   MaxOracle for $f$.
2: Initialization: play each base arm $e \in [n]$ once. Initialize empirical means $\hat{w}_{n+1}$ and set
   $T_{n+1}(e) \leftarrow 1, \forall e \in [n]$.
3: **for** $t = n + 1, n + 2, \dots$ **do**
4:    $\text{rad}_t(e) \leftarrow R\sqrt{2 \ln(\frac{4nt^3}{\delta})/T_t(e)}, \ \forall e \in [n]$
5:    $\underline{w}_t(e) \leftarrow \hat{w}_t(e) - \text{rad}_t(e), \ \forall e \in [n]$
6:    $\bar{w}_t(e) \leftarrow \hat{w}_t(e) + \text{rad}_t(e), \ \forall e \in [n]$
7:    $M_t \leftarrow \text{MaxOracle}(\mathcal{M}, \underline{w}_t)$
8:    $\tilde{M}_t \leftarrow \text{MaxOracle}(\mathcal{M} \setminus \{M_t\}, \bar{w}_t)$ or $\tilde{M}_t \leftarrow \text{MaxOracle}(\mathcal{M} \setminus \mathcal{S}(M_t), \bar{w}_t)$
9:    **if** $f(M_t, \underline{w}_t) \geq f(\tilde{M}_t, \bar{w}_t)$ **then**
10:       **return** $M_t$
11:   **end if**
12:   $p_t \leftarrow \text{argmax}_{M_t \cup \tilde{M}_t} \text{rad}_t(e)$
13:   Play base arm $p_t$ and observe the reward
14:   Update empirical means $\hat{w}_{t+1}$
15:   Update the number of pulls: $T_{t+1}(p_t) \leftarrow T_t(p_t) + 1$ and $T_{t+1}(e) \leftarrow T_t(e)$ for all $e \neq p_t$.
16: **end for**
---

Different from CPE-B or previous CPE works [11, 12] which only focus on the bottleneck base arms or those in symmetric difference, in CPE-G we select the base arm among the *entire* union set of two critical super arms, since for these two super arms, any base arm in their union can affect the reward difference and should be estimated.

### F.3  Implementation of the Oracle in GenLUCB

Now we discuss the implementation of MaxOracle in GenLUCB. For $\mathcal{F} = \mathcal{M}$, we simply calculate an optimal super arm from $\mathcal{M}$ with respect to $v$. Such a maximization oracle can be implemented efficiently for a rich class of decision classes and nonlinear reward functions, such as the densest subgraph [28], quadratic network flow problems [37] and quadratic network allocation problems [23]. For $\mathcal{F} = \mathcal{M} \setminus \{M_t\}$, it is more challenging to implement in polynomial time. We first discuss three common cases, where the step $\tilde{M}_t \leftarrow \text{MaxOracle}(\mathcal{M} \setminus \{M_t\}, \bar{w}_t)$ (labeled as (a)) can be replaced with a more practical statement $\tilde{M}_t \leftarrow \text{MaxOracle}(\mathcal{M} \setminus \mathcal{S}(M_t), \bar{w}_t)$ (labeled as (b)). Then, we can implement it as follows: repeatedly remove each base arm in $M_t$ and compute the best super arm from the remaining decision class, and then return the one with the maximum reward.

Below we formally state the three cases:

*Case (i). Any two super arms $M, M' \in \mathcal{M}$ satisfies $M \setminus M' \neq \varnothing$.*

In this case, $\mathcal{S}(M_t) = M_t$ and the statements (a),(b) are equivalent. Many decision classes such as top $k$, maximum cardinality matchings, spanning trees fall in this case.

*Case (ii). $f$ is set monotonically decreasing.*

As CPE-B, $f(M_t, \boldsymbol{w}) \geq f(M', \boldsymbol{w})$ for all $M' \in \mathcal{S}(M_t)$, and we only need to compare $M_t$ against super arms in $\mathcal{M} \setminus \mathcal{S}(M_t)$.

*Case (iii). $f$ is strictly set monotonically increasing.*

According to Line 7 of Algorithm 8, we have that $\mathcal{S}(M_t) = M_t$ and the statements (a),(b) are equivalent. Linear (CPE-L), quadratic, and continuous and separable (CPE-CS) reward functions satisfy this property when the expected rewards of base arms are non-negative.

If neither of the above cases holds, algorithm GenLUCB executes $\tilde{M}_t \leftarrow \text{MaxOracle}(\mathcal{M} \setminus \{M_t\}, \bar{w}_t)$ by disabling $M_t$ in some way and finding the best super arm from the remaining decision class with the basic maximization oracle. For the densest subgraph problem, for example, we can construct $\text{MaxOracle}(\mathcal{M} \setminus \{M_t\}, \bar{w}_t)$ efficiently by the following procedure. Given $M_t \subseteq E$, we consider the corresponding a set of vertices $S_t \subseteq V$. First, for each vertex $i \in S_t$, we remove $i \in S_t$ from $V$,

and obtain the best solution $S_i^*$ in the remaining graph by using any exact algorithms. Second, for each $j \notin S_t$, we force $\{j\} \cup S$ to be included, and obtain the best solution $S_j^*$. Then we output the best solution among them. Note that the second step can be efficiently done by an exact flow-based algorithm with a min-cut procedure [21].

### F.4 Sample Complexity of GenLUCB

Now we show the sample complexity of GenLUCB for CPE-G. For any $e \notin M_*$, let $\Delta_e^G = f(M_*, \boldsymbol{w}) - \max_{M \in \mathcal{M}: e \in M} f(M, \boldsymbol{w})$, and for any $e \in M_*$, let $\Delta_e^G = f(M_*, \boldsymbol{w}) - \max_{M \neq M_*} f(M, \boldsymbol{w}) = \Delta_{\min}$. We formally state the sample complexity of GenLUCB in Theorem 10.

**Theorem 10.** *With probability at least $1 - \delta$, the* GenLUCB *algorithm for CPE-G will return the optimal super arm with sample complexity*

$$
O\left(\sum_{e \in [n]} \frac{R^2 U^2}{(\Delta_e^G)^2} \ln\left(\sum_{e \in [n]} \frac{R^2 U^2 n}{(\Delta_e^G)^2 \delta}\right)\right).
$$

Compared to the uniform sampling algorithm (presented in Appendix G) which has the $O(\frac{R^2 U^2 n}{\Delta_{\min}^2} \ln(\frac{R^2 U^2 n}{\Delta_{\min}^2 \delta}))$ sample complexity, GenLUCB achieves a much tighter result owing to the adaptive sample strategy, which validates its effectiveness. Moreover, to our best knowledge, GenLUCB is the first algorithm with non-trivial sample complexity for CPE with general reward functions, which encompass a rich class of nonlinear combinatorial problems, such as the densest subgraph problem [21], quadratic network flow problem [37] and quadratic network allocation problem [23].

To prove the sample complexity of algorithm GenLUCB (Theorem 10), we first introduce the following Lemmas 18,19.

**Lemma 18** (Correctness of GenLUCB). *Assume that event $\xi$ occurs. Then, if algorithm* GenLUCB *(Algorithm 8) terminates at round t, we have $M_t = M_*$.*

*Proof.* According to the stop condition (Line 9 of Algorithm 8), when algorithm GenLUCB (Algorithm 8) terminates at round $t$, we have that for any $M \neq M_t$,

$$
f(M_t, \boldsymbol{w}) \geq f(M_t, \underline{\boldsymbol{w}}_t) \geq f(M, \bar{\boldsymbol{w}}_t) \geq f(M, \boldsymbol{w}).
$$

Thus, we have $M_t = M_*$. $\qquad\square$

**Lemma 19.** *Assume that event $\xi$ occurs. For any $e \in [n]$, if $\mathrm{rad}_t(e) < \frac{\Delta_e^G}{4U}$, then, base arm $e$ will not be pulled at round t, i.e., $p_t \neq e$.*

*Proof.* (i) Suppose that for some $e \notin M_*$, $\mathrm{rad}_t(e) < \frac{\Delta_e^G}{4U} = \frac{1}{4U}(f(M_*, \boldsymbol{w}) - \max_{M \in \mathcal{M}: e \in M} f(M, \boldsymbol{w}))$ and $p_t = e$. According to the selection strategy of $p_t$, we have that for any $e' \in M_t \cup \tilde{M}_t$, $\mathrm{rad}_t(e') \leq \mathrm{rad}_t(e) < \frac{\Delta_e^G}{4U}$.

First, we can prove that $M_t \neq M_*$ and $\tilde{M}_t \neq M_*$. Otherwise, one of $M_t, \tilde{M}_t$ is $M_*$ and the other is a sub-optimal super arm containing $e$, which is denoted by $M'$. Then,

$$
\begin{aligned}
f(M_*, \underline{\boldsymbol{w}}_t) - f(M', \bar{\boldsymbol{w}}_t) &\geq (f(M_*, \boldsymbol{w}) - 2U \max_{i \in M_*} \mathrm{rad}_i) - (f(M', \boldsymbol{w}) + 2U \max_{j \in M'} \mathrm{rad}_j) \\
&> \Delta_{M_*, M'}^G - \frac{\Delta_e^G}{2} - \frac{\Delta_e^G}{2} \\
&= 0,
\end{aligned}
$$

which gives a contradiction.

Then, if $e \in \tilde{M}_t$, we have

$$
\begin{aligned}
f(\tilde{M}_t, \bar{\boldsymbol{w}}_t) &\leq f(\tilde{M}_t, \boldsymbol{w}) + 2U \max_{i \in \tilde{M}_t} \mathrm{rad}_i \\
&< f(\tilde{M}_t, \boldsymbol{w}) + \frac{\Delta_e^G}{2}
\end{aligned}
$$

$$< f(M_*, \boldsymbol{w})$$
$$\leq f(M_*, \bar{\boldsymbol{w}}_t),$$

which contradicts the definition of $\tilde{M}_t$.

If $e \in M_t$, we have

$$f(\tilde{M}_t, \bar{\boldsymbol{w}}_t) - f(\tilde{M}_t, \underline{\boldsymbol{w}}_t) \geq f(M_*, \bar{\boldsymbol{w}}_t) - f(M_t, \underline{\boldsymbol{w}}_t)$$
$$\geq f(M_*, \boldsymbol{w}) - f(M_t, \boldsymbol{w})$$
$$= \Delta^G_{M_*, M_t}.$$

On the other hand, we have

$$f(\tilde{M}_t, \bar{\boldsymbol{w}}_t) - f(\tilde{M}_t, \underline{\boldsymbol{w}}_t) \leq (f(\tilde{M}_t, \hat{\boldsymbol{w}}) + U \max_{i \in \tilde{M}_t} \mathrm{rad}_i) - (f(\tilde{M}_t, \hat{\boldsymbol{w}}) - U \max_{i \in \tilde{M}_t} \mathrm{rad}_i)$$
$$= 2U \max_{i \in \tilde{M}_t} \mathrm{rad}_i$$
$$< \frac{\Delta^G_e}{2}$$
$$\leq \frac{\Delta^G_{M_*, M_t}}{2}$$
$$< \Delta^G_{M_*, M_t},$$

which gives a contradiction.

(ii) Suppose that for some $e \in M_*$, $\mathrm{rad}_t(e) < \frac{\Delta^G_e}{4U} = \frac{\Delta_{\min}}{4U}$ and $p_t = e$. According to the selection strategy of $p_t$, we have that for any $e' \in M_t \cup \tilde{M}_t$, $\mathrm{rad}_t(e') \leq \mathrm{rad}_t(e) < \frac{\Delta_{\min}}{4U}$.

First, we can prove that $M_t \neq M_*$ and $\tilde{M}_t \neq M_*$. Otherwise, one of $M_t, \tilde{M}_t$ is $M_*$ and the other is a sub-optimal super arm, which is denoted by $M'$. Then,

$$f(M_*, \underline{\boldsymbol{w}}_t) - f(M', \bar{\boldsymbol{w}}_t) \geq (f(M_*, \boldsymbol{w}) - 2U \max_{i \in M_*} \mathrm{rad}_i) - (f(M', \boldsymbol{w}) + 2U \max_{j \in M'} \mathrm{rad}_j)$$

$$> \Delta^G_{M_*, M'} - \frac{\Delta_{\min}}{2} - \frac{\Delta_{\min}}{2}$$
$$= 0,$$

which gives a contradiction.

Thus, both $M_t$ and $\tilde{M}_t$ are sub-optimal super arms.

However, on the other hand, we have

$$f(\tilde{M}_t, \bar{\boldsymbol{w}}_t) \leq f(\tilde{M}_t, \boldsymbol{w}) + 2U \max_{i \in \tilde{M}_t} \mathrm{rad}_i$$
$$< f(\tilde{M}_t, \boldsymbol{w}) + \frac{\Delta_{\min}}{2}$$
$$< f(M_*, \boldsymbol{w})$$
$$\leq f(M_*, \bar{\boldsymbol{w}}_t),$$

which contradicts the definition of $\tilde{M}_t$. □

Now, we prove Theorem 10.

*Proof.* For any $e \in [n]$, let $T(e)$ denote the number of samples for base arm $e$, and $t_e$ denote the last timestep at which $e$ is pulled. Then, we have $T_{t_e} = T(e) - 1$. Let $T$ denote the total number of samples. According to Lemma 19, we have

$$R\sqrt{\frac{2 \ln(\frac{4nt_e^3}{\delta})}{T(e) - 1}} \geq \frac{\Delta^G_e}{4U}$$

---

**Algorithm 9** UniformFC

---

1: **Input:** decision class $\mathcal{M}$, confidence $\delta \in (0,1)$, reward function $f$ and maximization oracle MaxOracle for $f$.
2: **for** $t = 1, 2, \ldots$ **do**
3:   For each base arm $e \in [n]$, pull $e$ once and then update its empirical mean $\hat{w}_t(e)$ and number of samples $T_t(e)$
4:   $\mathrm{rad}_t \leftarrow R\sqrt{2\ln(\frac{4nt^3}{\delta})/t}$
5:   $\underline{w}_t(e) \leftarrow \hat{w}_t(e) - \mathrm{rad}_t, \ \forall e \in [n]$
6:   $\bar{w}_t(e) \leftarrow \hat{w}_t(e) + \mathrm{rad}_t, \ \forall e \in [n]$
7:   $M_t \leftarrow \mathtt{MaxOracle}(\mathcal{M}, \underline{w}_t)$
8:   $\tilde{M}_t \leftarrow \mathtt{MaxOracle}(\mathcal{M} \setminus \mathcal{S}(M_t), \bar{w}_t)$
9:   **if** $f(M_t, \underline{w}_t) \geq f(\tilde{M}_t, \bar{w}_t)$ **then**
10:     **return** $M_t$
11:   **end if**
12: **end for**

---

Thus, we obtain

$$T(e) \leq \frac{32R^2U^2}{(\Delta_e^G)^2} \ln\left(\frac{4nt_e^3}{\delta}\right) + 1 \leq \frac{32R^2U^2}{(\Delta_e^G)^2} \ln\left(\frac{4nT^3}{\delta}\right) + 1$$

Summing over $e \in [n]$, we have

$$T \leq \sum_{e \in [n]} \frac{32R^2U^2}{(\Delta_e^G)^2} \ln\left(\frac{4nT^3}{\delta}\right) + n \leq \sum_{e \in [n]} \frac{96R^2U^2}{(\Delta_e^G)^2} \ln\left(\frac{2nT}{\delta}\right) + n,$$

where $\sum_{e \in [n]} \frac{R^2U^2}{(\Delta_e^G)^2} \geq n$. Then, applying Lemma 20, we have

$$T \leq \sum_{e \in [n]} \frac{576R^2U^2}{(\Delta_e^G)^2} \ln\left(\frac{2n^2}{\delta} \sum_{e \in [n]} \frac{96R^2U^2}{(\Delta_e^G)^2}\right) + n$$

$$= O\left(\sum_{e \in [n]} \frac{R^2U^2}{(\Delta_e^G)^2} \ln\left(\sum_{e \in [n]} \frac{R^2U^2n^2}{(\Delta_e^G)^2\delta}\right) + n\right)$$

$$= O\left(\sum_{e \in [n]} \frac{R^2U^2}{(\Delta_e^G)^2} \ln\left(\sum_{e \in [n]} \frac{R^2U^2n}{(\Delta_e^G)^2\delta}\right)\right).$$

$\square$

## G   Uniform Sampling Algorithms

In this section, we present the uniform sampling algorithms for the fixed-confidence and fixed-budget CPE problems.

Algorithm 9 illustrates the uniform sampling algorithm UniformFC for the fixed-confidence CPE problem. Below we state the sample complexity of algorithm UniformFC.

**Theorem 11.** *With probability at least $1 - \delta$, the* UniformFC *algorithm (Algorithm 9) will return the optimal super arm with sample complexity*

$$O\left(\frac{R^2U^2n}{\Delta_{\min}^2} \ln\left(\frac{R^2U^2n}{\Delta_{\min}^2\delta}\right)\right).$$

*Proof.* Let $\Delta_{\min} = \min_{e \in [n]} \Delta_e^G = f(M_*, \boldsymbol{w}) - f(M_{\mathrm{second}}, \boldsymbol{w})$. Assume that event $\xi$ occurs. Then, if $\mathrm{rad}_t < \frac{\Delta_{\min}}{4U}$, algorithm UniformFC will stop. Otherwise,

$$f(M_t, \underline{\boldsymbol{w}}_t) - f(\tilde{M}_t, \bar{\boldsymbol{w}}_t) \geq f(M_t, \boldsymbol{w}) - 2U \max_{i \in M_t} \mathrm{rad}_t - (f(\tilde{M}_t, \boldsymbol{w}) + 2U \max_{i \in \tilde{M}_t} \mathrm{rad}_t)$$

---

**Algorithm 10** UniformFB

---

1: **Input:** $\mathcal{M}$, budget $T$, reward function $f$ and maximization oracle MaxOracle for $f$.
2: Pull each base arm $e \in [n]$ for $\lfloor T/n \rfloor$ times
3: Update the empirical means $\hat{\boldsymbol{w}}_t$
4: $M_{\text{out}} \leftarrow$ MaxOracle$(\mathcal{M}, \hat{\boldsymbol{w}}_t)$
5: **return** $M_{\text{out}}$

---

$$
\begin{aligned}
=&f(M_t, \boldsymbol{w}) - f(\tilde{M}_t, \boldsymbol{w}) - 4U\text{rad}_t \\
>&\Delta_{\min} - \Delta_{\min} \\
=&0,
\end{aligned}
$$

which contradicts the stop condition.

Let $T_n$ denote the number of rounds and $T$ denote the total number of samples. Then, we have

$$
R\sqrt{\frac{2\ln(\frac{4nT_n^3}{\delta})}{T_n - 1}} \geq \frac{\Delta_{\min}}{4U}
$$

Thus, we obtain

$$
T_n \leq \frac{32R^2U^2}{\Delta_{\min}^2}\ln\left(\frac{4nT_n^3}{\delta}\right) + 1 \leq \frac{96R^2U^2}{\Delta_{\min}^2}\ln\left(\frac{2nT_n}{\delta}\right) + 1.
$$

Then, applying Lemma 20, we have

$$
\begin{aligned}
T_n \leq &\frac{576R^2U^2}{\Delta_{\min}^2}\ln\left(\frac{2n}{\delta}\frac{96R^2U^2}{\Delta_{\min}^2}\right) + 1 \\
=&O\left(\frac{R^2U^2}{\Delta_{\min}^2}\ln\left(\frac{R^2U^2n}{\Delta_{\min}^2\delta}\right)\right).
\end{aligned}
$$

Summing over the number of samples for all the base arms, we obtain

$$
T = O\left(\frac{R^2U^2n}{\Delta_{\min}^2}\ln\left(\frac{R^2U^2n}{\Delta_{\min}^2\delta}\right)\right).
$$

$\square$

Algorithm 10 illustrates the uniform sampling algorithm UniformFB for the fixed-budget CPE problem. Below we state the error probability of algorithm UniformFB.

Let $H^U = n(\Delta_{\min})^{-2}$, where $\Delta_{\min} = f(M_*, \boldsymbol{w}) - f(M_{\text{second}}, \boldsymbol{w})$.

**Theorem 12.** *For any $T > n$, algorithm* UniformFB *uses at most $T$ samples and returns the optimal super arm with the error probability bounded by*

$$
O\left(n\exp\left(-\frac{T}{R^2U^2H^U}\right)\right).
$$

*Proof.* Since algorithm UniformFB allocates $\lfloor T/n \rfloor$ samples to each base arm $e \in [n]$, the total number of samples is at most $T$.

Now we prove the error probablity. Define event

$$
\mathcal{G} = \left\{\forall i \in [n], \ |\hat{w}_t(i) - w(i)| < \frac{\Delta_{\min}}{4U}\right\}.
$$

According to the Hoeffding's inequality,

$$
\left\{|\hat{w}_t(i) - w(i)| \geq \frac{\Delta_{\min}}{4U}\right\} \leq 2\exp\left(-\frac{T\Delta_{\min}^2}{32R^2U^2n}\right).
$$

By a union bound over $e \in [n]$, we have

$$\Pr[\mathcal{G}] \geq 1 - 2n \exp\left(-\frac{T\Delta_{\min}^2}{32R^2U^2n}\right)$$

$$= 1 - 2n \exp\left(-\frac{T}{32R^2U^2H^U}\right)$$

Below we prove that conditioning on event $\mathcal{G}$, $M_{\text{out}} = M_*$. Suppose that $M_{\text{out}} \neq M_*$,

$$f(M_{\text{out}}, \hat{\boldsymbol{w}}_t) - f(M_*, \hat{\boldsymbol{w}}_t) \leq f(M_{\text{out}}, \boldsymbol{w}) + \frac{\Delta_{\min}}{4} - \left(f(M_*, \boldsymbol{w}) - \frac{\Delta_{\min}}{4}\right)$$

$$\leq -\Delta_{\min} + \frac{\Delta_{\min}}{2}$$

$$= -\frac{\Delta_{\min}}{2}$$

$$< 0,$$

which contradicts the selection strategy of $M_{\text{out}}$. Thus, conditioning on event $\mathcal{G}$, algorithm `UniformFB` returns $M_*$. Then, we obtain Theorem 12. $\qquad\square$

## H  Technical Tool

In this section, we present a technical tool used in the proofs of our results.

**Lemma 20.** *If $T \leq c_1 \ln(c_2 T) + c_3$ holds for some constants $c_1, c_2, c_3 \geq 1$ such that $\ln(c_1 c_2 c_3) \geq 1$, we have $T \leq 6c_1 \ln(c_1 c_2 c_3) + c_3$.*

*Proof.* In the inequality $T \leq c_1 \ln(c_2 T) + c_3$, the LHS is linear with respect to $T$ and the RHS is logarithmic with respect to $T$. Thus, we have $T > c_1 \ln(c_2 T) + c_3$ for a big enough $T$. Then, to prove $T \leq T_0 \triangleq 6c_1 \ln(c_1 c_2 c_3) + c_3$, it suffices prove that $T_0 > c_1 \ln(c_2 T_0) + c_3$. Since

$$c_1 \ln(c_2 T_0) + c_3 = c_1 \ln(c_2(6c_1 \ln(c_1 c_2 c_3) + c_3)) + c_3$$

$$= c_1 \ln(6c_1 c_2 \ln(c_1 c_2 c_3) + c_2 c_3) + c_3$$

$$\leq c_1 \ln(6c_1^2 c_2^2 c_3 + c_2 c_3) + c_3$$

$$\leq c_1 \ln(7c_1^2 c_2^2 c_3) + c_3$$

$$\leq 2c_1 \ln(7c_1 c_2 c_3) + c_3$$

$$= 2c_1 \ln(c_1 c_2 c_3) + 2c_1 \ln(7) + c_3$$

$$\leq 2c_1 \ln(c_1 c_2 c_3) + 2\ln(7)c_1 \ln(c_1 c_2 c_3) + c_3$$

$$= (2 + 2\ln(7))c_1 \ln(c_1 c_2 c_3) + c_3$$

$$\leq 6c_1 \ln(c_1 c_2 c_3) + c_3$$

$$= T_0,$$

we obtain the lemma. $\qquad\square$