# OpenReview forum: "Combinatorial Pure Exploration with Bottleneck Reward Function"
_NeurIPS.cc/2021/Conference — NeurIPS 2021 Poster_

### Official Review · Reviewer_minG · 2021-07-06

**Rating:** 6
**Confidence:** 4

**Summary:**

The authors focus on the Combinatorial Pure Exploration problem with bottleneck rewards (CPE-B) and consider two settings: fixed-confidence (FC) and fixed-budget (FB). They argue that the existing CPE algorithms cannot be adapted to solve the challenges arisen in CPE-B.  Then, for the FC setting, the paper proposes two algorithms, called BLUCB and BLUCB-Parallel, that provides optimal (within a logarithmic factor) sample-complexity for several important instances. Moreover, a lower-bound is also deduced that matches the guarantee of BLUCB-Parallel. For the FB setting, the authors provide an algorithm, called BSAR, that achieves the same error probability guarantee as that of the state-of-the-art work. The authors also claim that BLUCB-Parallel and BSAR are efficiently implementable in several important instances such as $s$-$t$ path, maximum cardinality matchings, spanning trees, etc.

**Ethical Concerns:**

No ethical issues with this paper.

**Limitations And Societal Impact:**

The authors adequately addressed the limitations and potential negative societal impact of their work.

**Main Review:**

+ Originality: The model is relatively new and well-motivated: although combinatorial pure exploration problem has been studied before, none of the previous works focuses on the case with bottleneck reward functions. Concerning the involved techniques, the main algorithms proposed in this paper are (non-trivial) adaptations of previous known approaches/frameworks (i.e., explore-verify-parallel scheme in the FC setting and accept-reject scheme in the FB setting) into the context of CPE-B. Bottleneck reward function poses novel challenges and to overcome them, the authors provide several efficient oracles (such as Bottlenecksearch, ExistOracle, AR-Oracle) specifically designed for the bottleneck setting. In my opinion, these oracles are novel and they seem to be the contributions that stand out the most in this paper. Unfortunately, these oracles are not shown/discussed very clearly (due to the organization and writing flows of the paper). Most of related work is adequately cited (except for the literature on the guarantees’ lower-bounds in the FC setting).

+ Quality: The main results on the performance guarantees of BLUCB, BLUCB-Parallel and BSAR algorithms seem technically sound: I do not spot any obvious error (note that I did not check through all details in the proofs). Several numerical experiments are conducted but it lacks adequate discussions. The paper also claims that the proposed algorithms are efficiently implementable. However, this claim is not shown clearly, especially on the efficiency of the oracles (see below). The results in the FC setting feel quite complete; on the other hand, the ones in the FB setting is less so: BSAR does not provide significant improvements in comparison to the benchmark CSAR and no lower-bound is available.

+ Clarity: The texts itself is quite well-written. Overall, the technical discussions are relatively adequate for readers to follow. Nevertheless, the organization of the paper should be revised. Particularly, in my opinion, the oracles designed for Bottleneck reward are the novelties that really differ this work to the literature and should be analyzed with details in the main text (currently in the appendix). To do this, probably the authors should find a way to trim down the main algorithms given that the explore-verify-parallel and accept-reject schemes are previously known.

As mentioned above, my main concern is the claim on efficiency. Particularly, the proof on the running time of ExistOracle (Appendix C.2.1) is vague and rushed toward conclusions (this is the core result for the proofs on Bottlenecksearch, AR-Oracle and eventually, the running time of the algorithms). How is $s$-$t$ path reduced to 2-vertex connectivity problem? When no super arm in $M_{\geq w(e)}$ contains the base arm $e$, how can one check this efficiently, given that $M_{\geq w(e)}$ might contain exponentially many elements? Maybe I misunderstood something here.

+ Significance: The paper provides the (first-ever) results for the CPE-B framework. They can be used as the foundation for other analyses on CPE-B and probably can be generalized to cases different from bottleneck rewards.

+ Other comments and questions:

1/ Page 7: BLUCB-parallel is better than BLUCB when $\delta \le \exp(-H_{E}/(H_{E}-H_{V}))$. Therefore, is it true that before running the algorithm, we cannot compute (efficiently) $\Delta^C_e$ and thus, no way to know if this condition on $\delta$ will hold? Neither can we tune a priori this $\delta$ in BLUCB-parallel?
2/ Page 5: line 4 of Algorithm 2, does this mean that ONLY when $t$ is an even number, we execute Verify_k? Why should we do this and is this a “wasteful” way of using odd iterations $t$?
3/ Line 293: Which CPE-L algorithm of [10] is discussed here? CSAR?
4/ Line 311: This sentence is very confusing. Result of BSAR in Theorem 4 matches with the bound of CSAR presented in Theorem 3 of [10] on the linear case; do the authors imply that even in the bottleneck case, CSAR will have the same guarantee? What is precisely the novelty of BSAR?
5/ Line 313: if $\Delta^B_i$ is the same order with $\Delta_{\textrm{min}}$, $H^B$ is also $\mathcal{O}(n/\Delta_{\textrm{min}})$; given this, how do the authors claim that BSAR’s guarantee is “significantly” better than that of the uniform sampling algorithm?
6/ Page 9: In Figure 2(a), it looks like the sample complexity of BLUCB and BLUCB-Parallel have a better order (in terms of $n$) than that of CLUCB. In Figure 2(b), this difference (in order) is much smaller. Why does this happen (the theoretical guarantees of CLUCB (in [10]) and BLUCB, BLUCB-Parallel have the same order in terms of $n$)? In Figure 2c and 2d, when the gap is large, CSAR can outperform BSAR; do we have a condition for when this will happen (based on the precise bound provide by [10])?
Can the authors also report the elapsed time of these algorithms in the experiments?

+ Some (possible) typos: line 137: denote  -> denotes; line 180” bottlenck -> bottleneck;  line 193: propose algorithm BLUCB-parallel -> propose the BLUCB-parallel algorithm;  Line 285: avoid them disturbing -> avoid disturbing


**Time Spent Reviewing:**

10

---

> ### Author Response · Authors · 2021-08-10
> **Response to Reviewer minG - Part 1/2**
>
> Thank you very much for your careful reviews and valuable suggestions.
>
> **Replies to Main Reviews:**
>
> 1 (Reduction for ExistOracle)
>
> (1) We show how to build a reduction from *"the problem of finding a $s$-$t$ path that contains a given edge $(u,v)$ (ExistOracle)"* to *"the 2-vertex connectivity problem [19]"* as follows.
>
> **Proof of Reduction.**
> First, we formally define these two problems.
>
> *Problem A (ExistOracle).* Given a graph $G$ with vertices $s, t, u, v$, check if there exists a $s$-$t$ simple path that contains $(u,v)$, and output such a path if it exists.
>
> *Problem B (2-vertex connectivity).* Given a graph $G$ with vertices $w, z$, check if there exist two vertex-disjoint paths connecting $w$ and $z$, and output such two vertex-disjoint paths if they exist.
>
> The reduction starts from a given instance of Problem A. Given a graph $G$ with vertices $s, t, u, v$, we divide edge $(u,v)$ into two edges $(u,w), (w,v)$ with an added virtual vertex $w$. Similarly, we also divide edge $(s,t)$ into two edges $(s,z),(z,t)$ with an added virtual vertex $z$.
> Now, we show that finding a $s$-$t$ simple path that contains $(u,v)$ is equivalent to finding two vertex-disjoint paths connecting $w$ and $z$.
>
> (i) If we have a $s$-$t$ simple path $p$ that contains $(u,v)$, then $p$ has two subpaths $p_1, p_2$ connecting $s,w$ and $w,t$, respectively, where $p_1, p_2$ do not have overlapped vertices. We concatenate $p_1$ and $(s,z)$, and concatenate $p_2$ and $(t,z)$. Then, we obtain two vertex-disjoint paths connecting $w$ and $z$.
>
> (ii) If we have two vertex-disjoint paths connecting $w$ and $z$, then using the facts that $w$ is only connected to vertices $u,v$ and $z$ is only connected to vertices $s,t$, we can obtain two vertex-disjoint paths $q_1, q_2$ connecting $s,u$ and $t,v$, respectively (or connecting $s,v$ and $t,u$, respectively). We concatenate $q_1, q_2$ and $(u,v)$. Then, we obtain a $s$-$t$ simple path that contains $(u,v)$.
>
> Therefore, we showed that for any given instance of Problem A, we can transform it to an instance of Problem B (by the above construction), and then use an existing oracle of Problem B [19] to solve the given instance of Problem A. $\square$
>
> (2) When no super arm in $\mathcal{M}_{\geq w(e)}$ contains base arm e (for $s$-$t$ simple path instances), we can still efficiently check this by checking the 2-vertex connectivity [19] according to the reduction in Reply (1).
>
> Specifically, in the construction of reduction in Reply (1), "there is no $s$-$t$ simple path that contains $e=(u,v)$" is equivalent to "there are no two vertex-disjoint paths connecting $w$ and $z$".
>
> We will certainly incorporate a formal proof of the reduction, and add more discussion on the efficiency of oracles in our revised paper.
>
> 2 (Novelty of BSAR)
>
> We remark that BSAR adopts innovative unique techniques to handle the bottleneck identification task, and its error probability analysis is also different from CSAR [10]. The theoretical analysis of CSAR [10] only works for the linear reward function and cannot be applied to our bottleneck reward problem.
>
> The novelty of BSAR mainly includes:
>
> (1) BSAR adopts a special acceptance scheme for bottleneck rewards, i.e., setting the estimated rewards of the accepted arms to infinity in order to avoid them to disturb the estimation of bottleneck reward. This acceptance scheme guarantees the algorithm correctness for the bottleneck identification task. (Naively applying CSAR's accept-reject scheme cannot guarantee the correctness.)
>
> (2) BSAR uses an efficient accept-reject oracle that only needs to work a single accepted arm (instead of the whole accepted arm set as in CSAR [10]), which enables it to firstly achieve polynomial time complexity on the fixed-budget s-t path instances among existing CPE algorithms.
>
> (3) The error probability analysis of BSAR is also unique for the bottleneck reward function. In analysis, we take advantage of the used special acceptance scheme to exclude all accepted arms in the calculation of bottleneck rewards, which effectively avoids the perturbation from the inaccurate estimation of accepted arms and guarantees the correctness.
>
> To our best knowledge, there is no existing lower bound result for the fixed-budget setting in the CPE literature [9,10,13,15,23]. We agree with the reviewer that this is an interesting and important research problem, which will be a subject of our future work.
>
> 3 (Organization of Paper)
>
> Thank you so much for your positive comments on our designed oracles. Following your suggestions, we will revise the organization of our paper and analyze the oracles in the main text.
>
> **Replies to Other Comments:**
>
> 1 (Tune $\delta$, in Page 7)
>
> Indeed, if we do not have any prior knowledge on $\Delta_e^C$ ($H_E$ and $H_V$), we cannot know if this condition holds or tune a good $\delta$ for BLUCB-Parallel.
> However, if we have some prior knowledge about what degrees the exploration task is harder than the verification task, e.g., $\frac{H_E}{H_V}$, we can check the condition to determine if BLUCB-Parallel is better than BLUCB for a given $\delta$, and we can also tune a good $\delta$ for BLUCB-Parallel.
>
> 2 (Algorithm 2, in Page 5)
>
> In BLUCB-Parallel (Algorithm 2), for each timestep t, we execute at least one sub-algorithm BLUCB-Verify_k, and do not waste the odd iterations t.
>
> Note that the index of BLUCB-Verify $k \in \mathbb{N}$ starts from $0$, since $\mathbb{N}$ contains $0$.
> Thus, for odd timestep $t$, BLUCB-Verify_0 is executed, which does not waste the odd iterations.
> We will revise $k \in \mathbb{N}$ to $k \geq 0, k \in Z$ in our revised paper to avoid confusion.
>
> 3 (The discussed CPE-L Algorithm in [10], in Line 293)
>
> Yes, in Line 293, the compared "existing CPE algorithm [10]" refers to the fixed-budget algorithm CSAR in [10]. We will clarify it in our revision.
>
> 4 (Results and Novelty of BSAR, in Line 311)
>
> In Line 311, we meant that BSAR has the same form of error probability bound as CSAR [10]. Indeed, due to the different settings of reward functions, this sentence is confused and not meaningful.
> We will delete this sentence in our revision.
>
> Please note that CSAR [10] cannot have correctness guarantee (with the analysis in [10]) under our bottleneck reward function, since the theoretical analysis of CSAR only holds for the linear reward function (even if one changes its linear reward calculation to the bottleneck reward calculation).
>
> The novelty of BSAR is presented in our Reply 2 to your main reviews.

---

> > ### Author Response · Authors · 2021-08-11
> > **Response to Reviewer minG - Part 2/2**
> >
> > 5 ("Significantly Better", in Line 313)
> >
> > Thank you for raising this problem.
> > The "significantly better" is for the instances where $\Delta^B_e$ is larger than $\Delta_{\min}$ for most base arms $e \in [n]$.
> > We will revise the sentence to "achieves a better guarantee when $\Delta^B_e>\Delta_{\min}$ for most $e \in [n]$" to make it more accurate.
> >
> > For the instance that you raised, i.e., $\Delta^B_e$ is of the same order as $\Delta_{\min}$ for all $e \in [n]$, BSAR has the same order of error probability as the uniform sampling algorithm. However, please note that this phenomenon also exists in many other CPE works, e.g., [9,10,13,15,23].
> > For general CPE instances, the reward gaps of base arms are usually different (have different orders from $\Delta_{\text{min}}$). The CPE literature [9,10,13,15,23] (including our work) mainly aims to design better algorithms than uniform sampling for these general cases.
> >
> >
> > 6 (Experiments, in Page 9)
> >
> > (1) (Figure 2(a), 2(b))
> >
> > In the sample complexity bounds of CLUCB [10], BLUCB and BLUCB-Parallel, the dependance on n also involves the reward gap of base arm $\Delta^C_e$ for each $e \in [n]$. Since our BLUCB and BLUCB-Parallel enjoy tighter definition of reward gaps than CLUCB, when the number of base arms that have tighter reward gap definition increases, BLUCB and BLUCB-Parallel can have better performance than CLUCB.
> >
> > In Figure 2(b), this difference looks "smaller" since the Y-ticks in Figure 2(b) are wider than those in Figure 2(a) (which is automatically adjusted by the drawing toolkit of Python to fit the figure size). If aligning the Y-ticks in Figure 2(b) with those in Figure 2(a), the curve trends in Figure 2(b) are similar to Figure 2(a).
> > We will certainly align up the Y-ticks in our revision to avoid confusion.
> >
> >
> > (2) (Figures 2(c),2(d))
> >
> > In Figures 2(c),2(d), when the gap is large enough, the error probabilities of most algorithms are almost close to 0, since in this case the problem hardness reduces and the error probability bound decreases exponentially. These points (which are close to 0 and close to each other)  cannot fully reflect the performance gaps among the compared algorithms.
> >
> > (Again,) Please note that the theoretical bounds/analysis provided by [10] does not hold under the bottleneck reward function, even if one changes the linear reward function to the bottleneck reward function in its analysis.
> >
> > (3) (Elapsed Time)
> >
> > Thank you for this valuable suggestion. We will certainly incorporate the running time to our experiments in our revision. Below we present the average running time across 50 and 3000 independent runs for fixed-confidence and fixed-budget settings, respectively.
> >
> > For the fixed-confidence setting, on the path instance with $\delta$=0.005, $\Delta_{\text{min}}$=0.5, n=56,
> > our BLUCB and BLUCB-Parallel cost 1.52s and 4.74s running time, respectively; The ablation variant BLUCB- spends 4.78s; CLUCB-B [10] has 5.31s running time; UniformFC costs 9.59s running time.
> >
> > For the fixed-budget setting, on the path instance with T=10000, $\Delta_{\text{min}}$=0.02 and n=30,
> > our BSAR costs 2.43s running time; The ablation variant BSR spends 2.31s; CSAR-B [10] has 2.48s running time; CUCB-B [12] costs 36.73s; UniformFB spends 0.02s running time.
> >
> > (CLUCB-B [10], CSAR-B [10] and CUCB-B [12] are run with the bottleneck reward calculation for fair comparison.
> > UniformFB is fast, since it simply draws T/n samples on each base arm, which can be implemented by a single step in code, and outputs the empirical best super arm.)
> >
> >
> > 7 (Typo)
> > Thank you for pointing out these typos, we will correct them in our revision.

---

> > > ### Comment · Reviewer_minG · 2021-09-02
> > > **Post-rebuttal assessment**
> > >
> > > I give my thanks to the authors for their responses. The issues raised in the initial review are addressed. I trust that the authors will make the necessary modifications. I would like to keep my original score of 6, i.e., tending toward acceptance.

---

> > > > ### Author Response · Authors · 2021-09-02
> > > > **Response to the Post-rebuttal Comment of Reviewer minG**
> > > >
> > > > Thank you very much for your time and effort in reviewing our paper! We will certainly revise our paper according to the reviewers' comments.

---

### Official Review · Reviewer_69qt · 2021-07-17

**Rating:** 6
**Confidence:** 3

**Summary:**

The paper studies the extension of a combinatorial bandits problem studied by S. Chen et al, NeurIPS 2014. The extension here is to consider  (expected) reward as the minimum reward of the elements of a set rather than a linear combination as studied in Chen et al.

We are given a set of super-arms (which may overlap). The reward function associated with each super-arm is the minimum expected value of the reward associated with each of the base-arms in that super-arm. The problem is to find the best super-arm, the one with the maximal expected reward.

The paper provides algorithms, and upper and lower bounds on sample complexity for a given confidence bound, and error probability bound for a given budget.

The paper (including the supplemental material) is densely written with sparse discussion of the results. A table of notations would considerably help the reader.

Reported simulation results are limited.

Some of the motivating examples need better articulation (e.g. bottleneck path problem).

**Limitations And Societal Impact:**


 The paper does not discuss limitations / potential negative societal impact.

**Main Review:**

The paper provides an extension of [10], extending the results from linear rewards considered in [10] to bottleneck or minimum rewards. It  provides algorithms to find the bottleneck arm and corresponding super-arm(s), upper and lower bounds on sample size, under fixed confidence level, and error probability under fixed budget.

The suggested use-case of bottleneck shortest path needs further clarification. The shortest path, for example in a communications network, is chosen so as to transfer information from a source node to a destination node; what does sampling a base arm mean in this context? Given a collection of S-D paths, it makes sense to sample the paths (or super-arms); during the exploration phase, this will achieve the goal of information transfer (even if it is inefficient). Sampling base arms is wasteful in this context. Further this implies that the routing must be global rather than distributed which is the practical case in a large network.  In theory, the number of paths may be exponential in the network size; but in practice, many constraints are impose which limit the number of paths, and hence the number of super-arms to be considered.

The experimental section is weak. The main paper says “we defer detailed setup description and more results to the supplementary material” (line 320). But the supplementary material does not provide details, and hence the experiments cannot be replicated.  Given the lack of details, it is not possible, for example, to assess if a gap of 0.04 (Fig 2) is significant. The simulation results are limited and comparisons are only to algorithms defined in this paper, not to competing algorithms. Simulation results are thus limited and provide limited insights.  This is a significant weakness.

The paper is densely written with sparse discussion of the results. A table of notations would considerably help the reader. The paper contains very little in the way of discussion of the 4 theorems stated in the paper, and the main body of the paper does not even contain an outline of the proofs.  (As noted elsewhere, the paper does not even describe the brand-new techniques that are claimed to be needed.) The supplemental material contains proofs of theorems and attendant lemmas, but is also lacking in any discussion. As a specific example, after the formal statements of the Theorems, slightly looser versions could have been provided in terms of min / max of the weights directly.

Time-complexity of the offline procedures (Table 1, Supplementary material) is O(E^3 \sqrt{V}) which can be prohibitive for large graphs, even for a single call. It is hard to call this “fast”.  It is not clear how often these algorithms are invoked.

What is the ‘family of instances’ for which BLUCB and BLUCB-Parallel achieve optimal sample complexity (L 134, L 179); BTW isn’t this ‘order optimal’ vs. ‘optimal’?

In a typical on-line setting, such as the bottleneck path problem, what does running BLUCB in parallel mean? How many instances are run in parallel?

Theorem 2 strengthens the results of Theorem 1 by separating base arms into arms that are better / worse than the optimal arm. However, the specific numbers provided for the example of Figure 1 are not convincing \tilde{O} (245 \log(1/\delta)) and \tilde{O} (211 \log(1/\delta)).

The paper repeatedly assumes the existence of “efficient oracles”, which is troubling (it is stated that such oracles exist for paths, bipartite matchings, and spanning trees; but it does not state how efficient they are).

What are the brand-new techniques introduced in this paper (line 75)? This should have been included in the summary of contributions.

Footnote 2: is this element-wise minimization?

The sub-Gaussian parameter R must be defined.

Grammar, syntax, and spelling demand careful attention.

 -- - ---
====== Comments added after author rebuttal and discussion =====
MO, the paper is densely written, and would definitely have improved considerably with added discussion. The authors could have used the extra pages for this.

The authors have provided detailed responses to many, but not all, of my comments. For example, the response to my 'gap' comment is weak. I still do not know if a difference of 0.04 is significant enough to claim that the new method works better.

The authors' response partially but not fully eliminates my concerns regarding the experimental section: lack of detail (addressed), and limited experimentation.

Based on the author responses (to my comments as well as those of other reviewers), I have raised my rating from '4' to '6'

**Time Spent Reviewing:**

About 12 hours - rather dense paper

---

> ### Author Response · Authors · 2021-08-10
> **Response to Reviewer 69qt - Part 1/2**
>
> Thank you for your constructive reviews. We will address them and revise our paper accordingly.
>
> 1 (Motivation)
>
> In many real-world communication networks [Hsu et al. 2004, Bapu et al. 2017], we usually sample individual links (base arm) to evaluate their Quality of Service (QoS), e.g., blocking probability and packet loss, and plan a high-quality transmission path (super arm) from source to destination.
> For example, in traffic groomed optical networks [Hsu et al. 2004], we sample blocking probabilities of individual links, and program a high-quality transmission path to avoid congested nodes. In this case, sampling individual links can both enjoy high statistical efficiency and collect the link information, which can be used to investigate the contributions of individual links to the overall congestion condition and improve the network performance.
>
> In contrast, sampling transmission paths cannot provide additional information of individual link quality, and will incur high costs due to the sampling inefficiency and long distance of transmission in large-scale networks.
> While sampling paths could transfer information, such inefficient transmission possibly incurs other issues, e.g., severe blocking and packet loss, which largely brings down the quality of user service.
> In addition, in many applications such as wireless censor networks [Camtepe et al. 2007] and MultiPath TCP (MPTCP) [Hussein et al. 2017], the number of feasible super arms (paths) is combinatorial, which further reduces the practicability of sampling super arms.
>
> Regarding global scheduling, there are various networks which require global scheduling and control, e.g., overlay networks [Guan et al. 2018] and Software Defined Networks (SDN) [Ferguson et al. 2021], and our CPE-B model may provide insights and theoretical guarantees for these applications.
> CPE-B also finds important applications in other real-world scenarios, such as traffic scheduling [Zhang et al. 2005] and network architecture searching [Pham et al. 2018].
>
> References:
> [1] Hsu, C-C., and Michael Devetsikiotis. An adaptive approach to fast simulation of traffic groomed optical networks. Proceedings of the 2004 Winter Simulation Conference, 2004.
> [2] Bapu, BR Tapas, and LC Siddanna Gowd. Link quality based opportunistic routing algorithm for QOS: aware wireless sensor networks security. Wireless Personal Communications, 2017.
> [3] Camtepe, Seyit A., and Blent Yener. Combinatorial design of key distribution mechanisms for wireless sensor networks. IEEE/ACM Transactions on networking, 2007.
> [4] Hussein, A., Elhajj, I. H., Chehab, A., & Kayssi, A. (2017, May). SDN for MPTCP: An enhanced architecture for large data transfers in datacenters. IEEE International Conference on Communications (ICC), 2017.
> [5] Guan, Y., Lei, W., Zhang, W., Liu, S., & Li, H. Scalable orchestration of software defined service overlay network for multipath transmission. Computer Networks, 2018.
> [6] Ferguson, Andrew D., et al. Orion: Google's Software-Defined Networking Control Plane. NSDI, 2021.
> [7] Zhang, X., Yang, H., Huang, H. J., & Zhang, H. M. Integrated scheduling of daily work activities and morning–evening commutes with bottleneck congestion. Transportation Research Part A: Policy and Practice, 2005.
> [8] Pham, Hieu, et al. Efficient neural architecture search via parameters sharing. International Conference on Machine Learning, 2018.
>
>
> 2 (Experiments)
>
> (1) (Experimental Setups)
>
> Thank you for your comments on our experiments. We would like to emphasize that we have provided source code for replicating our experiments.
>
> We present more detailed experimental setups as below, and will certainly incorporate them in our revision.
> (Some parameters and setups, e.g., $\Delta_{\text{min}}, n, T, \delta$, the reward distributions, number of simulations and  confidence intervals, have been shown in our paper. Here we supplement the details of $\boldsymbol{w}$ and decision classes.)
>
>
> In our experiments, $n \in [18,120], \Delta_{\text{min}} \in [0.02,0.9], T \in [5000,12000]$, whose specific values depend on different plotted points and have been shown in the x-axis of Figures 2-4.
> $\delta$ is set to 0.005 and exp(-1000) for the regular (Figure 2(a)) and high confidence (Figure 2(b)) cases, respectively.
> The expected reward vector of base arms $\boldsymbol{w}=[0.1, 0.1+\Delta_{\text{min}}, \dots, 0.1+(n-1)*\Delta_{\text{min}}]$ is an arithmetic sequence with common difference $\Delta_{\text{min}}$.
> The reward distribution of each base arm $e\in[n]$ is the Gaussian distribution $N(w(e),1)$. In the fixed-confidence setting, we perform 50 independent runs and present average sample complexity with 95% confidence intervals; in the fixed-budget setting, we perform 3000 independent runs and report the error probability across runs.
>
> For the top-k instance, we set $k=2$, and the decision class $\mathcal{M}$ is constituted by all k-cardinality subsets of the n base arms.
>
> For the path instance, we construct the following graph: there are a source node $s$, a destination node $t$, and two middle nodes $u_1$ and $u_2$ arranged as a row between $s$ and $t$. We add $m$=2-40 edges between the node pairs $(s,u_1)$, $(u_1,u_2)$ and $(u_2,t)$, respectively. We change $m$ to control the instance scale. For any $i\in [m]$, let $e_{1,i}$ denote the i-th edge between $(s,u1)$, $e_{2,i}$ denote the i-th edge between $(u_1,u_2)$ and $e_{3,i}$ denote the i-th edge between $(u_2,t)$. Then, we assign the expected rewards $\boldsymbol{w}$ to all edges $e_{1,1}$, $e_{2,1}$, $e_{3,1}$,$\dots$,$e_{1,m}$, $e_{2,m}$, $e_{3,m}$ from the biggest reward to the smallest reward.
> The decision class $\mathcal{M}$ contains all s-t paths in this graph.
>
> For the matching instance, we construct a complete bigraph: there are three nodes on the left and $V$=3-8 nodes on the right, where $V$ is changed  to control the size of instances. Each left node is connected to each right node.
> For $i\in[3]$ and $j \in [V]$, let $e_{i,j}$ denote the edge connecting the i-th node on the left and the j-th node on the right.
> We first assign the biggest three rewards in $\boldsymbol{w}$ to $e_{1,1}$, $e_{2,2}$, $e_{3,3}$ to guarantee the uniqueness of the best super arm, and then assign the remaining rewards to the remaining edges randomly.
> The decision class $\mathcal{M}$ consists of all perfect matchings in this bigraph.
>
> (2) (Gap of 0.04)
>
> We guess that the reviewer wants to ask if a gap of 0.04 is significant compared to the absolute values of expected rewards. In our experiments, with gap $\Delta_{\text{min}}=0.04$, the expected rewards of base arms are $\boldsymbol{w}=[0.1, 0.14, 0.18, 0.22, \dots]$.
> We remark that for the CPE-B problem, the problem hardness is directly decided by gap, and has nothing to do with the absolute values of expected rewards. This is because in CPE-B, the objective is to distinguish the differences between base arms, and different absolute values of expected rewards (with a fixed gap) do not affect the hardness of this identification task.
>
> (3) (Comparison Baselines)
>
> In our experiments, we did compare to existing algorithms. For example, CLUCB-B and CSAR-B are the state-of-the-art CPE algorithms for the linear reward function in [10]. CUCB-B is the state-of-the-art CPE algorithm for the regret minimization setting with general reward functions in [12].
> We run these algorithms with the bottleneck reward function for fair comparison. (We could have run their original algorithms and shown larger performance superiority, but we think it is unfair and does not make sense.)
>
> We remark that CPE-B is a new formulation and there is no existing CPE algorithm designed for the bottleneck reward function. To our best knowledge, we have compared all available CPE algorithms which can be implemented under our base arm sampling and bottleneck reward function setting.
>
> 3 (Discussion of results)
>
> Please note that, we have provided discussion and comparison for all the four theorems in their following paragraphs, i.e., Lines 172-181 following Theorem 1, Lines 243-249 following Theorem 2, Lines 259-266 following Theorem 3, Lines 310-314 following Theorem 4.
> For example, in Lines 172-181, we have highlighted that Theorem 1 is a tight “base-arm-gap dependent sample complexity” in bold font and discussed how the factor of reward gap $\Delta^C_e$ embodies this advantage, and we have also compared our results with those obtained by naive adaptations of existing algorithms [10,13,15] for the example instance in Figure 1.
> All other theorems have also been discussed and commented.
>
> In addition, please note that we have described and discussed the claimed brand-new techniques in all sections that we introduce our algorithms (Sections 3.1, 3.2, 5).
> Specifically, we have explained and discussed the following techniques:
> - "Bottleneck-adaptive sample strategy" for BLUCB   in Lines 158-166 (Section 3.1).
> - "Efficient bottleneck-searching offline subroutine" for BLUCB-Parallel in Lines 212-220 (Section 3.2).
> - "Check-near-bottleneck stopping condition" for BLUCB-Parallel in Lines 221-228 (Section 3.2).
> - "Special acceptance scheme for bottleneck" for BSAR in Lines 282-292 (Section 4).
>
> All these paragraphs have been highlighted in bold font in our paper.
>
> Certainly, following the reviewer's valuable suggestions, we will also add the proof outlines, discussion on lemmas, a notation table and looser versions of theorems in our revision to improve readability.

---

> > ### Author Response · Authors · 2021-08-10
> > **Response to Reviewer 69qt - Part 2/2**
> >
> > 4 (Time Complexity)
> >
> > Thank your for spotting this problem.
> > The "fast" meant that when compared to the naive enumeration method (that costs exponential time), our offline subroutine Bottleneck-Searching only costs polynomial time and can be performed efficiently. We will revise "fast" to "polynomial-time" to make it more accurate in our modification.
> >
> > Max-Oracle is invoked twice per round in BLUCB and once per round in BLUCB-Verify, respectively.
> > BottleneckSearch is invoked once per round in BLUCB-Explore.
> > AR-Oracle is invoked twice per round in BSAR.
> > Exist-Oracle is only used in other oracles (BottleneckSearch and AR-Oracle) instead of directly invoked by our algorithms, and thus the time complexity of Exist-Oracle has been calculated into that of BottleneckSearch and AR-Oracle, and does not incur extra time costs.
> > Certainly, we will add detailed explanation on how many times these oracles are invoked in our revision.
> >
> > 5 (Instances for Optimality)
> >
> > The "family of instances" (in Lines 134,139) refers to the constructed instances in our proof of lower bound, which has been described in details in Appendix D (Lines 809-812).
> > We restate the instances as follows.
> >
> > Consider a family of instances with Gaussian noise, such that each sub-optimal super arm has a single base arm with the reward below the optimal bottleneck value $\text{OPT}$, and the second best super arm is unique and has no overlapped base arms with $M_*$.
> >
> >
> > If the "order optimal" mentioned by the reviewer means that the upper and lower bounds only match for some factors but still have a gap for other factors, then we remark that our result is optimal in terms of both the problem hardness factor $\sum_{e \in M_* \cup N} \frac{ R^2 }{(\Delta^C_e)^2}$ and the confidence factor $\log(\frac{1}{\delta})$ in high confidence regime (up to a logarithmic factor).
> >
> > If the "order optimal" mentioned by the reviewer means that the upper and lower bounds match in $O(\cdot)$ notation but still have a constant gap, then our results are indeed order optimal, since we use $O(\cdot)$ notation and ignore the constant factors.
> > Notice that, in the bandit literature, this is a convention and most of other papers (e.g., [1,3,6,16,24,29]) are also order-optimal.
> >
> > 6 (Run in Parallel)
> >
> > We guess that the reviewer wants to ask how does the BLUCB-Parallel run its sub-algorithms BLUCB-Verify in parallel, and how many sub-algorithms are run in parallel.
> >
> > At each timestep t, each sub-algorithm k (k>=0) that satisfies $t \mod 2^k=0$ is executed to take a step (sample) and then suspended.
> > For example, sub-algorithm 0 takes a step at timestep t=1,2,3,...; sub-algorithm 1 takes a step at timestep t=2,4,6,...; sub-algorithm 2 takes a step at timestep t=4,8,12,..., and so on.
> >
> > For any timestep T, there are $\left \lfloor log T \right \rfloor+1$ alive sub-algorithms that are run in parallel, and among these alive sub-algorithms, some sub-algorithms are suspended (do not take steps).
> > For example, at timestep T=128, there are 8 alive sub-algorithms and all of them are executed to take  steps. At timestep T=129, there are 8 alive sub-algorithms but only one of them (sub-algorithm 0) is executed to take a step.
> >
> > 7 (Example of Figure 1)
> >
> > Figure 1 just gives a simple and small example for clarity. If we set $w(e_3),w(e_5),w(e_6)$ to 0.2 in Figure 1, then Theorem 1 becomes $\tilde{O}(511 \log \delta^{-1})$, and Theorem 2 gives a much tighter result $\tilde{O}(211 \log \delta^{-1})$. In addition, for larger regular instances, Theorem 2 can attain more advantages.
> >
> > 8 (Efficient Oracles)
> >
> > We have listed the time complexity of all the used efficient oracles in Table 1 (Line 935) for the s-t path, matching and spanning tree instances.
> >
> > For example, for Max-Oracle, it can be implemented by the existing offline "bottleneck shortest path [32]", "bottleneck bipartite matching [33]" and "minimum bottleneck spanning tree [7]" algorithms with time complexity $O(E)$, $O(V \sqrt{VE})$ and $O(E)$ for the path, matching and spanning tree instances, respectively.
> >
> > 9 (Brand-new Techniques)
> >
> > The brand-new techniques refer to the "bottleneck-adaptive sample strategy" in BLUCB, "efficient bottleneck-searching offline subroutine" and "delicate check-near-bottleneck stopping condition" in BLUCB-Explore, and "special acceptance scheme for bottleneck" in BSAR.
> >
> > Please notice that we have already incorporated these brand-new techniques in our summary of contributions (Lines 76-86).
> >
> > 10 (Footnote 2)
> >
> > In the definition $MinW(M, \boldsymbol{v})=\min_{e \in M} v(e)$ in Footnote 2, $\boldsymbol{v}$ is a n-dimensional weight vector, and $v(e)$ is the $e$-th entry of $\boldsymbol{v}$ that denotes the weight of base arm $e$.
> > Thus, this definition means that the bottleneck weight (reward) of super arm $M$ with respect to weight vector $\boldsymbol{v}$ is the minimum weight (reward) of the base arms that are contained in $M$.
> >
> > 11 (Definition of $R$)
> >
> > We will add the formal definition of sub-Gaussian parameter $R$ in our revision.

---

> ### Author Response · Authors · 2021-08-27
> **Response to the Additional Comments of Reviewer 69qt**
>
> Thank you so much for your increased score and effort in reviewing our paper! We will certainly incorporate the discussion in rebuttal into our revision.
>
> 1 (Response to the Question on Gap 0.04)
>
> We are not completely sure what the reviewer wanted to ask in the question "if a difference of 0.04 is significant enough to claim that the new method works better".
>
> One possibility is that the reviewer has considered the "gap" in X-axis of Figures 2(c)(d) as the performance difference between algorithms, but what we actually mean is the minimum gap of expected outcomes (weights) of base arms $\Delta_{\text{min}}$, *which is an instance parameter, not the performance difference between algorithms*.
> For example, gap 0.04 refers to the instance where the expected outcome gap $\Delta_{\text{min}}=0.04$ and the expected outcome vector $\boldsymbol{w}=[0.1,\ 0.1+\Delta_{\text{min}},\ \dots,\ 0.1+(n-1)*\Delta_{\text{min}}]=[0.1,\ 0.14,\ 0.18,\ \dots,\ 0.1+(n-1)*0.04]$.
> In Figures 2(c)(d), we change the expected outcome gap of base arms $\Delta_{\text{min}} \in [0.02,0.08]$ to generate different instances (plotted points) and compare the performance of algorithms under a wide range of generated instances.
>
> As shown in Figures 2(c)(d), our algorithm BSAR has a lower error probability than others under instances with different expected outcome gaps $\Delta_{\text{min}}\in [0.02,0.08]$. In addition, when the expected outcome gap $\Delta_{\text{min}}$ increases, the error probabilities of all algorithms decrease rapidly, due to the reduction of problem hardness $H^B$ (depending on $\Delta_{\text{min}}$). These empirical results demonstrate better performance of BSAR and match our theoretical bound $O( n^2 \exp ( - \frac{T-n}{ \tilde{\log}(n)  R^2 H^B } ) )$ (Theorem 4, Line 309).
>
> Another possibility is that the reviewer wanted to ask why the performance of our algorithm looks close to others at the point with the expected outcome gap $\Delta_{\text{min}}=0.04$ in Figure 2(d), it is because under this instance, the problem hardness $H^B$ is small and the error probabilities of all algorithms decease rapidly and approach to $0$. It is a common phenomenon in the fixed-budget pure exploration setting and also exists in prior works [6,10].
>
> Following the reviewer's comments, we will certainly clarify the meaning of "gap" in Figures 2(c)(d) to avoid confusion, and add more experimental results to show the performance difference of compared algorithms under a wider range of instances.

---

### Official Review · Reviewer_haVZ · 2021-07-17

**Rating:** 6
**Confidence:** 4

**Summary:**

The paper considers a combinatorial bandit problem, where the reward of super arm (a subset of base arms) is determined by the minimum of rewards from the individual base arms, while only base arm is played in exploration, i.e., semi-bandit feedback. The authors propose two algorithms for pure exploration in fixed-confidence and fixed-budget settings, with asymptotical optimality.


**Limitations And Societal Impact:**

.

**Main Review:**

The proposed method is proven to be efficient as it matches the asymptotical order of lower bound of sample complexity (Theorems 2 & 3). The simulation results support the superiority of the proposed methods. In what follows, I leave some concerns and comments on this work.

- The analysis in the fixed budget setting (Theorem 4) seems a straightforward extension of existing work [10], c.f., line 311. Please elaborate differences and challenges. In addition, it is unfortunate that there is no lower bound analysis for the fixed budget setting. It would be great if the paper includes some conjectures.

 - The paper studies semi-bandit feedback setting, where each base arm is individually explored. This needs to be justified. In addition, if the lower bound of sample complexity in Theorem 3 works for only semi-bandit feedback setting, it needs to be clearly stated.

- I'm not sure if the experiment on the fixed budget setting shows a clear dominance of the proposed method over the others.

**Time Spent Reviewing:**

8

---

> ### Author Response · Authors · 2021-08-10
> **Response to  Reviewer haVZ**
>
> Thank you very much for your time and effort in reviewing our paper.
>
> 1 (Novel Analysis for BSAR)
>
> The error probability analysis in [10] only holds for the linear reward function, and cannot be applied to our bottleneck problem.
>
> The main challenge of our analysis falls on how to take advantage of the bottleneck property to handle the disturbance from the accepted arm set (which are not pulled sufficiently) and guarantee the estimation accuracy of the bottleneck rewards.
>
> The differences between our analysis and prior analysis [10] are summarized as follows:
>
> (1) Prior analysis [10] relies on the linear property to cancel out the common part between two super arms when calculating their reward gap, in order to avoid the disturbance of accepted arms. In contrast, to achieve this goal, we utilize the property of special acceptance scheme in BSAR to exclude all accepted arms in the calculation of bottleneck rewards, which effectively addresses the perturbation from the inaccurate estimation of accepted arms.
>
> (2) Prior analysis [10] relies on the "exchange sets" technique, which only works for the linear reward function and leads to the dependence on the parameter of decision class structures.
> In contrast, our analysis takes advantage of the bottleneck property to build confidence intervals in the base arm level, and effectively avoids the dependence on the parameter of decision class structures.
>
> (3) Prior analysis [10] relies on an inefficient oracle (which costs exponential time complexity on the s-t path instances), while our analysis uses a simplified oracle which is efficient for the path, matching and spanning tree instances.
>
> To our best knowledge, there is no existing lower bound result for the fixed-budget setting in the CPE literature.
> We conjecture that the fixed-budget lower bound may be $\Omega(\exp(-\frac{T}{R^2 H^B}))$, since when restricting the error probability to $\delta$, this lower bound gives a sample complexity result that matches the fixed-confidence lower bound. However, the lower bound analysis of error probability largely differs from that of sample complexity, and requires additional advanced techniques.
>
> Following the reviewer's suggestions, we will incorporate our conjectures in the revision, and  further investigate this interesting problem in our future work.
>
>
>
> 2 (Semi-bandit Feedback Setting)
>
> The semi-bandit feedback setting, i.e., individually exploring base arms, is a common setting in the CPE literature, e.g., [6, 8, 9, 10, 23, 24], which can be applied to many real-world scenarios.
>
> For example, in large-scale communication networks, e.g., wireless censor networks [Camtepe et al. 2007] and MultiPath TCP (MPTCP) [Hussein et al. 2017], the number of possible routing paths (super arms) between source and destination nodes is combinatorial, which brings down the practicability of sampling super arms. In this case, we sample individual links (base arms) to evaluate the link quality and plan a high-quality routing path.
> In addition, in crowdsourcing systems, there are a large number of feasible matchings between tasks and workers, and the collected feedback is usually on individual task-worker pairs (base arms) rather than the whole task-worker matchings (super arms).
> In this case, the task-worker pairs are individually evaluated in order to efficiently identify the best task-worker assignment.
>
> Following the reviewer's suggestion, we will add more explanation on the semi-bandit feedback setting and clarify it in Theorem 3 in our revised paper.
>
> References:
> [1] Camtepe, Seyit A., and Blent Yener. Combinatorial design of key distribution mechanisms for wireless sensor networks. IEEE/ACM Transactions on networking, 2007.
> [2] Hussein, A., Elhajj, I. H., Chehab, A., & Kayssi, A. (2017, May). SDN for MPTCP: An enhanced architecture for large data transfers in datacenters. IEEE International Conference on Communications (ICC), 2017.
>
> 3 (Experiments)
>
> In the experiments for the fixed-budget setting, when the gap gets larger, the error probabilities of all algorithms decrease rapidly, which makes the performance differences between the compared algorithms hard to distinguish. These points with large gaps do not precisely reflect the error probability performance, since most of algorithms obtain almost zero error probabilities at these points.
>
> The comparison of error probability performance is more clear on the points with small gaps (harder instances). For these points, our BSAR shows clear performance superiority over other baselines. In addition, among the compared algorithms, our BSAR is the only one that has a non-trivial (gap-adaptive) theoretical guarantee.
>
> We will certainly incorporate more experimental results with a wider range of parameter settings to show clear performance comparison.

---

> > ### Comment · Reviewer_haVZ · 2021-09-02
> > **Thanks for reply**
> >
> > Thanks for the answers, which resolve most part of my concerns. However, some concerns on limited practicability of fixed-budget algorithm still remains due to the lack of theoretical support (huge gap between the upper bound analysis and the authors' conjecture on lower bound) and numerical evidence. Hence, I want to keep my score.

---

> > > ### Author Response · Authors · 2021-09-02
> > > **Response to the Additional Comments of Reviewer haVZ**
> > >
> > > Thank you very much for your time and effort in reviewing our paper! We try to address your remaining concerns in the following.
> > >
> > > 1 (Theoretical Support)
> > >
> > > We would like to remark that the fixed-budget lower bound is an ever-present open problem for the combinatorial pure exploration literature. Many prior pure exploration works, e.g., [Chen et al. 2014; Katz-Samuels et al. 2020; Kuroki et al. 2020], did not provide fixed-budget lower bound, either.
> > >
> > > Please note that when our problem reduces to conventional $K$-armed pure exploration problem [Bubeck et al. 2013], our upper bound $\tilde{O}(n^2 \exp(-\frac{T}{R^2 H^B}))$ (Theorem 4, Line 309) matches existing state-of-the-art result in [Bubeck et al. 2013].
> > > It also matches our conjectured lower bound $\Omega(\exp(-\frac{T}{R^2 H^B}))$ with respect to the most critical factor, i.e., problem hardness $H^B$.
> > > The only gap $n^2$ is not in the $\exp(\cdot)$ operation and just causes a *logarithmic* gap in the number of required samples. Such gap also exists in the state-of-the-art work [Bubeck et al. 2013] for conventional $K$-armed pure exploration problem.
> > >
> > > 2 (Numerical Evidence)
> > >
> > > We will add more experimental results in our revision to further demonstrate the performance superiority of our proposed fixed-budget algorithm.
> > >
> > >
> > > References:
> > > [1] Chen et al. Combinatorial pure exploration of multi-armed bandits. NIPS, 2014.
> > > [2] Katz-Samuels et al. An empirical process approach to the union bound: Practical algorithms for combinatorial and linear bandits. NeurIPS, 2020.
> > > [3] Kuroki et al. Online dense subgraph discovery via blurred-graph feedback. ICML, 2020.
> > > [4] Bubeck et al. Multiple identifications in multi-armed bandits. ICML, 2013.

---

> > > > ### Comment · Reviewer_haVZ · 2021-09-02
> > > > **Additional experiment**
> > > >
> > > > I understand that it is indeed hard to obtain a tight lower bound for the fixed budget setting. Thanks for the clarification.
> > > >
> > > > However, can you provide the additional experiment result, supporting your finding? Without it, I cannot assess the contribution of your analysis of the fixed-budget algorithm.

---

> > > > > ### Author Response · Authors · 2021-09-02
> > > > > **Response to Reviewer haVZ - Additional Experimental Results**
> > > > >
> > > > > Thank you for your further response.
> > > > > We are happy to report additional experimental results for the fixed-budget setting as follows.
> > > > >
> > > > > We perform 3000 independent runs for each algorithm to show their error probabilitiy results aross runs.
> > > > > The following table shows the error probability results on the path instance with $T=14000$, $n=9$, $\Delta_{\text{min}}=0.03$, $\boldsymbol{w}=[0.1,0.13,0.16,\dots,0.1+(n-1)*0.03]^\top$.
> > > > >
> > > > > |The Path Instance|Error Probability|
> > > > > | :---: | :---: |
> > > > > | BSAR (ours) | 0.030  |
> > > > > | BSR         | 0.289  |
> > > > > | CSAR-B [10] | 0.257  |
> > > > > | CUCB-B [12] | 0.300  |
> > > > > | UniformFB   | 0.258  |
> > > > >
> > > > > Another table below presents the error probability results on the matching instance with $T=14000$, $n=15$, $\Delta_{\text{min}}=0.04$, $\boldsymbol{w}=[0.1,0.14,0.18,\dots,0.1+(n-1)*0.04]^\top$.
> > > > >
> > > > > |The Matching Instance|Error Probability|
> > > > > | :---: | :---: |
> > > > > | BSAR (ours) | 0.019  |
> > > > > | BSR         | 0.215  |
> > > > > | CSAR-B [10] | 0.173  |
> > > > > | CUCB-B [12] | 0.500  |
> > > > > | UniformFB   | 0.216  |
> > > > >
> > > > > As shown in the above tables, our BSAR achieves the lowest error probability among all compared algorithms, which demonstrates the perfomance superiority of BSAR and the effectiveness of its special arm acceptance scheme for the bottheneck problem.
> > > > > We will certainly incorporate these experimental results in our revision.

---

### Official Review · Reviewer_vLPw · 2021-07-23

**Rating:** 7
**Confidence:** 3

**Summary:**

The authors are interested in the combinatorial pure exploration problem, in which one aims at identifying the best "super arm" (a subset of "base arms") among a decision class that can arise from a certain combinatorial structure. In this paper, they focus on the combinatorial pure exploration problem in the specific case where the reward of a super arm is the minimum of the rewards of the base arms it contains. This very specific function -- called the bottleneck reward function -- raises very specific challenges when solving the combinatorial pure exploration problem.
The authors explain that none of the existing work on pure exploration or on multi armed bandits give good guarantees in the specific case of the bottleneck function, and propose tailor-made algorithms that exploit the specificities of the said function to reach optimal guarantees, both in the fixed-confidence setting and in the fixed-budget setting.

**Ethical Concerns:**

I do not believe this paper raises any ethical issue.

**Limitations And Societal Impact:**

The authors provided some research directions arising from their work, one of which being the investigation of general non-linear reward functions (going beyond the bottleneck reward function and its specificities). They did not discuss the potential negative societal impact of their work, which I do believe is absolutely minimal due to its very theoretical nature.

**Main Review:**

—  Strengths
The paper is overall very well written and easy to follow, as is the supplement with very clear and well guided proofs.

The problem is well explained and related works -- to my knowledge -- adequately cited.
The studied problem — combinatorial pure exploration with bottleneck function (CPE-B) is part of the more general combinatorial pure exploration problem (CPE), for which there exists several published works in the case of several classes of reward functions (linear, continuous and separable).
This work however focuses on the combinatorial pure exploration problem in the specific case where the reward function is the bottleneck function — i.e. the reward of a super arm is the minimum of the rewards of the base arms it contains — for which classical algorithms do not give significant guarantees. The author clearly explicit the specific challenges associated with CPE-B, namely, (1) the fact that the optimal sample complexity relies on base arm gaps which are tighter than super arm gaps, (2) the existence of unnecessary base arms which do not contribute to the reward of super arms.

The authors then propose several contributions to solve these problems. In the fixed-confidence setting, the authors propose the BLUCB algorithm which uses the specificities of the bottleneck function (e.g. its monotonicity) to focus on promising “critical” arms (i.e. the arms on which the reward on super arms relies) and solve the first challenge [Theorem 1], while BLUCB-parallel offers a solution to the second problem [Theorem 2]. The authors propose a lower bound [Theorem 3] which shows the optimality of their previous results.
In the fixed-budget setting, the authors propose the BSAR algorithm which provides optimal error probability [Theorem 4].

As far as I understand, the claims and associated proofs are correct, and all these contributions are novel. The authors therefore provide a very complete and realistically implementable solution to the CPE-B problem.

The proposed algorithms rely on several offline procedures/oracles, which complexity the authors adequately discuss in the supplement, removing any possible interrogation regarding the realistic implementability of the proposed methods.


— Minor remarks
There are a few typos here and there (e.g. L72 "in *an* online environment", L84 "for *the* bottleneck identification task", L137 "denote*s* the set", L159 "has *a* higher bottleneck value than \tilde{M}_t")

The authors solve a very specific instance of a problem that they themselves introduce. While I absolutely do not think that this is a rejection argument since pioneer work is crucial, I think that the motivation for the problem -- given its specificities -- could be better brought forward. The authors motivate the study of CPE with the bottleneck reward function by its natural occurence in communication networks, traffic scheduling and neural architectural search, I am however not aware in any prior interest in this reward function in the “theoretical” MBA/PE community, and would have liked a more detailed theoretical motivation, or experiments that reflect the applied motivation (by using data simulating one of the mentioned application fields for example).
Another possibility would be to give more insights in the main text concerning how solving the CPE-B problem could be a stepping stone to solving the CPE problem with a larger class of non-linear reward functions (as mentioned in the future work section and in the supplement) -- which I believe would be a significantly more impactful contribution.

The complexity of all proposed algorithms is polynomial in E and V. My understanding is however that in some combinatorial problems, E and V might me exponential in n (i.e. subset selection problem), in which case the algorithms would not run in polynomial time with respect to n. This is not an important issue but I find the text of the first sections a little ambiguous concerning which quantities the "polynomial time" refers to.

The reliance on small delta (<0.1, <0.01), both for Theorem 2 and 3 could be explained in a clearer way in the main text.

---- POST-REBUTTAL UPDATE
After reading the other reviews, the author responses to these reviews and the additional comments from reviewer 69qt, I wish to keep my original score of 7. I agree that the paper is densely written and packs a lot of information, but I believe that the authors managed to present the material in a structured and clear way nevertheless, which in my opinion makes it a strength of the paper in the end. The "experiments" section could benefit from additional results in a "real-world" setup that would help with the motivation of the problem introduced in the paper, which the author propose to do in their revised version.

**Time Spent Reviewing:**

9

---

> ### Author Response · Authors · 2021-08-10
> **Response to Reviewer vLPw**
>
> Thank you very much for your time and effort in reviewing our paper. We appreciate your positive comments. We will fix the typos in our revision.
>
> 1 (Theoretical Motivation)
>
> Our work is theoretically motivated by the classic offline combinatorial problem with the bottleneck reward function, such as the bottleneck shortest path [32], bottleneck bipartite matching [33] and minimum bottleneck spanning tree [7] problems. These offline bottleneck problems have been extensively studied for several decades and find many real-world applications, e.g., network scheduling and traffic systems.
> Since these applications often involve online decision making (e.g., in networks we usually evaluate the link quality to choose transmission paths), it is very interesting and natural to study these bottleneck problems in an online environment.
> In addition, our theoretical results and analysis show that compared to prior online combinatorial works [10,13,15], this bottleneck reward problem is non-trivial and requires unique techniques to achieve the statistical efficiency.
>
> Thank you for your valuable suggestions on experiments, and we will incorporate more experimental results on the data simulating real-world applications in our revision.
>
> 2 (Nonlinear Reward Functions)
>
> This is a very insightful comment.
> We have designed an algorithm GenLUCB and provided the sample complexity analysis for general nonlinear reward functions in Appendix G. This general algorithm GenLUCB enjoys a better (non-trivial) sample complexity compared to the uniform sampling algorithm.
> However, compared to the algorithms designed for specific reward functions, e.g., our BLUCB, the sample complexity of GenLUCB still has a gap to BLUCB when reducing the general reward function problem to the specific bottleneck reward function setting.
>
> The algorithm design and theoretical analysis in CPE-B provide valuable insights for solving CPE with general nonlinear reward functions, especially in designing an adaptive sample strategy to fit the nonlinear reward and building the connection between offline oracles and nonlinear online estimation.
> However, to derive an optimal algorithm for general reward functions, more advanced nonlinear techniques are required. This is an interesting research direction and will be a subject of our future work.
>
> 3 (Polynomial Time)
>
> The presented time complexity with respect to E and V refers to the graph instances where each edge is a base arm and $E=n$. The "polynomial time" refers to polynomial time in $n$ for the instances $E=n$. For example, for the s-t path instances, each edge is a base arm and each feasible s-t path is a super arm. For the $k$-cardinality subset selection instances, each edge is a base arm and each $k$-cardinality subset of edges is a super arm.
>
> Certainly, we will clarify that the "polynomial time" is with respect to $n$ for the instances $E=n$ in our revision.
>
> 4 (Small $\delta$)
>
> The reliance on $\delta<0.01$ in Theorem 2 (upper bound of BLUCB-Parallel) is due to that the used explore-verify-parallel framework needs a small $\delta$ to guarantee the overall sample complexity of the BLUCB-Parallel can keep the same order as its sub-algorithm BLUCB-Verify_k. Prior pure exploration works [9,25] also have such reliance on $\delta$.
>
> The reliance on $\delta<0.1$ in Theorem 3 (lower bound) comes from the lower bound analysis. We use $\delta<0.1$ to bound the binary entropy of two events that an algorithm returns correct or wrong answer, which derives the $\ln \delta^{-1}$ factor in the lower bound. Existing CPE works [9,10,13] also have such reliance on $\delta$.
>
> Following the reviewer's suggestion, we will certainly add more explanation on the reliance on $\delta$ in our revision.

---

> > ### Author Response · Authors · 2021-08-26
> > **Response to the Additional Comments of Reviewer vLPw**
> >
> > Thank you very much for your positive comments and efforts in reviewing our paper! We will certainly incorporate your suggestions and add more experimental results on real-world data in our revision.

---

### Decision · Program_Chairs · 2021-09-27

**Decision:**

Accept (Poster)

**Comment:**

This paper examines a combinatorial exploration problem in which the learner aims to identify the best "slate" of arms, with a "bottleneck" criterion that defines the reward of each slate as the minimum possible reward of its constituent elements. The authors propose a series of tailor-made algorithms that exploit the problem's bottleneck structure to achieve optimal guarantees, both in a fixed-confidence and fixed-budget setting.

One of the reviewers expressed concerns about the writing style of the paper and the strength of the authors' experiments. During the discussion committee phase, these concerns were, to a large extent, alleviated by the authors' replies, so a decision was reached to accept the paper. At the same time, I would urge the authors to implement a version of their exchanges the reviewers and adjust the denser parts of the paper accordingly, as this would significantly strengthen their contribution.